# FEDCOMPASS: EFFICIENT CROSS-SILO FEDERATED LEARNING ON HETEROGENEOUS CLIENT DEVICES USING A COMPUTING POWER-AWARE SCHEDULER

**Zilinghan Li**[1,2,3]**, Pranshu Chaturvedi**[1,2,3]**, Shilan He**[1,2,3]**, Han Chen**[2]**, Gagandeep Singh**[2,4]**,
Volodymyr Kindratenko**[2,3]**, E. A. Huerta**[1,2,5]**, Kibaek Kim**[1]**, Ravi Madduri**[1]
[1]Argonne National Laboratory, [2]University of Illinois at Urbana-Champaign, [3]National Center
for Supercomputing Applications, [4]VMWare Research, [5]The University of Chicago
`{zl52, pranshu3, shilanh2, hanc7, ggnds, kindrtnk}@illinois.edu`
`{elihu, kimk, madduri}@anl.gov`

## ABSTRACT

Cross-silo federated learning offers a promising solution to collaboratively train robust and generalized AI models without compromising the privacy of local datasets, e.g., healthcare, financial, as well as scientific projects that lack a centralized data facility. Nonetheless, because of the disparity of computing resources among different clients (i.e., device heterogeneity), synchronous federated learning algorithms suffer from degraded efficiency when waiting for straggler clients. Similarly, asynchronous federated learning algorithms experience degradation in the convergence rate and final model accuracy on non-identically and independently distributed (non-IID) heterogeneous datasets due to stale local models and client drift. To address these limitations in cross-silo federated learning with heterogeneous clients and data, we propose `FedCompass`, an innovative semi-asynchronous federated learning algorithm with a computing power-aware scheduler on the server side, which adaptively assigns varying amounts of training tasks to different clients using the knowledge of the computing power of individual clients. `FedCompass` ensures that multiple locally trained models from clients are received almost simultaneously as a group for aggregation, effectively reducing the staleness of local models. At the same time, the overall training process remains asynchronous, eliminating prolonged waiting periods from straggler clients. Using diverse non-IID heterogeneous distributed datasets, we demonstrate that `FedCompass` achieves faster convergence and higher accuracy than other asynchronous algorithms while remaining more efficient than synchronous algorithms when performing federated learning on heterogeneous clients. The source code for `FedCompass` is available at https://github.com/APPFL/FedCompass.

## 1 INTRODUCTION

Federated learning (FL) is a model training approach where multiple clients train a global model under the orchestration of a central server (Konečný et al., 2016; McMahan et al., 2017; Yang et al., 2019; Kairouz et al., 2021). In FL, there are $m$ distributed clients solving the optimization problem: $\min_{w \in \mathbb{R}^d} F(w) = \sum_{i=1}^m p_i F_i(w)$, where $p_i$ is the importance weight of client $i$'s local data, and is usually defined as the relative data sample size, and $F_i(w)$ is the loss on client $i$'s local data. FL typically runs two steps iteratively: (i) the server distributes the global model to clients to train it using their local data, and (ii) the server collects the locally trained models and updates the global model by aggregating them. Federated Averaging (`FedAvg`) (McMahan et al., 2017) is the most popular FL algorithm where each client trains a model using local data for $Q$ local steps in each training round, after which the server aggregates all local models by performing a weighted averaging and sends the updated global model back to all clients for the next round of training. By leveraging the training data from multiple clients without explicitly sharing, FL empowers the training of more robust and generalized models while preserving the privacy of client data. The data in FL is generally heterogeneous, which means that client data is non-identically and independently distributed (non-IID). FL is most commonly conducted in two settings: cross-device FL and cross-silo FL (Kairouz et al., 2021). In cross-device FL, numerous mobile or IoT devices participate, but not in every training round because of reliability issues (Mills et al., 2019). In cross-silo FL, a smaller number of reliable

clients contribute to training in each round, and the clients are often represented by large institutions in domains such as healthcare or finance, where preserving data privacy is essential (Wang et al., 2019; Wang, 2019; Kaissis et al., 2021; Pati et al., 2022; Hoang et al., 2023).

The performance of FL suffers from device heterogeneity, where the disparity of computing power among participating client devices leads to significant variations in local training times. Classical synchronous FL algorithms such as `FedAvg`, where the server waits for all client models before global aggregation, suffer from compute inefficiency as faster clients have to wait for slower clients (stragglers). Several asynchronous FL algorithms have been developed to immediately update the global model after receiving one client model to improve computing resource utilization. The principal shortcoming in asynchronous FL is that straggler clients may train on outdated global models and obtain stale local models, which can potentially undermine the performance of the global model. To deal with stale local models, various solutions have been proposed, including client selection, weighted aggregation, and tiered FL. Client selection strategies such as `FedAR` (Imteaj & Amini, 2020) select a subset of clients according to their robustness or computing power. However, these strategies may not be applicable for cross-silo FL settings, where all clients are expected to participate to maintain the robustness of the global model. Algorithms such as `FedAsync` (Xie et al., 2019) introduce a staleness factor to penalize the local models trained on outdated global models. While the staleness factor alleviates the negative impact of stale local models, it may cause the global model to drift away from the slower clients (client drift) even with minor disparities in client computing power, resulting in a further decrease in accuracy on non-IID heterogeneous distributed datasets. Tiered FL methods such as `FedAT` (Chai et al., 2021) group clients into tiers based on their computing power, allowing each tier to update at its own pace. Nonetheless, the waiting time inside each tier can be significant, and it is not resilient to sudden changes in computing power.

In this paper, we focus on improving the performance in cross-silo FL settings. Particularly, we propose a semi-asynchronous cross-silo FL algorithm, `FedCompass`, which employs a COMputing Power-Aware Scheduler (`Compass`) to reduce client local model staleness and enhance FL efficiency in cross-silo settings. `Compass` assesses the computing power of individual clients based on their previous response times and dynamically assigns varying numbers of local training steps to ensure that a group of clients can complete local training almost simultaneously. `Compass` is also designed to be resilient to sudden client computing power changes. This enables the server to perform global aggregation using the local models from a group of clients without prolonged waiting times, thus improving computing resource utilization. Group aggregation leads to less global update frequency and reduced local model staleness, thus mitigating client drift and improving the performance on non-IID heterogeneous data. In addition, the design of `FedCompass` is orthogonal to and compatible with several server-side optimization strategies for further enhancement of performance on non-IID data. In summary, the main contributions of this paper are as follows:

- A novel semi-asynchronous cross-silo FL algorithm, `FedCompass`, which employs a resilient computing power-aware scheduler to make client updates arrive in groups almost simultaneously for aggregation, thus reducing client model staleness for enhanced performance on non-IID heterogeneous distributed datasets while ensuring computing efficiency.
- A suite of convergence analyses which demonstrate that `FedCompass` can converge to a first-order critical point of the global loss function with heterogeneous clients and data, and offer insights into how the number of local training steps impacts the convergence rate.
- A suite of experiments with heterogeneous clients and datasets which demonstrate that `FedCompass` converges faster than synchronous and asynchronous FL algorithms, and achieves higher accuracy than other asynchronous algorithms by mitigating client drift.

## 2 RELATED WORK

**Device Heterogeneity in Federated Learning.** Device heterogeneity occurs when clients have varying computing power, leading to varied local training times (Li et al., 2020; Xu et al., 2021). For synchronous FL algorithms that wait for all local models before global aggregation, device heterogeneity leads to degraded efficiency due to the waiting time for slower clients, often referred to as stragglers. Some methods address device heterogeneity by waiting only for some clients and ignoring the stragglers (Bonawitz et al., 2019). However, this approach disregards the contributions of straggler clients, resulting in wasted client computing resources and a global model significantly biased towards faster clients. Other solutions explicitly select the clients that perform updates. Chen et al. (2021) employs a greedy selection method with a heuristic based on client computing re-

sources. Hao et al. (2020) defines a priority function according to computing power and model accuracy to select participating clients. `FedAr` (Imteaj & Amini, 2020) assigns each client a trust score based on previous activities. `FLNAP` (Reisizadeh et al., 2022) starts from faster nodes and gradually involves slower ones. `FedCS` (Nishio & Yonetani, 2019) queries client resource information for client selection. `SAFA` (Wu et al., 2020) chooses the clients with lower crash probabilities. However, these methods may not be suitable for cross-silo FL settings, where the participation of every client is crucial for maintaining the robustness of the global model and there are stronger assumptions with regard to the trustworthiness and reliability of clients.

**Asynchronous Federated Learning.** Asynchronous FL algorithms provide another avenue to mitigate the impact of stragglers. In asynchronous FL, the server typically updates the global model once receiving one or few client models (Xie et al., 2019; Chen et al., 2020) or stores the local models in a size-$K$ buffer (Nguyen et al., 2022; So et al., 2021), and then immediately sends the global model back for further local training. Asynchronous FL offers significant advantages in heterogeneous client settings by reducing the idle time for each client. However, one of the key challenges in asynchronous FL is the staleness of local models. In asynchronous FL, global model updates can occur without waiting for all local models. This flexibility leads to stale local models trained on outdated global models, which may negatively impact the accuracy of the global model. Various solutions aimed to mitigate these adverse effects have been proposed. One common strategy is weighted aggregation, which is prevalent in numerous asynchronous FL algorithms (Xie et al., 2019; Chen et al., 2019; 2020; Wang et al., 2021; Nguyen et al., 2022). This approach applies a staleness weight factor to penalize outdated local models during the aggregation. While effective in reducing the negative impact of stale models, this often causes the global model to drift away from the local data on slower clients, and the staleness factor could significantly degrade the final model performance in settings with minor computing disparities among clients. Tiered FL methods offer another option for improvement. `FedAT` (Chai et al., 2021) clusters clients into several faster and slower tiers, allowing each tier to update the global model at its own pace. However, the waiting time within tiers can be substantial, and it becomes less effective when the client pool is small and client heterogeneity is large. The tiering may also not get updated timely to become resilient to speed changes. Similarly, `CSAFL` (Zhang et al., 2021) groups clients based on gradient direction and computation time. While this offers some benefits, the fixed grouping can become a liability if client computing speeds vary significantly over the course of training.

## 3 PROPOSED METHOD: FEDCOMPASS

The design choices of `FedCompass` are motivated by the shortcomings of other asynchronous FL algorithms caused by the potential staleness of client local models. In `FedAsync`, as the global model gets updated frequently, the client model is significantly penalized by a staleness weight factor, leading to theoretically and empirically sub-optimal convergence characteristics, especially when the disparity in computing power among clients is small. This can cause the global model to gradually drift away from the data of slow clients and have decreased performance on non-IID datasets. `FedBuff` attempts to address this problem by introducing a size-$K$ buffer for client updates to reduce the global update frequency. However, the optimal buffer size varies for different client numbers and heterogeneity settings and needs to be selected heuristically. It is also possible that one buffer contains several updates from the same client, or one client may train on the same global model several times. Inspired by those shortcomings, we aim to group clients for global aggregation to reduce model staleness and avoid repeated aggregation or local training. As cross-silo FL involves a small number of reliable clients, this enables us to design a scheduler that can profile the computing power of each client. The scheduler then dynamically and automatically creates groups accordingly by assigning clients with different numbers of local training steps to allow clients with similar computing power to finish local training in close proximity to each other. The scheduler tries to strike a balance between the time efficiency of asynchronous algorithms and the objective consistency of synchronous ones. Additionally, we keep the scheduler operations separated from the global aggregation, thus making it possible to combine the benefits of aggregation strategies in other FL algorithms, such as server-side optimizations for improving performance when training data are non-IID (Hsu et al., 2019; Reddi et al., 2020) and normalized averaging to eliminates objective inconsistency when clients perform various steps (Wang et al., 2020; Horváth et al., 2022). Further, from the drawbacks of tiered FL methods such as `FedAT`, the scheduler is designed to cover multiple corner cases to ensure its resilience to sudden speed changes, especially those occasional yet substantial changes that may arise in real-world cross-silo FL as a result of HPC/cloud auto-scaling.

## 3.1 COMPASS: COMPUTING POWER-AWARE SCHEDULER

To address the aforementioned desiderata, we introduce `Compass`, a COMputing Power-Aware Scheduler that assesses the computing power of individual clients according to their previous response times. `Compass` then dynamically assigns varying numbers of local steps $Q$ within a predefined range $[Q_{\min}, Q_{\max}]$ to different clients in each training round. This approach ensures that multiple client models are received nearly simultaneously, allowing for a grouped server aggregation. By incorporating several local models in a single global update and coordinating the timing of client responses, we effectively reduce the frequency of global model updates. This, in turn, reduces client model staleness while keeping the client idle time to a minimum during the training process.

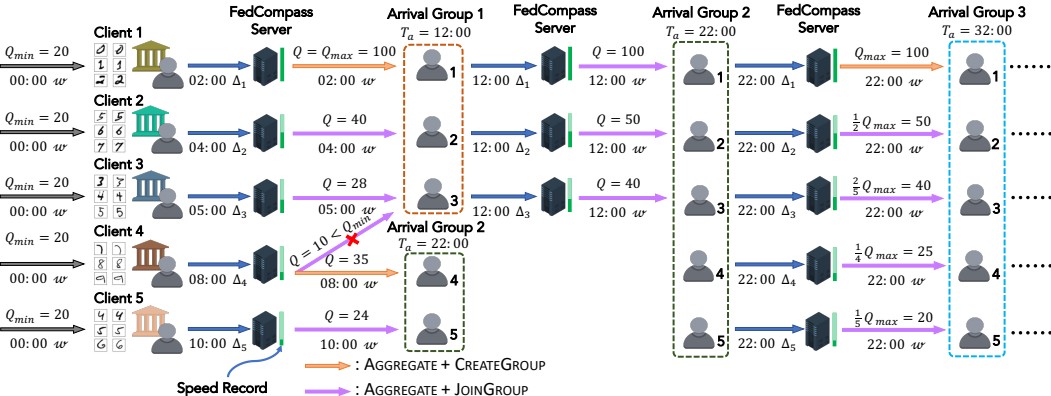

Figure 1: Overview of an example FL run using `Compass` scheduler on five clients with the minimum number of local steps $Q_{\min} = 20$ and maximum number of local steps $Q_{\max} = 100$.

Figure 1 illustrates an example run of federated learning using `Compass` on five clients with different computing power. For the sake of brevity, this illustration focuses on a simple scenario, and more details are described in Section 3.2. All times in the figure are absolute wall-clock times from the start of FL. First, all clients are initialized with $Q_{\min} = 20$ local steps. When client 1 finishes local training and sends the update $\Delta_1$ to the server at time 02:00, `Compass` records its speed and creates the first arrival group with the expected arrival time $T_a = 12:00$, which is computed based on the client speed and the assigned number of local steps $Q = Q_{\max} = 100$. We assume negligible communication and aggregation times in this example, so client 1 receives an updated global model $w$ immediately at 02:00 and begins the next round of training. Upon receiving the update from client 2, `Compass` slots it into the first arrival group with $Q = 40$ local steps, aiming for a completion time of $T_a = 12:00$ as well. Client 3 follows suit with 28 steps. When `Compass` computes local steps for client 4 in order to finish by $T_a = 12:00$, it finds that $Q = 10$ falls below $Q_{\min} = 20$. Therefore, `Compass` creates a new arrival group for client 4 with expected arrival time $T_a = 22:00$. `Compass` optimizes the value of $T_a$ such that the clients in an existing arrival group may join this newly created group in the future for group merging. After client 5 finishes, it is assigned to the group created for client 4. When clients 1–3 finish their second round of local training at 12:00, `Compass` assigns them to the arrival group of clients 4 and 5 (i.e., arrival group 2) with the corresponding values of $Q$ so that all the clients are in the same arrival group and expected to arrive nearly simultaneously for global aggregation in the future. The training then achieves equilibrium with a fixed number of existing groups and group assignments, and future training rounds will proceed with certain values of $Q$ for each client in the ideal case. The detailed implementation and explanation of `Compass` can be found in Appendix A. We also provide analyses and empirical results for non-ideal scenarios with client computing powering changes in Appendix G.2, where we show how the design of `Compass` makes it resilient when clients experience speedup or slowdown during the course of training, and compare it with the tiered FL method `FedAT` (Chai et al., 2021).

## 3.2 FEDCOMPASS: FEDERATED LEARNING WITH COMPASS

`FedCompass` is a semi-asynchronous FL algorithm that leverages the `Compass` scheduler to coordinate the training process. It is semi-asynchronous as the server may need to wait for a group of clients for global aggregation, but the `Compass` scheduler minimizes the waiting time by managing to have clients within the same group finish local training nearly simultaneously, so the overall training process still remains asynchronous without long client idle times. The function FEDCOMPASS-

---

**Algorithm 1:** `FedCompass` Algorithm

---

1    **Function** FEDCOMPASS-CLIENT$(i, w, Q, \eta_\ell, B)$:
    **Input:** Client id $i$, global model $w$, number of local steps $Q$, local learning rate $\eta_\ell$, batch size $B$
2    **for** $q \in [1, \ldots, Q]$ **do**
3       $\mathcal{B}_{i,q} :=$ random training batch with batch size $B$;
4       $w_{i,q} := w_{i,q-1} - \eta_\ell \nabla f_i(w_{i,q-1}, \mathcal{B}_{i,q})$   $\triangleright$ $w_{i,q}$ is client $i$'s model after $q$ steps, $w_{i,0} = w$;
5    Send $\Delta_i := w_{i,0} - w_{i,Q}$ to server;
6
7    **Function** FEDCOMPASS-SERVER$(N, \eta_\ell, B, w)$:
    **Input:** Number of comm. rounds $N$, local learning rate $\eta_\ell$, batch size $B$, initial model weight $w$
8    FEDCOMPASS-CLIENT$(i, w, Q_{\min}, \eta_\ell, B)$ for all client $i$'s;
9    Initialize client info dictionary $\mathcal{C}$, group info dictionary $\mathcal{G}$, and global model timestamp $\tau_g := 0$;
10   **for** $n \in [1, \ldots, N]$ **do**
11     **if** *receive update $\Delta_i$ from client $i$* **then**
12       Record client speed into $\mathcal{C}[i].S$;
13       **if** *client $i$ has an arrival group $g$* **then**
14         $\mathcal{G}[g].CL.$remove$(i)$, $\mathcal{G}[g].ACL.$add$(i)$   $\triangleright$ move client $i$ to arrived client list;
15         **if** $T_{now} > \mathcal{G}[g].T_{\max}$ **then**
16           $\overline{\Delta} := \overline{\Delta} + st(\tau_g - \mathcal{C}[i].\tau) * p_i * \Delta_i; \mathcal{C}[i].\tau := \tau_g;$   $\triangleright$ $st(\cdot)$ is staleness factor;
17           ASSIGNGROUP$(i)$   $\triangleright$ this computes $\mathcal{C}[i].Q$, details in Alg. 2 from Appendix;
18           FEDCOMPASS-CLIENT$(i, w, \mathcal{C}[i].Q, \eta_\ell, B)$;
19           delete $\mathcal{G}[g]$ if len$(\mathcal{G}[g].CL) = 0$;
20         **else**
21           $\overline{\Delta}^g := \overline{\Delta}^g + st(\tau_g - \mathcal{C}[i].\tau) * p_i * \Delta_i$   $\triangleright$ $st(\cdot)$ is staleness factor;
22           **if** len$(\mathcal{G}[g].CL = 0)$ **then**
23             GROUPAGGREGATE$(g)$   $\triangleright$ details in Alg. 3 from Appendix; delete $\mathcal{G}[g]$;
24       **else**
25         $w := w - st(\tau_g - \mathcal{C}[i].\tau) * p_i * \Delta_i; \; \tau_g := \tau_g + 1; \; \mathcal{C}[i].\tau := \tau_g;$
26         ASSIGNGROUP$(i)$   $\triangleright$ this computes $\mathcal{C}[i].Q$, details in Alg. 2 from Appendix;
27         FEDCOMPASS-CLIENT$(i, w, \mathcal{C}[i].Q, \eta_\ell, B)$;

---

CLIENT in Algorithm 1 outlines the client-side algorithm for `FedCompass`. The client performs $Q$ local steps using mini-batch stochastic gradient descent and sends the difference between the initial and final models $\Delta_i$ as the client update to the server.

The function FEDCOMPASS-SERVER in Algorithm 1 presents the server algorithm for `FedCompass`. Initially, in the absence of prior knowledge of clients' computing speeds, the server assigns a minimum of $Q_{\min}$ local steps to all clients for a warm-up. Subsequently, client speed information is gathered and updated each time a client responds with the local update. For each client's first response (lines 25 to 28), the server immediately updates the global model using the client update and assigns it to an arrival group. All clients within the same arrival group are expected to arrive almost simultaneously at a specified time $T_a$ for the next global group aggregation. This is achieved by assigning variable local steps based on individual computing speeds. The AS-SIGNGROUP function first attempts to assign the client to an existing group with local steps within the range of $[Q_{\min}, Q_{\max}]$. If none of the existing groups have local step values that can accommodate the client, a new group is created for the client with a local step value that facilitates future group merging opportunities. Further details for this function are elaborated in Appendix A.

After the initial arrivals, each client is expected to adhere to the group-specific arrival time $T_a$. Nonetheless, to accommodate client speed fluctuations and potential estimation inaccuracies, a latest arrival time $T_{\max}$ per group is designated. If a client arrives prior to $T_{\max}$ (as per lines 20 to 24), the server stores the client update in a group-specific buffer $\overline{\Delta}^g$, and checks whether this client is the last within its group to arrive. If so, the server invokes GROUPAGGREGATE to update the global model with contributions from all clients in the group and assign new training tasks to them with the latest global model. The details of GROUPAGGREGATE are shown in Algorithm 3 in the Appendix. If there remain pending clients within the group, the server will wait until the last client arrives or the latest arrival time $T_{\max}$ before the group aggregation. Client(s) that arrive after the latest time $T_{\max}$ and miss the group aggregation (lines 15 to 19), due to an unforeseen delay or slowdown, have their updates stored in a general buffer $\overline{\Delta}$, and the server assigns the client to an arrival group

immediately using the ASSIGNGROUP function for the next round of local training. The updates in the general buffer $\overline{\Delta}$ will be incorporated into the group aggregation of the following arrival group.

## 4 CONVERGENCE ANALYSIS

We provide the convergence analysis of FedCompass for smooth and non-convex loss functions $F_i(w)$. We denote by $\nabla F(w)$ the gradient of global loss, $\nabla F_i(w)$ the gradient of client $i$'s local loss, and $\nabla f_i(w, \mathcal{B}_i)$ the gradient of loss on batch $\mathcal{B}_i$. With that, we make the following assumptions.

**Assumption 1.** *Lipschitz smoothness. The loss function of each client $i$ is Lipschitz smooth.* $||\nabla F_i(w) - \nabla F_i(w')|| \le L||w - w'||$.

**Assumption 2.** *Unbiased client stochastic gradient.* $\mathbb{E}_{\mathcal{B}_i}[\nabla f_i(w, \mathcal{B}_i))] = \nabla F_i(w)$.

**Assumption 3.** *Bounded gradient, and bounded local and global gradient variance.* $||\nabla F_i(w)||^2 \le M$, $\mathbb{E}_{\mathcal{B}_i}[||\nabla f_i(w, \mathcal{B}_i)) - \nabla F_i(w)||^2] \le \sigma_l^2$, *and* $\mathbb{E}[||\nabla F_i(w) - \nabla F(w)||^2] \le \sigma_g^2$.

**Assumption 4.** *Bounded client heterogeneity and staleness. Suppose that client $a$ is the fastest client, and client $b$ is the slowest, $\mathcal{T}_a$ and $\mathcal{T}_b$ are the times per one local step of these two clients, respectively. We then assume that $\mathcal{T}_b/\mathcal{T}_a \le \mu$. This assumption also implies that the staleness* $(\tau_g - \mathcal{C}[i].\tau)$ *of each client model is also bounded by a number $\tau_{max}$.*

Assumptions 1–3 are common assumptions in the convergence analysis of federated or distributed learning algorithms (Stich, 2018; Li et al., 2019; Yu et al., 2019; Karimireddy et al., 2020; Nguyen et al., 2022). Assumption 4 indicates that the speed difference among clients is finite and is equivalent to the assumption that all clients are reliable, which is reasonable in cross-silo FL.

**Theorem 1.** *Suppose that* $\eta_\ell \le \frac{1}{2LQ_{\max}}$, $Q = Q_{\max}/Q_{\min}$, *and* $\mu' = Q^{\lfloor \log_Q \mu \rfloor}$. *Then, after $T$ updates for global model $w$, FedCompass achieves the following convergence rate:*

$$\frac{1}{T} \sum_{t=0}^{T-1} \mathbb{E}||\nabla F(w^{(t)})||^2 \le \frac{2\gamma_2 F^*}{\eta_\ell Q_{\min} T} + \frac{2\gamma_2 \eta_\ell m^2 \sigma_l^2 L Q_{\max}^2}{\gamma_1^2 Q_{\min}} + \frac{6\gamma_2 m \eta_\ell^2 \sigma^2 L^2 Q_{\max}^3}{\gamma_1 Q_{\min}}(\tau_{\max}^2 + 1),$$

(1)

*where $m$ is the number of clients, $w^{(t)}$ is the global model after $t$ global updates, $\gamma_1 = 1 + \frac{m-1}{\mu'}$, $\gamma_2 = 1 + \mu'(m-1)$, $F^* = F(w^{(0)}) - F(w^*)$, and $\sigma^2 = \sigma_l^2 + \sigma_g^2 + M$.*

**Corollary 1.** *When $Q_{\min} \le Q_{\max}/\mu$ and $\eta_\ell = \mathcal{O}(1/Q_{\max}\sqrt{T})$ while satisfying $\eta_\ell \le \frac{1}{2LQ_{\max}}$, then*

$$\min_{t \in [T]} \mathbb{E}||\nabla F(w^{(t)})||^2 \le \mathcal{O}\Big(\frac{mF^*}{\sqrt{T}}\Big) + \mathcal{O}\Big(\frac{m\sigma_l^2 L}{\sqrt{T}}\Big) + \mathcal{O}\Big(\frac{m\tau_{\max}^2 \sigma^2 L^2}{T}\Big).$$

(2)

The proof is provided in Appendix C. Based on Theorem 1 and Corollary 1, we remark the following algorithmic characteristics.

**Worst-Case Convergence.** Corollary 1 provides an upper bound for the expected squared norm of the gradients of the global loss, which is monotonically decreasing with respect to the total number of global updates $T$. This leads to the conclusion that FedCompass converges to a first-order stationary point in expectation, characterized by a zero gradient, of the smooth and non-convex global loss function $F(w) = \sum_{i=1}^m p_i F_i(w)$. Specifically, the first term of equation 2 represents the influence of model initialization, and the second term accounts for stochastic noise from local updates. The third term reflects the impact of device heterogeneity ($\tau_{\max}^2$) and data heterogeneity ($\sigma^2$) on the convergence process, which theoretically provides a convergence guarantee for FedCompass on non-IID heterogeneous distributed datasets and heterogeneous client devices.

**Effect of Number of Local Steps.** Theorem 1 establishes the requirement $\eta_\ell \le \frac{1}{2LQ_{\max}}$, signifying a trade-off between the maximum number of client local steps $Q_{\max}$ and the client learning rate $\eta_\ell$. A higher $Q_{\max}$ necessitates a smaller $\eta_\ell$ to ensure the convergence of FL training. The value of $Q_{\min}$ also presents a trade-off: a smaller $Q_{\min}$ leads to a reduced $\mu'$, causing an increase in $\gamma_1$ and a decrease in $\gamma_2$, which ultimately lowers the gradient bound. Appendix B shows that $Q_{\min}$ also influences the grouping of FedCompass. There are at most $\lceil \log_Q \mu \rceil$ existing groups at the equilibrium state, so a smaller $Q_{\min}$ results in fewer groups and a smaller $\tau_{\max}$. However, since $Q_{\min}$ appears in the denominator in equation 1, a smaller $Q_{\min}$ could potentially increase the gradient bound. Corollary 1 represents an extreme case where the value of $Q_{\min}$ makes $\mu'$ reach the

minimum value of 1. We give empirical ablation study results on the effect of the number of local steps in Section 5.2 and Appendix G.1. The ablation study results show that the performance of `FedCompass` is not sensitive to the choice of the ratio $Q = Q_{\max}/Q_{\min}$, and a wide range of $Q$ still leads to better performance than other asynchronous FL algorithms such as `FedBuff`.

# 5 EXPERIMENTS

## 5.1 EXPERIMENT SETUP

**Datasets.** To account for the dataset inherent statistical heterogeneity, we use two types of datasets: (i) artificially partitioned MNIST (LeCun, 1998) and CIFAR-10 (Krizhevsky et al., 2009) datasets, and (ii) two naturally split datasets from the cross-silo FL benchmark FLamby (Ogier du Terrail et al., 2022), Fed-IXI and Fed-ISIC2019. Artificially partitioned datasets are created by dividing classic datasets into client splits. We introduce *class partition* and *dual Dirichlet partition* strategies. The *class partition* makes each client hold data of only a few classes, and the *dual Dirichlet parition* employs Dirichlet distributions to model the class distribution within each client and the variation in sample sizes across clients. However, these methods might not accurately capture the intricate data heterogeneity in real federated datasets. Naturally split datasets, on the other hand, are composed of federated datasets obtained directly from multiple real-world clients, which retain the authentic characteristics and complexities of the underlying data distribution. The Fed-IXI dataset takes MRI images as inputs to generate 3D brain masks, and the Fed-ISIC2019 dataset takes dermoscopy images as inputs and aims to predict the corresponding melanoma classes. The details of the partition strategies and the FLamby datasets are elaborated in Appendix D.

**Client Heterogeneity Simulation.** To simulate client heterogeneity, let $t_i$ be the average time for client $i$ to finish one local step, and assume that $t_i$ follows a random distribution. We use three different distributions in the experiments: normal distribution $t_i \sim \mathcal{N}(\mu, \sigma^2)$ with $\sigma = 0$ (i.e., homogeneous clients), normal distribution $t_i \sim \mathcal{N}(\mu, \sigma^2)$ with $\sigma = 0.3\mu$, and exponential distribution $t_i \sim Exp(\lambda)$. The client heterogeneity increases accordingly for these three heterogeneity settings. The mean values of the distributions ($\mu$ or $1/\lambda$) for different experiment datasets are given in Appendix E.1. Additionally, to account for the variance in training time within each client across different training rounds, we apply another normal distribution $t_i^{(k)} \sim \mathcal{N}(t_i, (0.05t_i)^2)$ with smaller variance. This distribution simulates the local training time per step for client $i$ in the training round $k$, considering the variability experienced by the client during different training rounds. The combination of distributions adequately captures the client heterogeneity in terms of both average training time among clients and the variability in training time within each client across different rounds.

**Training Settings.** We employ a convolutional neural network (CNN) for the partitioned MNIST dataset, and a ResNet-18 (He et al., 2016) for the partitioned CIFAR-10 dataset. For both datasets, we conduct experiments with $m$={5, 10, 20} clients and the three client heterogeneity settings above, using the *class partition* and *dual Dirichlet partition* strategies. For datasets Fed-IXI and Fed-ISIC2019 from FLamby, we use the default experiment settings provided by FLamby. We list the detailed model architectures and experiment hyperparameters in Appendix E.

## 5.2 EXPERIMENT RESULTS

**Performance Comparison.** Table 1 shows the relative wall-clock time for various FL algorithms to first reach the target validation accuracy on the partitioned MNIST and CIFAR-10 datasets with different numbers of clients and client heterogeneity settings. Table 2 reports the relative wall-clock time to reach 0.98 DICE accuracy (Dice, 1945) on the Fed-IXI dataset and 50% balanced accuracy on the Fed-ISIC2019 dataset in three heterogeneity settings. The numbers are the average results of ten runs with different random seeds, and the wall-clock time for `FedCompass` is used as a baseline. A dash "-" signifies that a certain algorithm cannot reach the target accuracy in at least half of the experiment runs. The dataset target accuracy is carefully selected based on each algorithm's top achievable accuracy during training (given in Appendix F). Among the benchmarked algorithms, `FedAvg` (McMahan et al., 2017) is the most commonly used synchronous FL algorithm, and `FedAvgM` (Hsu et al., 2019) is a variant of `FedAvg` using server-side momentum to address non-IID data. `FedAsync` (Xie et al., 2019) and `FedBuff` (Nguyen et al., 2022) are two popular asynchronous FL algorithms, and `FedAT` (Chai et al., 2021) is a tiered FL algorithm. Since the `Compass` scheduler is separated from global aggregation, making `FedCompass` compatible with server-side optimization strategies, we also benchmark the performance of using server-side momentum and normalized averaging (Wang et al., 2020) in `FedCompass` (`FedCompass+M`

Table 1: Relative wall-clock time to first reach the target validation accuracy on different datasets for various FL algorithms with different numbers of clients and client heterogeneity settings.

| Dataset | Method | m = 5 homo | normal | exp | m = 10 homo | normal | exp | m = 20 homo | normal | exp |
|---|---|---|---|---|---|---|---|---|---|---|
| MNIST-*Class Partition* (90%) | FedAvg | 1.35× | 2.82× | 4.32× | 1.25× | 2.10× | 5.01× | 1.44× | 4.20× | 9.03× |
| | FedAvgM | 1.27× | 2.75× | 4.23× | 1.50× | 2.51× | 6.01× | 2.20× | 5.36× | 13.73× |
| | FedAsync | 2.15× | 2.34× | 2.35× | - | 2.17× | 1.32× | - | 4.44× | 1.32× |
| | FedBuff | 1.30× | 1.57× | 1.81× | 1.58× | 1.39× | 1.43× | 1.37× | 1.18× | 1.39× |
| | FedAT | 1.36× | 2.16× | 3.07× | 1.13× | 1.77× | 2.46× | 0.97× | 1.58× | 4.32× |
| | FedCompass | 1.00× | 1.00× | 1.00× | 1.00× | 1.00× | 1.00× | 1.00× | 1.00× | 1.00× |
| | FedCompass+M | 1.10× | 1.19× | 0.81× | 1.18× | 0.99× | 0.81× | 1.10× | 1.09× | 1.08× |
| | FedCompass+N | 0.89× | 1.24× | 0.94× | 1.44× | 1.28× | 0.92× | 0.94× | 1.01× | 1.09× |
| MNIST-*Dirichlet Partition* (90%) | FedAvg | 0.91× | 1.85× | 4.90× | 1.18× | 2.44× | 5.68× | 1.14× | 2.48× | 5.94× |
| | FedAvgM | 0.91× | 1.85× | 4.89× | 1.22× | 2.45× | 5.83× | 1.21× | 2.70× | 6.12× |
| | FedAsync | 1.78× | 1.68× | 2.01× | - | 2.28× | 1.29× | - | - | 1.76× |
| | FedBuff | 1.23× | 1.52× | 1.86× | 1.81× | 1.51× | 1.49× | 2.77× | 1.22× | 1.40× |
| | FedAT | 1.29× | 1.97× | 2.08× | 1.08× | 1.70× | 2.44× | 1.02× | 1.83× | 2.83× |
| | FedCompass | 1.00× | 1.00× | 1.00× | 1.00× | 1.00× | 1.00× | 1.00× | 1.00× | 1.00× |
| | FedCompass+M | 0.72× | 0.92× | 0.95× | 0.93× | 0.91× | 0.88× | 0.87× | 1.31× | 0.81× |
| | FedCompass+N | 0.92× | 1.55× | 1.26× | 0.99× | 1.12× | 0.95× | 0.93× | 0.90× | 1.08× |
| CIFAR10-*Class Partition* (60%) | FedAvg | 1.84× | 2.64× | 9.40× | 3.18× | 4.18× | 11.51× | 3.20× | 3.24× | 7.13× |
| | FedAvgM | 2.57× | 3.73× | 13.12× | 2.19× | 2.88× | 9.04× | 3.13× | 3.19× | 6.91× |
| | FedAsync | - | 3.18× | 3.07× | - | - | 5.18× | - | - | - |
| | FedBuff | 1.68× | 2.18× | 2.47× | - | - | 2.62× | - | - | 4.62× |
| | FedAT | 1.05× | 1.42× | 2.33× | 1.21× | 1.83× | 3.21× | 1.20× | 1.23× | 2.21× |
| | FedCompass | 1.00× | 1.00× | 1.00× | 1.00× | 1.00× | 1.00× | 1.00× | 1.00× | 1.00× |
| | FedCompass+M | 0.82× | 0.82× | 0.95× | 0.89× | 0.78× | 0.96× | 0.78× | 0.81× | 0.83× |
| | FedCompass+N | 0.98× | 1.01× | 1.18× | 0.99× | 1.22× | 1.49× | 1.32× | 1.19× | 1.16× |
| CIFAR10-*Dirichlet Partition* (50%) | FedAvg | 2.19× | 2.81× | 10.86× | 4.55× | 5.05× | 11.47× | 5.88× | 5.83× | 9.05× |
| | FedAvgM | 2.17× | 2.97× | 10.77× | 3.77× | 4.21× | 11.06× | 5.19× | 5.44× | 8.63× |
| | FedAsync | - | 3.18× | 1.70× | - | - | 5.35× | - | - | - |
| | FedBuff | 1.62× | 1.99× | 1.67× | - | - | 2.42× | - | - | - |
| | FedAT | 1.04× | 1.69× | 2.90× | 1.36× | 1.66× | 2.30× | 1.45× | 1.58× | 2.21× |
| | FedCompass | 1.00× | 1.00× | 1.00× | 1.00× | 1.00× | 1.00× | 1.00× | 1.00× | 1.00× |
| | FedCompass+M | 0.82× | 0.80× | 1.03× | 1.07× | 0.85× | 0.94× | 0.86× | 0.69× | 0.98× |
| | FedCompass+N | 1.05× | 0.98× | 1.65× | 0.89× | 1.29× | 1.45× | 0.77× | 0.87× | 1.23× |

Table 2: Relative wall-clock time to first reach the target validation accuracy on the Fed-IXI and Fed-ISIC2019 datasets for various FL algorithms in different client heterogeneity settings.

| Method | Fed-IXI (0.98) homo | normal | exp | Fed-ISIC2019 (50%) homo | normal | exp |
|---|---|---|---|---|---|---|
| FedAvg | 1.18× | 1.80× | 2.20× | 1.35× | 2.00× | 1.90× |
| FedAvgM | 1.14× | 1.83× | 2.20× | 2.74× | 2.74× | 4.65× |
| FedAsync | 1.40× | 1.31× | 1.45× | 1.50× | 1.75× | 1.69× |
| FedBuff | 1.20× | 1.25× | 1.38× | 2.59× | 3.76× | 1.03× |
| FedAT | 1.07× | 1.30× | 1.35× | 1.08× | 1.65× | 2.64× |
| FedCompass | 1.00× | 1.00× | 1.00× | 1.00× | 1.00× | 1.00× |
| FedCompass+M | 1.00× | 0.98× | 1.02× | 2.22× | 2.02× | 2.03× |
| FedCompass+N | 0.98× | 0.97× | 1.03× | 1.04× | 0.84× | 0.97× |

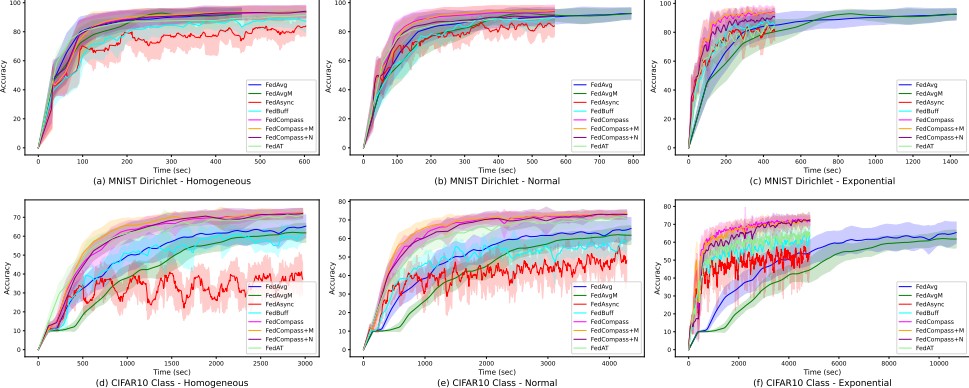

Figure 2: Change in validation accuracy and standard deviation for different FL algorithms on the *dual Dirichlet partitioned* MNIST dataset and the *class partitioned* CIFAR-10 dataset with five clients and three client heterogeneity settings. (Synchronous algorithms take the same amount of time, and asynchronous algorithms take the same amount of time in the same experiment setting.)

and `FedCompass+N`). From the table we obtain the following key observations: (i) The speedup provided by asynchronous FL algorithms increases as client heterogeneity increases. (ii) `FedAT` converges slower as client heterogeneity increases due to non-negligible inner-tier waiting time, while `FedCompass` minimizes inner-group waiting time by dynamic local step assignment and the usage of the latest arrival time. (iii) Similar to synchronous FL where server-side optimizations can sometimes speed up the convergence, applying server momentum or normalized averaging to `FedCompass` in group aggregation can sometimes speed up the convergence as well. This indicates that the design of `FedCompass` allows it to easily take advantage of other FL strategies. (iv) Because of client drift and model staleness, `FedAsync` and `FedBuff` occasionally fail to achieve the target accuracy, particularly in settings with minimal client heterogeneity and unnecessary application of the staleness factor. On the other hand, `FedCompass` can quickly converge in all cases, as `Compass` dynamically and automatically groups clients for global aggregation, reducing client model staleness and mitigating the client drift problem. This shows that `FedCompass` outperforms other asynchronous algorithms on heterogeneous non-IID data without any prior knowledge of clients. Additionally, Figure 2 illustrates the change in validation accuracy during the training for different FL algorithms on the *dual Dirichlet partitioned* MNIST dataset (subplots a to c) and the *class partitioned* CIFAR-10 dataset (subplots d to f) with five clients and three client heterogeneity settings. In the plots, synchronous algorithms take the same amount of time, and asynchronous algorithms take the same amount of time in the same experiment setting. These plots further illustrate how `FedCompass` converges faster than other algorithms and also achieves higher accuracy than other asynchronous algorithms. For example, subplots d and e explicitly demonstrate that `FedAsync` and `FedBuff` only converge to much lower accuracy in low-heterogeneity settings, mainly because of client drift from unnecessary staleness factors. More plots of the training process and the maximum achievable accuracy for each FL algorithm are given in Appendix F.

**Effect of Number of Local Steps.** We conduct ablation studies on the values of *1/Q* = $Q_{\min}/Q_{\max}$ to investigate its impact on `FedCompass`'s performance, with results listed in Table 3. From the table, we observe that smaller *1/Q* values (ranging from 0.05 to 0.2) usually lead to faster convergence speed. Additionally, a wide range of *1/Q* (ranging from 0.05 to 0.8) can achieve a reasonable convergence rate compared with `FedBuff` or `FedAT`, which showcases that `FedCompass` is not sensitive to the value of *Q*. Therefore, users do not need to heuristically select *Q* for different client numbers and heterogeneity settings. In the extreme case that $Q = 1$, i.e. $Q_{\min} = Q_{\max}$, there is a significant performance drop for `FedCompass`, as `Compass` loses the flexibility to assign different local steps to different clients. Therefore, each client has its own arrival group, and `FedCompass` becomes equivalent to `FedAsync`. More ablation study results are given in Appendix G.1.

Table 3: Relative wall-clock time for `FedCompass` to first reach the target validation accuracy on the *dual Dirichlet partitioned* MNIST/CIFAR10 datasets for various values of *1/Q* = $Q_{\min}/Q_{\max}$.

| Client number | | $m = 5$ | | | $m = 10$ | | | $m = 20$ | | |
| Dataset | *1/Q* | homo | normal | exp | homo | normal | exp | homo | normal | exp |
|---|---|---|---|---|---|---|---|---|---|---|
| | 0.05 | 0.89/1.27× | 1.01/0.97× | 0.85/0.76× | 0.96/1.05× | 0.97/0.98× | 0.82/1.19× | 1.00/1.05× | 1.02/0.97× | 0.96/0.90× |
| MNIST / | 0.10 | 0.91/1.15× | 1.05/1.01× | 0.92/0.73× | 0.96/0.98× | 0.95/1.11× | 0.85/1.27× | 0.97/1.01× | 1.02/1.04× | 0.96/1.01× |
| CIFAR10 | 0.20 | 1.00/1.00× | 1.00/1.00× | 1.00/1.00× | 1.00/1.00× | 1.00/1.00× | 1.00/1.00× | 1.00/1.00× | 1.00/1.00× | 1.00/1.00× |
| *Dirichlet* | 0.40 | 1.02/1.28× | 1.03/1.06× | 1.70/1.36× | 1.04/1.02× | 1.05/1.06× | 1.28/2.19× | 1.00/0.96× | 1.14/1.14× | 1.25/1.08× |
| *Partition* | 0.60 | 1.04/1.23× | 1.01/0.99× | 1.83/1.65× | 1.00/0.97× | 1.01/1.61× | 1.19/2.52× | 0.92/1.11× | 1.11/1.06× | 1.16/1.74× |
| (90%/50%) | 0.80 | 1.04/1.23× | 1.13/1.24× | 1.58/1.64× | 0.91/1.32× | 1.25/1.91× | 1.22/3.44× | 0.86/1.33× | 1.22/2.95× | 1.21/1.64× |
| | 1.00 | 1.78×/- | 1.68/3.18× | 2.01/1.70× | -/- | 2.28×/- | 1.29/5.35× | -/- | -/- | 1.76×/- |
| FedBuff | | 1.23/1.62× | 1.52/1.99× | 1.86/1.67× | 1.81×/- | 1.51×/- | 1.49/2.42× | 2.77×/- | 1.22×/- | 1.40×/- |
| FedAT | | 1.29/1.04× | 1.97/1.69× | 2.08/2.90× | 1.08/1.36× | 1.70/1.66× | 2.44/2.30× | 1.02/1.45× | 1.83/1.58× | 2.83/2.21× |

## 6 CONCLUSION

This paper introduces `FedCompass`, which employs a novel computing power-aware scheduler to make several client models arrive almost simultaneously for group aggregation to reduce global aggregation frequency and mitigate the stale local model issues in asynchronous FL. `FedCompass` thus enhances the FL training efficiency with heterogeneous client devices and improves the performance on non-IID heterogeneous distributed datasets. Through a comprehensive suite of experiments, we have demonstrated that `FedCompass` achieves faster convergence than other prevalent FL algorithms in various training tasks with heterogeneous training data and clients. This new approach opens up new pathways for the development and use of robust AI models in settings that require privacy, with the additional advantage that it can readily leverage the existing disparate computing ecosystem.

REPRODUCIBILITY STATEMENT

The source code and instructions to reproduce all the experiment results can be found in the supplemental materials. The details of the classic dataset partition strategies are provided in Appendix D, and the corresponding implementations are included in the source code. The setups and hyperparameters used in the experiments are given in Appendix E. Additionally, the proofs of Theorem 1 and Corollary 1 are provided in Appendix C.

ACKNOWLEDGMENTS

This work was supported by Laboratory Directed Research and Development (LDRD) funding from Argonne National Laboratory, provided by the Director, Office of Science, of the U.S. Department of Energy under Contract No. DE-AC02-06CH11357. This research is also part of the Delta research computing project, which is supported by the National Science Foundation (award OCI 2005572), and the State of Illinois. Delta is a joint effort of the University of Illinois at Urbana-Champaign and the National Center for Supercomputing Applications. E.A.H. was partially supported by National Science Foundation awards OAC-1931561 and OAC-2209892.

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

## A  DETAILED IMPLEMENTATION OF FEDCOMPASS

Table 4 lists all frequently used notations and their corresponding explanations in the algorithm description of `FedCompass`.

Table 4: Summary of notations used in algorithm description of `FedCompass`.

| Notation | Explanation |
| --- | --- |
| $\mathcal{C}$ | client information dictionary |
| $\mathcal{G}$ | arrival group information dictionary |
| $G$ | index of the arrival group for a client |
| $Q$ | number of local steps for a client |
| $S$ | estimated computing speed for a client |
| $T_{\text{start}}$ | start time of current local training round for a client |
| $CL$ | list of non-arrived clients for an arrival group |
| $ACL$ | list of arrived clients for an arrival group |
| $T_a$ | expected arrival time for an arrival group |
| $T_{\max}$ | latest arrival time for an arrival group |
| $T_{\text{now}}$ | current time |
| $Q_{\max}$ | maximum number of local steps for clients |
| $Q_{\min}$ | minimum number of local steps for clients |
| $\lambda$ | latest time factor |
| $\eta_\ell$ | client learning rate |
| $N$ | total number of communication rounds |
| $B$ | batch size for local training |
| $st(\cdot)$ | staleness function |
| $\tau$ | timestamp of the global model that a client trained on |
| $\tau_g$ | timestamp of the global model |
| $p_i$ | importance weight of client $i$ |
| $\Delta_i$ | the difference between the original model and the trained model of client $i$ |
| $\overline{\Delta}^g$ | client update buffer for group $g$ |
| $\overline{\Delta}$ | general client update buffer |
| $w$ | global model |
| $w_{i,q}$ | local model of client $i$ after $q$ local steps |

`Compass` utilizes the concept of arrival group to determine which clients are supposed to arrive together. To manage the client arrival groups, `Compass` employs two dictionaries: $\mathcal{C}$ for client information and $\mathcal{G}$ for arrival group information. Specifically, for client $i$, $\mathcal{C}[i].G, \mathcal{C}[i].Q, \mathcal{C}[i].S$, and $\mathcal{C}[i].T_{\text{start}}$ represent the arrival group, number of local steps, estimated computing speed (time per step), and start time of current communication round of client $i$. $\mathcal{C}[i].\tau$ represents the timestamp of the global model upon which client $i$ trained (i.e., client local model timestamp). For arrival group $g$, $\mathcal{G}[g].CL, \mathcal{G}[g].ACL, \mathcal{G}[g].T_a$, and $\mathcal{G}[g].T_{\max}$ represent the list of non-arrived and arrived clients, expected arrival time for clients in the group, and latest arrival time for clients in the group. According to the client speed estimation, all clients in group $g$ are expected to finish local training and send the trained local updates back to the server before $\mathcal{G}[g].T_a$, however, to account for potential errors in speed estimation and variations in computing speed, the group can wait until the latest arrival time $\mathcal{G}[g].T_{\max}$ for the clients to send their trained models back. The difference between $\mathcal{G}[g].T_{\max}$ and the group creation time is $\lambda$ times longer than the difference between $\mathcal{G}[g].T_a$ and the group creation time.

Algorithm 2 presents the process through which `Compass` assigns a client to an arrival group. The dictionaries $\mathcal{C}$ and $\mathcal{G}$ along with the constant parameters $Q_{\max}$, $Q_{\min}$, and $\lambda$ are assumed to be available implicitly and are not explicitly passed as algorithm inputs. When `Compass` assigns a client to a group, it first checks whether it can join an existing arrival group (function JOINGROUP, lines 5 to 16). For each group, `Compass` uses the current time, group expected arrival time, and estimated client speed to calculate a tentative local step number $q$ (line 8). To avoid large disparity in the number of local steps, `Compass` uses two parameters $Q_{\max}$ and $Q_{\min}$ to specify a valid range of local steps. If `Compass` can find the largest $q$ within the range $[Q_{\min}, Q_{\max}]$, it assigns the client to the corresponding group. Otherwise, `Compass` creates a new arrival group for the client. When

---

**Algorithm 2:** ASSIGNGROUP

---

1 **Function** ASSIGNGROUP($i$):  **Input:** Client id $i$
2 **if** JOINGROUP($i$) = $False$ **then**
3   $\quad$ CREATEGROUP($i$);

4
5 **Function** JOINGROUP($i$):  **Input:** Client id $i$
6 Initialization: $G_{\text{assign}}, Q_{\text{assign}} := -1, -1$;
7 **for** $g \in \mathcal{G}$ **do**
8   $\quad q := \lfloor (\mathcal{G}[g].T_a - T_{\text{now}})/\mathcal{C}[i].S \rfloor$;
9   $\quad$ **if** $q \geq Q_{\min}$ $and$ $q \geq Q_{assign}$ $and$ $q \leq Q_{\max}$ **then**
10   $\quad\quad G_{\text{assign}}, Q_{\text{assign}} := g, q$;
11 **if** $G_{assign} \neq -1$ **then**
12   $\quad \mathcal{C}[i].(G, Q, T_{\text{start}}) := (G_{\text{assign}}, Q_{\text{assign}}, T_{\text{now}})$;
13   $\quad \mathcal{G}[g].CL.\text{add}(i)$;
14   $\quad$ **return** True;
15 **else**
16   $\quad$ **return** False

17
18 **Function** CREATEGROUP($i$):  **Input:** Client id $i$
19 Initialization: $Q_{\text{assign}} := -1$;
20 **for** $g \in \mathcal{G}$ **do**
21   $\quad$ **if** $T_{now} < \mathcal{G}[g].T_a$ **then**
22   $\quad\quad S :=$ speed of the fastest client in group $g$;
23   $\quad\quad T_{\text{exp}} := \mathcal{G}[g].T_a + S * Q_{\max}$;
24   $\quad\quad Q_{\text{assign}} := \max(Q_{\text{assign}}, \lfloor (T_{\text{exp}} - T_{\text{now}})/\mathcal{C}[i].S \rfloor)$;
25 **if** $Q_{assign} \geq 0$ $and$ $Q_{assign} < Q_{\min}$ **then**
26   $\quad Q_{\text{assign}} := Q_{\min}$;
27 **else**
28   $\quad$ **if** $Q_{assign} < 0$ $or$ $Q_{assign} > Q_{\max}$ **then**
29   $\quad\quad Q_{\text{assign}} := Q_{\max}$;
30 $G_{\text{new}} :=$ a unique id for the new group;
31 $\mathcal{G}[G_{\text{new}}].(CL, ACL) := ([i], [])$;
32 $\mathcal{G}[G_{\text{new}}].T_a := T_{\text{now}} + Q_{\text{assign}} * \mathcal{C}[i].S$;
33 $\mathcal{G}[G_{\text{new}}].T_{\max} := T_{\text{now}} + Q_{\text{assign}} * \mathcal{C}[i].S * \lambda$;
34 $\mathcal{C}[i].(G, Q, T_{\text{start}}) := (G_{\text{new}}, Q_{\text{assign}}, T_{\text{now}})$;
35 Add a timer event at $\mathcal{G}[G_{\text{new}}].T_{\max}$ to invoke GROUPAGGREGATE($G_{\text{new}}$);

---

**Algorithm 3:** GROUPAGGREGATE Algorithm

---

1 **Function** GROUPAGGREGATE($g$):
  **Input:** group index $g$
2 **if** $g \in \mathcal{G}$ **then**
3   $\quad w := w - \overline{\Delta}_g - \overline{\Delta}$;  $\overline{\Delta} := \mathbf{0}$;  $\tau_g := \tau_g + 1$;
4   $\quad$ sort $\mathcal{G}[g].ACL$ from fastest client to slowest client;
5   $\quad$ **for** $i \in \mathcal{G}[g].ACL$ **do**
6   $\quad\quad \mathcal{C}[i].\tau := \tau_g$;
7   $\quad\quad$ ASSIGNGROUP($i$);
8   $\quad\quad$ FEDCOMPASS-CLIENT($i, w, \mathcal{C}[i].Q, \eta_\ell, B$);

---

creating this new group, Compass aims to ensure that a specific existing arrival group can join the newly created group for group merging once all of its clients have arrived, as described in function CREATEGROUP (lines 18 to 35). To achieve this, Compass first identifies the computing speed $S$ of the fastest client in group $g$. It assumes that all clients in the group will arrive at the expected time $\mathcal{G}[g].T_a$ and the fastest client will perform $Q_{\max}$ local steps in the next round. Subsequently,

`Compass` utilizes the expected arrival time in the next round (line 23) to determine the appropriate number of local steps (line 24). It then selects the largest number of local steps while ensuring that it is within the range $[Q_{\min}, Q_{\max}]$. In cases where `Compass` is unable to set the number of local steps in a way that allows any existing group to be suitable for joining in the next round, it assigns $Q_{\max}$ to the client upon creating the group. Whenever a new group is created, `Compass` sets up a timer event to invoke the GROUPAGGREGATE function (Algorithm 3) to aggregate the group of client updates after passing the group's latest arrival time $\mathcal{G}\,[\,g\,]\,.T_{\max}$. In GROUPAGGREGATE, the server updates the global model using the group-specific buffer $\overline{\Delta}^g$, containing updates from clients in arrival group $g$, and the general buffer $\overline{\Delta}$, containing updates from clients arriving late in other groups. After the global update, the server updates the global model timestamp and assigns new training tasks to all arriving clients in group $g$.

## B   UPPER BOUND ON THE NUMBER OF GROUPS

If the client speeds do not have large variations for a sufficient number of global updates, then `FedCompass` can reach an equilibrium state where the arrival group assignment for clients is fixed and the total number of existing groups (denoted by $\#_g$) is bounded by $\lceil \log_Q \mu \rceil$, where $Q = Q_{\max}/Q_{\min}$ and $\mu$ is the ratio of the time per local step between the slowest client and the fastest client.

Consider one execution of `FedCompass` with a list of sorted client local computing times $\{\mathcal{T}_1, \mathcal{T}_2, \cdots, \mathcal{T}_m\}$, where $\mathcal{T}_i$ represents the time for client $i$ to complete one local step and $\mathcal{T}_i \leq \mathcal{T}_j$ if $i < j$. Let $\mathcal{G}_1 = \{i \mid \frac{T_i}{T_1} \leq \frac{Q_{\max}}{Q_{\min}}\} = \{1, \cdots, g_1\}$ be the first group of clients. Then all the clients $i \in \mathcal{G}_1$ will be assigned to the same arrival group at the equilibrium state with $Q_i = \frac{T_1}{T_i} Q_{\max} \in [Q_{\min}, Q_{\max}]$. Similarly, if $g_1 < m$, we can get the second group of clients $\mathcal{G}_2 = \{i \mid \frac{T_i}{T_{g_1+1}} \leq \frac{Q_{\max}}{Q_{\min}}\} = \{g_1 + 1, \cdots, g_2\}$ where the clients in $\mathcal{G}_2$ will be assigned to another arrival group at the equilibrium state. Therefore, the maximum number of existing arrival groups at the equilibrium state is equal to the smallest integer such that

$$\left(\frac{Q_{\max}}{Q_{\min}}\right)^{\#_g} = Q^{\#_g} \leq \frac{\mathcal{T}_m}{\mathcal{T}_1} = \mu \implies \#_g \leq \lceil \log_Q \mu \rceil. \tag{3}$$

Additionally, this implies that the maximum ratio of time for two different clients to finish one communication round at the equilibrium state is bounded by $\mu' = Q^{\lfloor \log_Q \mu \rfloor}$. First, we can assume that the maximum ratio of time for two different clients to finish one communication round is the same as that for two different arrival groups to finish one communication round. In the extreme case that there are maximum $\lceil \log_Q \mu \rceil$ arrival groups. Then the time ratio between the slowest group and fastest group is bounded by $Q^{\lceil \log_Q \mu \rceil - 1} = Q^{\lfloor \log_Q \mu \rfloor}$.

## C  FEDCOMPASS CONVERGENCE ANALYSIS

### C.1  LIST OF NOTATIONS

Table 5 lists frequently used notations in the convergence analysis of `FedCompass`.

Table 5: Frequently used notations in the convergence analysis of `FedCompass`.

| Notation | Explanation |
|---|---|
| $m$ | total number of clients |
| $\eta_\ell$ | client local learning rate |
| $L$ | Lipschitz constant |
| $\mu$ | ratio of the time per local step between the slowest client and the fastest client |
| $\sigma_l$ | bound for local gradient variance |
| $\sigma_g$ | bound for global gradient variance |
| $M$ | bound for gradient |
| $Q_{\min}$ | minimum number of client local steps in each training round |
| $Q_{\max}$ | maximum number of client local steps in each training round |
| $Q$ | ratio between $Q_{\max}$ and $Q_{\min}$, i.e., $Q_{\max}/Q_{\min}$ |
| $Q_{i,t}$ | number of local steps for client $i$ in round $t$ |
| $w^{(t)}$ | global model after $t$ global updates |
| $w_{i,q}^{(t-\tau_{i,t})}$ | local model of client $i$ after $q$ local steps, where the original model has timestamp $t - \tau_{i,t}$ |
| $\mathcal{B}_{i,q}$ | a random training batch that client $i$ uses in the $q$-th local step |
| $F(w)$ | global loss for model $w$ |
| $F_i(w)$ | client $i$'s local loss for model $w$ |
| $f_i(w, \mathcal{B}_i)$ | loss for model $w$ evaluated on batch $\mathcal{B}_i$ from client $i$ |
| $\nabla F(w)$ | gradient of global loss w.r.t. $w$ |
| $\nabla F_i(w)$ | gradient of client $i$'s local loss w.r.t. $w$ |
| $\nabla f_i(w, \mathcal{B}_i)$ | gradient of loss evaluated on batch $\mathcal{B}_i$ w.r.t. $w$ |
| $\mathcal{S}_t$ | set of arriving clients at timestamp $t$ |
| $\Delta_i^{(t-\tau_{i,t})}$ | update from client $i$ at timestamp $t$ trained on global model with staleness $\tau_{i,t}$ |
| $\overline{\Delta}^{(t)}$ | aggregated local updates from clients for updating the global model at timestamp $t$ |

### C.2  PROOF OF THEOREM 1

**Lemma 1.** $\mathbb{E}\left[\|\nabla f_i(w, \mathcal{B}_i)\|^2\right] \leq 3(\sigma_l^2 + \sigma_g^2 + M)$ *for any model* $w$.

*Proof of Lemma 1.*

$$
\begin{aligned}
\mathbb{E}\left[\|\nabla f_i(w, \mathcal{B}_i)\|^2\right] &= \mathbb{E}\left[\|\nabla f_i(w, \mathcal{B}_i) - \nabla F_i(w) + \nabla F_i(w) - \nabla F(w) + \nabla F(w)\|^2\right] \\
&\overset{(a)}{\leq} 3\mathbb{E}\left[\|\nabla f_i(w, \mathcal{B}_i) - \nabla F_i(w)\|^2 + \|\nabla F_i(w) - \nabla F(w)\|^2 + \|\nabla F(w)\|^2\right] \\
&\overset{(b)}{\leq} 3(\sigma_l^2 + \sigma_g^2 + M),
\end{aligned}
\tag{4}
$$

where step (a) utilizes *Cauchy-Schwarz inequality*, and step (b) utilizes Assumption 3.

**Lemma 2.** *When a function* $F : \mathbb{R}^n \to \mathbb{R}$ *is Lipschitz smooth, then we have*

$$
F(w') \leq F(w) + \langle \nabla F(w), w' - w \rangle + \frac{L}{2}\|w' - w\|^2.
\tag{5}
$$

*Proof of Lemma 2.*

Let $w_t = w + t(w' - w)$ for $t \in [0, 1]$. Then according to the definition of gradient, we have

$$
\begin{aligned}
F(w') &= F(w) + \int_0^1 \langle \nabla F(w_t), w' - w \rangle dt \\
&= F(w) + \langle \nabla F(w), w' - w \rangle + \int_0^1 \langle \nabla F(w_t) - \nabla F(w), w' - w \rangle dt \\
&\stackrel{(a)}{\leq} F(w) + \langle \nabla F(w), w' - w \rangle + \int_0^1 \|\nabla F(w_t) - \nabla F(w)\| \cdot \|w' - w\| dt \\
&\stackrel{(b)}{\leq} F(w) + \langle \nabla F(w), w' - w \rangle + L \int_0^1 \|w_t - w\| \cdot \|w' - w\| dt \\
&= F(w) + \langle \nabla F(w), w' - w \rangle + L\|w' - w\|^2 \int_0^1 t \, dt \\
&= F(w) + \langle \nabla F(w), w' - w \rangle + \frac{L}{2}\|w' - w\|^2,
\end{aligned}
\tag{6}
$$

where step (a) utilizes *Cauchy-Schwarz inequality* $\langle u, v \rangle \leq |\langle u, v \rangle| \leq \|u\| \cdot \|v\|$, and step (b) utilizes the definition of *Lipschitz smoothness*.

*Proof of Theorem 1.*

According to Lemma 2, we have

$$
\begin{aligned}
F(w^{(t+1)}) &\leq F(w^{(t)}) + \langle \nabla F(w^{(t)}), w^{(t+1)} - w^{(t)} \rangle + \frac{L}{2}\|w^{(t+1)} - w^{(t)}\|^2 \\
&\leq F(w^{(t)}) - \langle \nabla F(w^{(t)}), \overline{\Delta}^{(t)} \rangle + \frac{L}{2}\|\overline{\Delta}^{(t)}\|^2 \\
&= F(w^{(t)}) \underbrace{- \frac{1}{m} \sum_{k \in \mathcal{S}_t} \langle \nabla F(w^{(t)}), \Delta_k^{(t-\tau_{k,t})} \rangle}_{P_1} + \underbrace{\frac{L}{2m^2}\Big\| \sum_{k \in \mathcal{S}_t} \Delta_k^{(t-\tau_{k,t})} \Big\|^2}_{P_2},
\end{aligned}
\tag{7}
$$

where $\mathcal{S}_t$ represents the set of clients arrived at timestamp $t$, and $\Delta_k^{(t-\tau_{k,t})}$ represents the update from client $k$ trained on the global model with staleness $\tau_{k,t}$. Rearranging part $P_1$,

$$
P_1 = -\frac{1}{m} \sum_{k \in \mathcal{S}_t} \langle \nabla F(w^{(t)}), \Delta_k^{(t-\tau_{k,t})} \rangle = -\frac{\eta_\ell}{m} \sum_{k \in \mathcal{S}_t} \Big\langle \nabla F(w^{(t)}), \sum_{q=0}^{Q_{k,t}-1} \nabla f_k(w_{k,q}^{(t-\tau_{k,t})}, \mathcal{B}_{k,q}) \Big\rangle,
\tag{8}
$$

where $Q_{k,t}$ represents the number of local steps client $k$ has taken before sending the update to the server at timestamp $t$, $w_{k,q}^{(t-\tau_{k,t})}$ represents the model of client $k$ in local step $q$, and $\mathcal{B}_{k,q}$ represents a random training batch that the client $k$ uses in local step $q$. To compute an upper bound for the expectation of part $P_1$, we utilize conditional expectation: $\mathbb{E}[P_1] = \mathbb{E}_{\mathcal{H}} \mathbb{E}_{\{i\}\sim[m]|\mathcal{H}} \mathbb{E}_{\mathcal{B}_i|\{i\}\sim[m],\mathcal{H}}[P_1]$. Specifically, $\mathbb{E}_{\mathcal{H}}$ is the expectation over the history $\mathcal{H}$ of iterates up to timestamp $t$, $\mathbb{E}_{\{i\}\sim[m]|\mathcal{H}}$ is the expectation over the random group of clients arriving at timestamp $t$, and $\mathbb{E}_{\mathcal{B}_i|\{i\}\sim[m],\mathcal{H}}$ is the expectation over the mini-batch stochastic gradient on client $i$.

$$\mathbb{E}[P_1] = -\frac{\eta_\ell}{m}\mathbb{E}\Big[\sum_{k\in\mathcal{S}_t}\Big\langle\nabla F(w^{(t)}), \sum_{q=0}^{Q_{k,t}-1}\nabla f_k(w_{k,q}^{(t-\tau_{k,t})},\mathcal{B}_{k,q})\Big\rangle\Big]$$

$$\overset{(a)}{=} -\frac{\eta_\ell}{m}\mathbb{E}\Big[\sum_{k\in\mathcal{S}_t}\sum_{q=0}^{Q_{k,t}-1}\Big\langle\nabla F(w^{(t)}),\nabla F_k(w_{k,q}^{(t-\tau_{k,t})})\Big\rangle\Big],$$

$$\overset{(b)}{=} -\frac{\eta_\ell}{2m}\mathbb{E}\Big[\sum_{k\in\mathcal{S}_t}\sum_{q=0}^{Q_{k,t}-1}\Big(\|\nabla F(w^{(t)})\|^2 - \|\nabla F(w^{(t)}) - \nabla F_k(w_{k,q}^{(t-\tau_{k,t})})\|^2\Big)\Big]$$

$$-\frac{\eta_\ell}{2m}\mathbb{E}\Big[\sum_{k\in\mathcal{S}_t}\sum_{q=0}^{Q_{k,t}-1}\|\nabla F_k(w_{k,q}^{(t-\tau_{k,t})})\|^2\Big]$$

$$\overset{(c)}{=} -\frac{\eta_\ell}{2m}\mathbb{E}_{\mathcal{H}}\Big[\sum_{i=1}^{m}\pi_i\sum_{q=0}^{Q_{i,t}-1}\Big(\|\nabla F(w^{(t)})\|^2 - \|\nabla F(w^{(t)}) - \nabla F_i(w_{i,q}^{(t-\tau_{i,t})})\|^2\Big)\Big]$$

$$-\frac{\eta_\ell}{2m}\mathbb{E}\Big[\sum_{k\in\mathcal{S}_t}\sum_{q=0}^{Q_{k,t}-1}\|\nabla F_k(w_{k,q}^{(t-\tau_{k,t})})\|^2\Big]$$

$$= -\frac{\eta_\ell\mathcal{P}}{2m}\mathbb{E}_{\mathcal{H}}\Big[\sum_{i=1}^{m}\pi_i'\sum_{q=0}^{Q_{i,t}-1}\|\nabla F(w^{(t)})\|^2\Big]$$

$$+\frac{\eta_\ell\mathcal{P}}{2m}\mathbb{E}_{\mathcal{H}}\Big[\sum_{i=1}^{m}\pi_i'\sum_{q=0}^{Q_{i,t}-1}\|\nabla F(w^{(t)}) - \nabla F_i(w_{i,q}^{(t-\tau_{i,t})})\|^2\Big]$$

$$-\frac{\eta_\ell}{2m}\mathbb{E}\Big[\sum_{k\in\mathcal{S}_t}\sum_{q=0}^{Q_{k,t}-1}\|\nabla F_k(w_{k,q}^{(t-\tau_{k,t})})\|^2\Big]$$

$$\leq -\frac{\eta_\ell}{2m}\mathbb{E}_{\mathcal{H}}\Big[\sum_{i=1}^{m}\pi_i'\sum_{q=0}^{Q_{i,t}-1}\|\nabla F(w^{(t)})\|^2\Big]$$

$$+\frac{\eta_\ell}{2}\mathbb{E}_{\mathcal{H}}\Big[\sum_{i=1}^{m}\pi_i'\sum_{q=0}^{Q_{i,t}-1}\|\nabla F(w^{(t)}) - \nabla F_i(w_{i,q}^{(t-\tau_{i,t})})\|^2\Big]$$

$$-\frac{\eta_\ell}{2m}\mathbb{E}\Big[\sum_{k\in\mathcal{S}_t}\sum_{q=0}^{Q_{k,t}-1}\|\nabla F_k(w_{k,q}^{(t-\tau_{k,t})})\|^2\Big],$$

(9)

where step (a) utilizes Assumption 2, step (b) utilizes the identity $\langle u, v\rangle = \frac{1}{2}(\|u\|^2 + \|v\|^2 - \|u - v\|^2)$, and step (c) utilizes conditional expectation with the probability $\pi_i$ of client $i$ to arrive at timestamp $t$. We let $\mathcal{P} = \sum_{i=1}^{m}\pi_i \in [1, m]$ and $\pi_i' := \pi_i/\mathcal{P}$ such that $\sum_{i=1}^{m}\pi_i' = 1$.

To derive the upper and lower bounds of $\pi_i'$, according to the conclusion we get in Appendix B, we have $\pi_{\max}'/\pi_{\min}' \leq Q^{\lfloor\log_Q\mu\rfloor} = \mu'$, i.e., $\pi_{\min}' \geq \frac{1}{\mu'}\pi_{\max}'$. Therefore, we can derive the upper and lower bounds of $\pi_i'$ as follows.

$$1 = \sum_{i=1}^{m}\pi_i' \geq \pi_{\max}' + (m-1)\pi_{\min}' \geq \pi_{\max}' + \frac{m-1}{\mu'}\pi_{\max}'$$

$$\pi_{\max}' \leq \frac{1}{1 + \frac{m-1}{\mu'}}$$

(10)

$$1 = \sum_{i=1}^{m} \pi_i' \leq \pi_{\min}' + (m-1)\pi_{\max}' \leq \pi_{\min}' + \mu'(m-1)\pi_{\min}'$$

$$\pi_{\min}' \geq \frac{1}{1 + \mu'(m-1)} \tag{11}$$

Let $\gamma_1 = 1 + \frac{m-1}{\mu'}$ and $\gamma_2 = 1 + \mu'(m-1)$. We have $\pi_i' \in [\frac{1}{\gamma_2}, \frac{1}{\gamma_1}]$. Then,

$$
\mathbb{E}[P_1] \leq - \frac{\eta_\ell Q_{\min}}{2\gamma_2}\mathbb{E}\Big[\|\nabla F(w^{(t)})\|^2\Big] - \frac{\eta_\ell}{2m}\mathbb{E}\Big[\sum_{k \in \mathcal{S}_t}\sum_{q=0}^{Q_{k,t}-1}\|\nabla F_k(w_{k,q}^{(t-\tau_{k,t})})\|^2\Big]
$$

$$
+ \underbrace{\frac{\eta_\ell}{2\gamma_1}\mathbb{E}_{\mathcal{H}}\Big[\sum_{i=1}^{m}\sum_{q=0}^{Q_{\max}-1}\|\nabla F(w^{(t)}) - \nabla F_i(w_{i,q}^{(t-\tau_{i,t})})\|^2\Big]}_{P_3}. \tag{12}
$$

For part $P_3$, by utilizing Assumption 1, we can get

$$
P_3 = \frac{\eta_\ell}{2\gamma_1}\mathbb{E}_{\mathcal{H}}\Big[\sum_{i=1}^{m}\sum_{q=0}^{Q_{\max}-1}\|\nabla F(w^{(t)}) - \nabla F_i(w_{i,q}^{(t-\tau_{i,t})})\|^2\Big]
$$

$$
\leq \frac{\eta_\ell}{\gamma_1}\mathbb{E}_{\mathcal{H}}\Big[\sum_{i=1}^{m}\sum_{q=0}^{Q_{\max}-1}\Big(\|\nabla F_i(w^{(t)}) - \nabla F_i(w^{(t-\tau_{i,t})})\|^2 + \|\nabla F_i(w^{(t-\tau_{i,t})}) - \nabla F_i(w_{i,q}^{(t-\tau_{i,t})})\|^2\Big)\Big]
$$

$$
\leq \frac{\eta_\ell L^2}{\gamma_1}\mathbb{E}_{\mathcal{H}}\Big[\sum_{i=1}^{m}\sum_{q=0}^{Q_{\max}-1}\Big(\|w^{(t)} - w^{(t-\tau_{i,t})}\|^2 + \|w^{(t-\tau_{i,t})} - w_{i,q}^{(t-\tau_{i,t})}\|^2\Big)\Big]. \tag{13}
$$

For $\|w^{(t)} - w^{(t-\tau_{i,t})}\|^2$, we derive its upper bound in the following way, with step (a) utilizing *Cauchy-Schwarz inequality* and step (b) utilizing Lemma 1.

$$
\|w^{(t)} - w^{(t-\tau_{i,t})}\|^2 = \Big\|\sum_{\rho=t-\tau_{i,t}}^{t}(w^{(\rho+1)} - w^{(\rho)})\Big\|^2 = \Big\|\sum_{\rho=t-\tau_{i,t}}^{t}\frac{1}{m}\sum_{k \in \mathcal{S}_\rho}\Delta_k^{(\rho-\tau_{k,\rho})}\Big\|^2
$$

$$
= \frac{\eta_\ell^2}{m^2}\Big\|\sum_{\rho=t-\tau_{i,t}}^{t}\sum_{k \in \mathcal{S}_\rho}\sum_{q=0}^{Q_{k,\rho}-1}\nabla f_k(w_{k,q}^{(\rho-\tau_{k,\rho})}, \mathcal{B}_{k,q})\Big\|^2 \tag{14}
$$

$$
\overset{(a)}{\leq} \frac{\eta_\ell^2 \tau_{\max}Q_{\max}}{m}\sum_{\rho=t-\tau_{i,t}}^{t}\sum_{k \in \mathcal{S}_\rho}\sum_{q=0}^{Q_{k,\rho}-1}\Big\|\nabla f_k(w_{k,q}^{(\rho-\tau_{k,\rho})}, \mathcal{B}_{k,q})\Big\|^2
$$

$$
\overset{(b)}{\leq} 3\eta_\ell^2\tau_{\max}^2 Q_{\max}^2(\sigma_l^2 + \sigma_g^2 + M)
$$

For $\|w^{(t-\tau_{i,t})} - w_{i,q}^{(t-\tau_{i,t})}\|^2$, we derive its upper bound also using Lemma 1.

$$
\|w^{(t-\tau_{i,t})} - w_{i,q}^{(t-\tau_{i,t})}\|^2 = \eta_\ell^2\Big\|\sum_{e=0}^{q-1}\nabla f_i(w_{i,e}^{(t-\tau_{i,t})}, \mathcal{B}_{i,e})\Big\|^2 \leq 3\eta_\ell^2 Q_{\max}^2(\sigma_l^2 + \sigma_g^2 + M) \tag{15}
$$

Combining equation 14 and equation 15, we can obtain the upper bound of $P_3$:

$$
P_3 \leq \frac{3mL^2\eta_\ell^3 Q_{\max}^3}{\gamma_1}(\tau_{\max}^2 + 1)(\sigma_l^2 + \sigma_g^2 + M). \tag{16}
$$

Putting the upper bound of $P_3$ (equation 16) into equation 12, we can get:

$$\mathbb{E}[P_1] \le -\frac{\eta_\ell Q_{\min}}{2\gamma_2}\mathbb{E}\Big[\|\nabla F(w^{(t)})\|^2\Big] \underbrace{-\frac{\eta_\ell}{2m}\mathbb{E}\Big[\sum_{k\in\mathcal{S}_t}\sum_{q=0}^{Q_{k,t}-1}\|\nabla F_k(w_{k,q}^{(t-\tau_{k,t})})\|^2\Big]}_{P_4}$$

$$+\frac{3mL^2\eta_\ell^3 Q_{\max}^3}{\gamma_1}(\tau_{\max}^2+1)(\sigma_l^2+\sigma_g^2+M). \tag{17}$$

To compute the upper bound for the expectation of part $P_2$, similar to $P_1$, we also apply conditional expectation: $\mathbb{E}[P_2] = \mathbb{E}_\mathcal{H}\mathbb{E}_{\{i\}\sim[m]|\mathcal{H}}[P_2]$.

$$\mathbb{E}[P_2] = \frac{L}{2m^2}\mathbb{E}\Big[\Big\|\sum_{k\in\mathcal{S}_t}\Delta_k^{(t-\tau_{k,t})}\Big\|^2\Big] = \frac{\eta_\ell^2 L}{2m^2}\mathbb{E}\Big[\Big\|\sum_{k\in\mathcal{S}_t}\sum_{q=0}^{Q_{k,t}-1}\nabla f_k(w_{k,q}^{(t-\tau_{k,t})},\mathcal{B}_{k,q})\Big\|^2\Big]$$

$$= \frac{\eta_\ell^2 L}{2m^2}\mathbb{E}\Big[\Big\|\sum_{k\in\mathcal{S}_t}\sum_{q=0}^{Q_{k,t}-1}\Big(\nabla f_k(w_{k,q}^{(t-\tau_{k,t})},\mathcal{B}_{k,q})-\nabla F_k(w_{k,q}^{(t-\tau_{k,t})})\Big)+\sum_{k\in\mathcal{S}_t}\sum_{q=0}^{Q_{k,t}-1}\nabla F_k(w_{k,q}^{(t-\tau_{k,t})})\Big\|^2\Big]$$

$$\overset{(a)}{\le} \frac{\eta_\ell^2 L}{m^2}\mathbb{E}_\mathcal{H}\Big[\Big\|\sum_{i=1}^{m}\pi_i\sum_{q=0}^{Q_{i,t}-1}\Big(\nabla f_i(w_{i,q}^{(t-\tau_{i,t})},\mathcal{B}_{i,q})-\nabla F_i(w_{i,q}^{(t-\tau_{i,t})})\Big)\Big\|^2\Big]$$

$$+\frac{\eta_\ell^2 L}{m^2}\mathbb{E}\Big[\Big\|\sum_{k\in\mathcal{S}_t}\sum_{q=0}^{Q_{k,t}-1}\nabla F_k(w_{k,q}^{(t-\tau_{k,t})})\Big\|^2\Big]$$

$$= \frac{\eta_\ell^2 L\mathcal{P}^2}{m^2}\mathbb{E}_\mathcal{H}\Big[\Big\|\sum_{i=1}^{m}\pi_i'\sum_{q=0}^{Q_{i,t}-1}\Big(\nabla f_i(w_{i,q}^{(t-\tau_{i,t})},\mathcal{B}_{i,q})-\nabla F_i(w_{i,q}^{(t-\tau_{i,t})})\Big)\Big\|^2\Big]$$

$$+\frac{\eta_\ell^2 L}{m^2}\mathbb{E}\Big[\Big\|\sum_{k\in\mathcal{S}_t}\sum_{q=0}^{Q_{k,t}-1}\nabla F_k(w_{k,q}^{(t-\tau_{k,t})})\Big\|^2\Big]$$

$$\le \frac{\eta_\ell^2 L}{\gamma_1^2}\mathbb{E}_\mathcal{H}\Big[\Big\|\sum_{i=1}^{m}\sum_{q=0}^{Q_{i,t}-1}\Big(\nabla f_i(w_{i,q}^{(t-\tau_{i,t})},\mathcal{B}_{i,q})-\nabla F_i(w_{i,q}^{(t-\tau_{i,t})})\Big)\Big\|^2\Big]$$

$$+\frac{\eta_\ell^2 L}{m^2}\mathbb{E}\Big[\Big\|\sum_{k\in\mathcal{S}_t}\sum_{q=0}^{Q_{k,t}-1}\nabla F_k(w_{k,q}^{(t-\tau_{k,t})})\Big\|^2\Big]$$

$$\overset{(b)}{\le} \frac{m\eta_\ell^2 LQ_{\max}}{\gamma_1^2}\sum_{i=1}^{m}\sum_{q=0}^{Q_{\max}-1}\mathbb{E}\Big[\Big\|\nabla f_i(w_{i,q}^{(t-\tau_{i,t})},\mathcal{B}_{i,q})-\nabla F_i(w_{i,q}^{(t-\tau_{i,t})})\Big\|^2\Big]$$

$$+\frac{\eta_\ell^2 L}{m^2}\mathbb{E}\Big[\Big\|\sum_{k\in\mathcal{S}_t}\sum_{q=0}^{Q_{k,t}-1}\nabla F_k(w_{k,q}^{(t-\tau_{k,t})})\Big\|^2\Big]$$

$$\overset{(c)}{\le} \frac{m^2\eta_\ell^2\sigma_l^2 LQ_{\max}^2}{\gamma_1^2}+\underbrace{\frac{\eta_\ell^2 L}{m^2}\mathbb{E}\Big[\Big\|\sum_{k\in\mathcal{S}_t}\sum_{q=0}^{Q_{k,t}-1}\nabla F_k(w_{k,q}^{(t-\tau_{k,t})})\Big\|^2\Big]}_{P_5},$$

$$\tag{18}$$

where step (a) utilizes conditional expectation and *Cauchy-Schwarz inequality*, step (b) utilizes *Cauchy-Schwarz inequality*, and step (c) utilizes Assumption 3.

To find the upper bound of $\mathbb{E}[P_1+P_2]$, we need to make sure that $P_4+P_5 \le 0$ so that we can eliminate these two parts when evaluating the upper bound. The following step (a) employs *Cauchy-*

*Schwarz inequality.*

$$
\begin{aligned}
P4 + P5 &= -\frac{\eta_\ell}{2m}\mathbb{E}\Big[\sum_{k\in\mathcal{S}_t}\sum_{q=0}^{Q_{k,t}-1}\|\nabla F_k(w_{k,q}^{(t-\tau_{k,t})})\|^2\Big] + \frac{\eta_\ell^2 L}{m^2}\mathbb{E}\Big[\Big\|\sum_{k\in\mathcal{S}_t}\sum_{q=0}^{Q_{k,t}-1}\nabla F_k(w_{k,q}^{(t-\tau_{k,t})})\Big\|^2\Big] \\
&\overset{(a)}{\leq} -\frac{\eta_\ell}{2m}\mathbb{E}\Big[\sum_{k\in\mathcal{S}_t}\sum_{q=0}^{Q_{k,t}-1}\|\nabla F_k(w_{k,q}^{(t-\tau_{k,t})})\|^2\Big] + \frac{\eta_\ell^2 L Q_{\max}}{m}\mathbb{E}\Big[\sum_{k\in\mathcal{S}_t}\sum_{q=0}^{Q_{k,t}-1}\Big\|\nabla F_k(w_{k,q}^{(t-\tau_{k,t})})\Big\|^2\Big] \\
&= \Big(\frac{\eta_\ell^2 L Q_{\max}}{m} - \frac{\eta_\ell}{2m}\Big)\mathbb{E}\Big[\sum_{k\in\mathcal{S}_t}\sum_{q=0}^{Q_{k,t}-1}\|\nabla F_k(w_{k,q}^{(t-\tau_{k,t})})\|^2\Big] \leq 0
\end{aligned}
\tag{19}
$$

Therefore, we need to have $\eta_\ell \leq \frac{1}{2LQ_{\max}}$. In such cases, combining equation 17 and equation 18, we have

$$
\begin{aligned}
\mathbb{E}[F(w^{(t+1)})] \leq{}& \mathbb{E}[F(w^{(t)})] - \frac{\eta_\ell Q_{\min}}{2\gamma_2}\mathbb{E}\Big[\|\nabla F(w^{(t)})\|^2\Big] + \frac{m^2\eta_\ell^2\sigma_l^2 L Q_{\max}^2}{\gamma_1^2} \\
&+ \frac{3mL^2\eta_\ell^3 Q_{\max}^3}{\gamma_1}(\tau_{\max}^2+1)(\sigma_l^2+\sigma_g^2+M).
\end{aligned}
\tag{20}
$$

Summing up $t$ from 0 to $T-1$, we can get

$$
\begin{aligned}
\frac{\eta_\ell Q_{\min}}{2\gamma_2}\sum_{t=0}^{T-1}\mathbb{E}\Big[\|\nabla F(w^{(t)})\|^2\Big] \leq{}& \sum_{t=0}^{T-1}\Big(\mathbb{E}[F(w^{(t)})] - \mathbb{E}[F(w^{(t+1)})]\Big) + \frac{m^2\eta_\ell^2\sigma_l^2 L Q_{\max}^2 T}{\gamma_1^2} \\
&+ \frac{3mL^2\eta_\ell^3 Q_{\max}^3 T}{\gamma_1}(\tau_{\max}^2+1)(\sigma_l^2+\sigma_g^2+M) \\
\leq{}& \Big(F(w^{(0)}) - F(w^*)\Big) + \frac{m^2\eta_\ell^2\sigma_l^2 L Q_{\max}^2 T}{\gamma_1^2} \\
&+ \frac{3mL^2\eta_\ell^3 Q_{\max}^3 T}{\gamma_1}(\tau_{\max}^2+1)(\sigma_l^2+\sigma_g^2+M).
\end{aligned}
\tag{21}
$$

Therefore, we obtain the conclusion in Theorem 1.

$$
\begin{aligned}
\frac{1}{T}\sum_{t=0}^{T-1}\mathbb{E}\Big[\|\nabla F(w^{(t)})\|^2\Big] \leq{}& \frac{2\gamma_2\Big(F(w^{(0)}) - F(w^*)\Big)}{\eta_\ell Q_{\min}T} + \frac{2\gamma_2\eta_\ell m^2\sigma_l^2 L Q_{\max}^2}{\gamma_1^2 Q_{\min}} \\
&+ \frac{6\gamma_2 m\eta_\ell^2 L^2 Q_{\max}^3}{\gamma_1 Q_{\min}}(\tau_{\max}^2+1)(\sigma_l^2+\sigma_g^2+M)
\end{aligned}
\tag{22}
$$

*Proof of Corollary 1.*

When $Q_{\min} \leq Q_{\max}/\mu$, then $\mu \leq Q_{\max}/Q_{\min} = Q$. Therefore,
$$
\mu' = Q^{\lfloor\log_\varrho\mu\rfloor} = Q^0 = 1.
\tag{23}
$$
Consequently, $\gamma_1 = 1 + \frac{m-1}{\mu'} = m$, $\gamma_2 = 1 + \mu'(m-1) = m$. Letting $F^* = F(w^{(0)}) - F(w^*)$ and $\sigma^2 = \sigma_l^2 + \sigma_g^2 + M$, equation 22 can be written as

$$
\frac{1}{T}\sum_{t=0}^{T-1}\mathbb{E}\Big[\|\nabla F(w^{(t)})\|^2\Big] \leq \frac{2mF^*}{\eta_\ell Q_{\min}T} + \frac{2\eta_\ell m\sigma_l^2 L Q_{\max}^2}{Q_{\min}} + \frac{6m\eta_\ell^2\sigma^2 L^2 Q_{\max}^3}{Q_{\min}}(\tau_{\max}^2+1).
\tag{24}
$$

If choosing $\eta_\ell = \mathcal{O}(1/Q_{\max}\sqrt{T})$ and considering $Q_{\max}$ and $Q_{\min}$ having the same order in the $\mathcal{O}$-class estimation, we can derive the conclusion of Corollary 1.

$$
\min_{t\in[T]}\mathbb{E}\|\nabla F(w^{(t)})\|^2 \leq \frac{1}{T}\sum_{t=0}^{T-1}\mathbb{E}\Big[\|\nabla F(w^{(t)})\|^2\Big] \leq \mathcal{O}\Big(\frac{mF^*}{\sqrt{T}}\Big) + \mathcal{O}\Big(\frac{m\sigma_l^2 L}{\sqrt{T}}\Big) + \mathcal{O}\Big(\frac{m\tau_{\max}^2\sigma^2 L^2}{T}\Big)
\tag{25}
$$

# D  DETAILS OF THE EXPERIMENT DATASETS

## D.1  CLASSIC DATASET PARTITION STRATEGY

To simulate data heterogeneity on the classic datasets, we employ two partition strategies: (i) *class partition* and (ii) *dual Dirichlet partition*, and apply them to both the MNIST (LeCun, 1998) and CIFAR-10 (Krizhevsky et al., 2009) datasets. These two datasets are the two most commonly used datasets in evaluating FL algorithms (MNIST is used in 32% cases and CIFAR-10 is used in 28% cases) (Ma et al., 2022). The details of these two partition strategies are elaborated in the following subsections.

### D.1.1  CLASS PARTITION

---

**Algorithm 4:** CLASSPARTITION

---

**Data:** Classic dataset $\mathcal{D}$, sample indices for different classes $class\_indices$, number of clients $m$, total number of distinct classes $n$, minimum and maximum number of sample classes for each client split $n_{\min}$ and $n_{\max}$, mean value of the normal distribution $\mu$, standard deviation of the normal distribution $\sigma$.

1  $client\_datasets := \{i : [\,] \textbf{ for } i \in [1, m]\}$ ▷ Return value: local datasets for each client;

2  **repeat**

3       $class\_partition := \{\}$ ▷ For a certain class, which client(s) have samples of that class in its own client split;

4       **for** $i \in [1, m]$ **do**

5           $n_i := \texttt{randint}(n_{\min}, n_{\max})$;

6           $classes_i := \texttt{randpermutation}(n)\,[:n_i]$;

7           **for** $c \in classes_i$ **do**

8               $class\_partition[c] := [i]$ **if** $c \notin class\_partition$ **else** $class\_partition\,[c]\,.add(i)$;

9  **until** $\texttt{len}(class\_partition) = n$;

10  **for** $c \in class\_partition$ **do**

11       $n_s := \texttt{normal}(\texttt{mean}{=}\mu, \texttt{std}{=}\sigma, \texttt{size}{=}\texttt{len}(class\_partition\,[c]))$;

12       $n_s := (n_s/\texttt{sum}(n_s)) * \texttt{len}(class\_indices\,[c])$ ▷ Number of samples with class $c$ for each assigned client;

13       **for** $num, i \in \texttt{zip}(n_s, class\_partition\,[c])$ **do**

14           Add $num$ samples of $\mathcal{D}$ with distinct indices from $classes\_indices\,[c]$ to $client\_datasets\,[i]$;

15  **return** $client\_datasets$;

---

Algorithm 4 outlines the process of *class partition*, which partitions a classic dataset into several client splits such that each client split has samples from only a few classes. Many previous research works partition the classical dataset in a similar way to artificially create non-IID federated datasets (McMahan et al., 2017; Chen et al., 2019). To partition a dataset $\mathcal{D}$, we first obtain a dictionary $class\_indices$ that contains the indices of samples belonging to each class. Namely, $class\_indices\,[c]$ represents the indices of all samples in $\mathcal{D}$ with class $c$. Lines 2 to 9 of the algorithm assign each client $i$ with $n_i$ classes for its own client split, where $n_i \in [n_{\min}, n_{\max}]$. Additionally, the algorithm ensures that each sample class is assigned to at least one client split. Subsequently, from lines 10 to 14, the algorithm determines the number of samples of class $c$ that each assigned client split should have based on a normal distribution, and then assigns the corresponding number of data samples to the respective clients. The *class partition* strategy results in imbalanced and non-IID FL datasets, with clients having varying sample classes within their local datasets.

In the experiments, the mean value of the normal distribution $\mu = 10$, and the standard deviation $\sigma = 3$. Both the MNIST and the CIFAR-10 datasets have 10 distinct classes, namely, $n = 10$. When the number of clients $m = 5$, $n_{\min} = 5$ and $n_{\max} = 6$, when $m = 10$ or 20, $n_{\min} = 3$ and $n_{\max} = 5$. Figure 3 shows a sample class distribution generated by the *class partition* strategy among $m = 10$ clients with $n_{\min} = 3$, $n_{\max} = 5$, and $n = 10$.

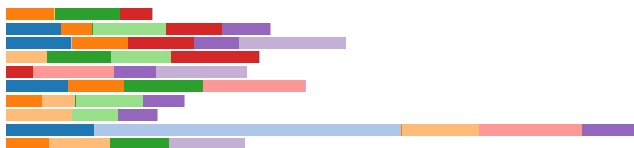

Figure 3: Sample class distribution generated by the *class partition* strategy among ten clients.

### D.1.2 DUAL DIRICHLET PARTITION

---

**Algorithm 5:** DUALDIRICHLETPARTITION

---

**Data:** Classic dataset $\mathcal{D}$, sample indices for different classes $class\_indices$, number of clients $m$, total number of distinct classes $n$, two concentration parameters $\alpha_1$ and $\alpha_2$ for Dirichlet distributions.

1  $client\_datasets := \{i : []$ **for** $i \in [1, m]\}$  ▷ Return value: local datasets for each client;
2  $\mathbf{p_1} := [1/m$ **for** $i \in [1, m]]$  ▷ Prior distribution for the number of samples for each client;
3  $\mathbf{p_2} := [\texttt{len}(class\_indices[c])$ **for** $c \in class\_indices]$;
4  $\mathbf{p_2} := [p/\texttt{sum}(\mathbf{p_2})$ **for** $p \in \mathbf{p_2}]$ ▷ Prior distribution for the number of samples for each class;
5  $\boldsymbol{\alpha_1} := \alpha_1 * \mathbf{p_1}$;
6  $\boldsymbol{\alpha_2} := \alpha_2 * \mathbf{p_2}$;
7  $client\_wts := \texttt{dirichlet}(\boldsymbol{\alpha_1})$  ▷ Normalized number of samples for each client, shape is $\mathbb{R}^m$, sum is 1;
8  $class\_wts := \texttt{dirichlet}(\boldsymbol{\alpha_2}, \texttt{size}=m)$  ▷ Normalized number of samples for each class within each client, shape is $\mathbb{R}^{m \times n}$, row sum is 1;
9  **for** $i \in [1, m]$ **do**
10    **for** $c \in [1, n]$ **do**
11      $num :=$
      $client\_wts[i] * class\_wts[i][c] * \texttt{len}(class\_indices[c]) / \langle client\_wts, class\_wts[:, c] \rangle$;
12      Add $num$ samples of $\mathcal{D}$ with distinct indices from $classes\_indices[c]$ to $client\_datasets[i]$;

13  **return** $client\_datasets$;

---

Some prior works that model data heterogeneity also employ a Dirichlet distribution to model the class distribution within each local client dataset (Yurochkin et al., 2019; Hsu et al., 2019). We extend the work of others using two Dirichlet distributions, with details outlined in Algorithm 5. The first Dirichlet distribution uses concentration parameter $\alpha_1$ and prior distribution $\mathbf{p_1}$ to generate the normalized number of samples for each client, $client\_wts \in \mathbb{R}^m$. The second Dirichlet distribution uses concentration parameter $\alpha_2$ and prior distribution $\mathbf{p_2}$ to generate the normalized number of samples for each class within each client, $class\_wts \in \mathbb{R}^{m \times n}$. From lines 9 to 12, the algorithm combines $client\_wts$, $class\_wts$, and the total number of samples for class $c$ to calculate the number of samples with class $c$ to be assigned for each client, and assigns the samples accordingly. In summary, the *dual Dirichlet partition* strategy employs two Dirichlet distributions to model two levels of data heterogeneity: the distribution across clients and the distribution across different classes within each client. Therefore, the generated client datasets are imbalanced and non-IID, which more accurately reflect real-world federated datasets.

In the experiments, we set $\alpha_1 = m$ and $\alpha_2 = 0.5$. The concentration parameter $\alpha$ governs the similarity among clients or classes. When $\alpha \to \infty$, the distribution becomes identical to the prior distribution. Conversely, when $\alpha \to 0$, only one item is selected with a value of 1 while the remaining items have a value of 0. Since concentration parameter $\alpha_2$ has a very small value, each client has data samples from only a few classes. Figure 4 illustrates a sample class distribution generated by the *dual Dirichlet partition* strategy among $m = 10$ clients, where $n = 10$, $\alpha_1 = 10$, and $\alpha_2 = 0.5$.

### D.2 NATURALLY SPLIT DATASETS: FLAMBY

FLamby is an open-source suite that provides naturally split datasets specifically designed for benchmarking cross-silo federated learning algorithms (Ogier du Terrail et al., 2022). These datasets are sourced directly from distributed real-world entities, ensuring that they preserve the

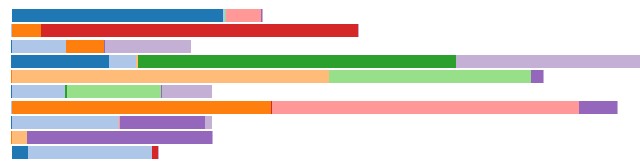

Figure 4: Sample class distribution generated by the *dual Dirichlet partition* strategy among ten clients.

authentic characteristics and complexities of data distribution in FL. In our study, we selected two medium-sized datasets from FLamby – Fed-IXI, and Fed-ISIC2019 – to evaluate the performance of `FedCompass` and other FL algorithms in the experiments. The two datasets offer valuable perspectives into two healthcare domains, brain MRI interpretation (Fed-IXI), and skin cancer detection (Fed-ISIC2019). By leveraging these datasets, our goal is to showcase the robustness and versatility of `FedCompass` across diverse machine learning tasks in real-world scenarios. Table 6 provides the overview of the two selected datasets in FLamby. Detailed descriptions of these datasets are provided in the following subsections.

Table 6: Overview of the two selected datasets in FLamby.

| Dataset | **Fed-IXI** | **Fed-ISIC2019** |
|---|---|---|
| Inputs | T1 weighted image (T1WI) | Dermoscopy |
| Task type | 3D segmentation | Classification |
| Prediction | Brain mask | Melanoma class |
| Client number | 3 | 6 |
| Model | 3D U-net(Çiçek et al., 2016) | EfficientNet(Tan & Le, 2019) |
| Metric | DICE (Dice, 1945) | Balanced accuracy |
| Input dimension | $48 \times 60 \times 48$ | $200 \times 200 \times 3$ |
| Dataset size | 444MB | 9GB |

### D.2.1 FED-IXI

The Fed-IXI dataset is sourced from the Information eXtraction from Images (IXI) database and includes brain T1 magnetic resonance images (MRIs) from three hospitals (Pérez-García et al., 2021). These MRIs have been repurposed for a brain segmentation task; the model performance is evaluated with the DICE score (Dice, 1945). To ensure consistency, the images undergo preprocessing steps, such as volume resizing to $48 \times 60 \times 48$ voxels, and sample-wise intensity normalization. The 3D U-net (Çiçek et al., 2016) serves as the baseline model for this task.

### D.2.2 FED-ISIC2019

The Fed-ISIC2019 dataset is sourced from the ISIC2019 dataset, which includes dermoscopy images collected from four hospitals (Tschandl et al., 2018; Codella et al., 2018; Combalia et al., 2019). The images are restricted to a subset from the public train set and are split based on the imaging acquisition system, resulting in a 6-client federated dataset. The task involves image classification among eight different melanoma classes, and the performance is measured through balanced accuracy. Images are preprocessed by resizing and normalizing brightness and contrast. The baseline classification model is an EfficientNet (Tan & Le, 2019) pretrained on ImageNet (Deng et al., 2009).

## E EXPERIMENT SETUP DETAILS

All the experiments are implemented using the open-source software framework for privacy-preserving federated learning (Ryu et al., 2022; Li et al., 2023).

### E.1 DETAILS OF CLIENT HETEROGENEITY SIMULATION

When simulating the client heterogeneity, we consider the average time $t_i$ required for client $i$ to complete one local step. We assume that $t_i$ follows a random distribution $P$, namely, $t_i \sim P(t)$. In

our experiments, we employ three different distributions to simulate client heterogeneity: the normal distribution $t_i \sim \mathcal{N}(\mu, \sigma^2)$ with $\sigma = 0$ (i.e., homogeneous clients), the normal distribution $t_i \sim \mathcal{N}(\mu, \sigma^2)$ with $\sigma = 0.3\mu$, and the exponential distribution $t_i \sim Exp(\lambda)$. The exponential distribution models the amount of time until a certain event occurs (arrival time), specifically events that follow a Poisson process. The mean values of the distribution ($\mu$ or $1/\lambda$) are adjusted for different datasets according to the size of the model and training data. Table 7 presents the mean values of the client training time distribution for different datasets used in the experiments. Furthermore, Figure 5 illustrates the distribution of the training time to complete one local step among 50 clients under normal distribution with $\sigma = 0.3\mu$ and exponential distribution, both with a mean value of 0.15.

Table 7: Mean values of the training time distribution for different datasets.

| Dataset Name | Batch Size | Mean ($\mu$ or $1/\lambda$) |
|---|---|---|
| MNIST | 64 | 0.15s |
| CIFAR-10 | 128 | 0.5s |
| Fed-IXI | 2 | 0.8s |
| Fed-ISIC2019 | 64 | 1.5s |

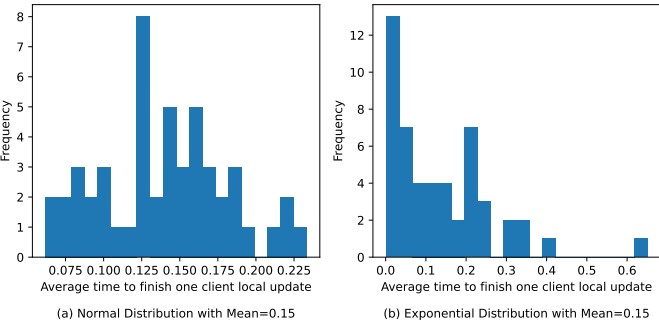

(a) Normal Distribution with Mean=0.15

(b) Exponential Distribution with Mean=0.15

Figure 5: Sample distribution of the training time to complete one local step among 50 clients under (a) normal distribution with $\sigma = 0.3\mu$ and (b) exponential distribution.

### E.2 DETAILED SETUPS AND HYPERPARAMETERS FOR EXPERIMENTS ON THE MNIST DATASET

Table 8 shows the detailed architecture of the convolutional neural network (CNN) used for the experiments on the MNIST dataset. Table 9 provides a comprehensive list of the hyperparameter values used for training. Specifically, $Q$ is the number of client local steps for each training round, $Q_{\min}$ and $Q_{\max}$ are the minimum and maximum numbers of client local steps in `FedCompass`, $\eta_\ell$ is the client local learning rate, $\beta$ is the server-side momentum, and $B$ is the batch size. For all asynchronous FL algorithms, we adopt the polynomial staleness function introduced in (Xie et al., 2019), denoted as $st(t - \tau) = \alpha(t - \tau + 1)^{-a}$. The values of $\alpha$ and $a$ are also listed in Table 9 for all asynchronous algorithms. Additionally, $K$ is the buffer size for the `FedBuff` algorithm, $\upsilon$ is the speed ratio factor for creating tiers for clients (clients with speed ratio within $\upsilon$ are tiered together and its value is selected via grid search), and $\lambda$ is the latest time factor for `FedCompass`. For hyperparameters given as length-three tuples, the tuple items correspond to the hyperparameter values when the number of clients is 5, 10, and 20, respectively. For hyperparameters that are not applicable to certain algorithms, the corresponding value is marked as a dash "-". To ensure a fair comparison, similar algorithms are assigned the same hyperparameter values, as depicted in the table. The number of global training rounds is chosen separately for different experiment settings and FL algorithms such that the corresponding FL algorithm converges, synchronous FL algorithms take the same amount of wall-clock time, and asynchronous algorithms take the same amount of wall-clock time in the same setting. It is also worth mentioning that the buffer size $K$ for the `FedBuff` algorithm (Nguyen et al., 2022) is selected from a search of multiple possible values. Though the original paper states that $K = 10$ is a good setting across all their benchmarks and does not require tuning, the value does not apply in the cross-silo cases where there are less than or equal

to ten clients in most experiment settings, while all the experiments in the original paper have more than thousands of clients.

Table 8: MNIST CNN model architecture.

| Layer | Outsize | Setting | Parama # |
|---|---|---|---|
| Input | (1, 28, 28) | - | 0 |
| Conv1 | (32, 24, 24) | kernel=5, stride=1 | 832 |
| ReLU1 | (32, 24, 24) | - | 0 |
| MaxPool1 | (32, 12, 12) | kernel=2, stride=2 | 0 |
| Conv2 | (64, 8, 8) | kernel=5, stride=1 | 51,264 |
| ReLU2 | (64, 8, 8) | - | 0 |
| MaxPool2 | (64, 4, 4) | kernel=2, stride=2 | 0 |
| Flatten | (1024,) | - | 0 |
| FC1 | (512,) | - | 524,800 |
| ReLU2 | (512,) | - | 0 |
| FC2 | (10,) | - | 5,130 |

Table 9: Hyperparameters for various FL algorithms on the MNIST dataset.

|  | FedAvg | FedAvgM | FedAsync | FedBuff | FedAT | FedCompass(+N) | FedCompass+M |
|---|---|---|---|---|---|---|---|
| $Q$ | (200, 200, 100) | (200, 200, 100) | (200, 200, 100) | (200, 200, 100) | (200, 200, 100) | - | - |
| $Q_{\min}$ | - | - | - | - | - | (40, 40, 20) | (40, 40, 20) |
| $Q_{\max}$ | - | - | - | - | - | (200, 200, 100) | (200, 200, 100) |
| Optim | Adam | Adam | Adam | Adam | Adam | Adam | Adam |
| $\eta_\ell$ | 0.003 | 0.003 | 0.003 | 0.003 | 0.003 | 0.003 | 0.003 |
| $\beta$ | - | 0.9 | - | - | - | - | 0.9 |
| $B$ | 64 | 64 | 64 | 64 | 64 | 64 | 64 |
| $\alpha$ | - | - | 0.9 | 0.9 | - | 0.9 | 0.9 |
| $a$ | - | - | 0.5 | 0.5 | - | 0.5 | 0.5 |
| $K$ | - | - | - | (3, 3, 5) | - | - | - |
| $\upsilon$ | - | - | - | - | 2.0 | - | - |
| $\lambda$ | - | - | - | - | - | 1.2 | 1.2 |

### E.3 DETAILED SETUPS AND HYPERPARAMETERS FOR EXPERIMENTS ON THE CIFAR-10 DATASET

For experiments on the CIFAR-10 dataset, a randomly initialized ResNet-18 model (He et al., 2016) is used in training. Table 10 shows the detailed architecture of the ResNet-18 model, which contains eight residual blocks. Each residual block contains two convolutional + batch normalization layers, as shown in Table 11, and the corresponding output is added by the input itself if the output channel is equal to the input channel, otherwise, the input goes through a 1x1 convolutional + batch normalization layer before added to the output. An additional ReLU layer is appended at the end of the residual block. Table 12 lists the hyperparameter values used in the training for various FL algorithms. The number of global training rounds is chosen separately for different experiment settings and FL algorithms such that the corresponding FL algorithm converges, synchronous FL algorithms take the same amount of wall-clock time, and asynchronous algorithms take the same amount of wall-clock time in the same setting.

### E.4 DETAILED SETUPS AND HYPERPARAMETERS FOR EXPERIMENTS ON THE FLAMBY DATASETS

For the datasets from FLamby (Ogier du Terrail et al., 2022), we use the default experiments settings provided by FLamby. The details of the experiment settings are shown in Table 13

Table 10: CIFAR-10 ResNet18 model architecture.

| Layer | Outsize | Setting | Param # |
|---|---|---|---|
| Input | (3, 32, 32) | - | 0 |
| Conv1 | (64, 32, 32) | kernel=3, stride=1, padding=1, bias=False | 1,728 |
| BatchNorm1 | (64, 32, 32) | - | 128 |
| ReLU1 | (64, 32, 32) | - | 0 |
| ResidualBlock1-1 | (64, 32, 32) | stride=1 | 73,984 |
| ResidualBlock1-2 | (64, 32, 32) | stride=1 | 73,984 |
| ResidualBlock2-1 | (128, 16, 16) | stride=2 | 230,144 |
| ResidualBlock2-2 | (128, 16, 16) | stride=1 | 295,424 |
| ResidualBlock3-1 | (256, 8, 8) | stride=2 | 919,040 |
| ResidualBlock3-2 | (256, 8, 8) | stride=1 | 1,180,672 |
| ResidualBlock4-1 | (512, 4, 4) | stride=2 | 3,673,088 |
| ResidualBlock4-2 | (512, 4, 4) | stride=1 | 4,720,640 |
| AvgPool | (512,) | kernel=4, stride=4 | 0 |
| Linear | (10,) | - | 5,130 |

Table 11: ResidualBlock in ResNet18 architecture.

| Layer | Outsize | Setting | Param # |
|---|---|---|---|
| Input | (in_planes, H, W) | - | 0 |
| Conv1 | (planes, H/stride, W/stride) | kernel=3, stride=stride, padding=1 | in_planes*planes*9 |
| BatchNorm1 | (planes, H/stride, W/stride) | - | 2*planes |
| ReLU1 | (planes, H/stride, W/stride) | - | 0 |
| Conv2 | (planes, H/stride, W/stride) | kernel=3, stride=1, padding=1 | planes*planes*9 |
| BatchNorm2 | (planes, H/stride, W/stride) | - | 2*planes |
| "Identity" | (planes, H/stride, W/stride) | 1x1 Conv + BatchNorm if in_plane $\neq$ plane | - |
| ReLU2 | (planes, H/stride, W/stride) | $\triangleright$ Apply after adding BatchNorm2 and "Identity" outputs | 0 |

Table 12: Hyperparameters for various FL algorithms on the CIFAR-10 dataset.

| | FedAvg | FedAvgM | FedAsync | FedBuff | FedAT | FedCompass(+N) | FedCompass+M |
|---|---|---|---|---|---|---|---|
| $Q$ | (200, 200, 100) | (200, 200, 100) | (200, 200, 100) | (200, 200, 100) | (200, 200, 100) | - | - |
| $Q_{min}$ | - | - | - | - | - | (40, 40, 20) | (40, 40, 20) |
| $Q_{max}$ | - | - | - | - | - | (200, 200, 100) | (200, 200, 100) |
| Optim | SGD | SGD | SGD | SGD | SGD | SGD | SGD |
| $\eta_\ell$ | 0.1 | 0.1 | 0.1 | 0.1 | 0.1 | 0.1 | 0.1 |
| $\beta$ | - | 0.9 | - | - | - | - | 0.9 |
| $B$ | 128 | 128 | 128 | 128 | 128 | 128 | 128 |
| $\alpha$ | - | - | 0.9 | 0.9 | - | 0.9 | 0.9 |
| $a$ | - | - | 0.5 | 0.5 | - | 0.5 | 0.5 |
| $K$ | - | - | - | (3, 3, 5) | - | - | - |
| $\upsilon$ | - | - | - | - | 2.0 | - | - |
| $\lambda$ | - | - | - | - | - | 1.2 | 1.2 |

Table 13: Detailed settings for experiments on FLamby datasets.

| | Fed-IXI | Fed-ISIC2019 |
|---|---|---|
| Model | 3D U-net (Çiçek et al., 2016) | EfficientNet (Tan & Le, 2019) |
| Metric | DICE | Balanced accuracy |
| $m$ (Client #) | 3 | 6 |
| $Q$ | 50 | 50 |
| $Q_{min}$ | 20 | 20 |
| $Q_{max}$ | 100 | 100 |
| Optimizer | AdamW (Loshchilov & Hutter, 2017) | Adam |
| $\eta_\ell$ | 0.001 | 0.0005 |
| $\beta$ | 0.9 | 0.5 |
| $B$ | 2 | 64 |
| $\alpha$ | 0.9 | 0.9 |
| $a$ | 0.5 | 0.5 |
| $K$ | 2 | 3 |
| $\lambda$ | 1.2 | 1.2 |

# F ADDITIONAL EXPERIMENT RESULTS

## F.1 ADDITIONAL EXPERIMENT RESULTS ON THE MNIST DATASET

Table 14 shows the top validation accuracy and the corresponding standard deviation on the *class partitioned* MNIST dataset for various federated learning algorithms with different numbers of clients and client heterogeneity settings among ten independent experiment runs with different random seeds, and Table 15 shows that on the *dual Dirichlet partitioned* MNIST dataset. As all asynchronous FL algorithms take the same amount of training time in the same experiment setting, the results indicate that `FedCompass` not only converges faster than other asynchronous FL algorithms but also can converge to higher accuracy. Notably, in the homogeneous client settings, `FedCompass` does not behave exactly the same as `FedAvg` as the clients also have local speed variance in each local training round, and this prevents the `Compass` scheduler to assign $Q_{\max}$ to all clients and have exactly the same behavior as `FedAvg`. The global model gets updated more frequently for `FedCompass` and leads to relatively faster convergence.

Figures 6–11 depict the change in validation accuracy during the training process for various FL algorithms in different experiment settings. Those figures illustrate how `FedCompass` can achieve faster convergence in different data and device heterogeneity settings.

Table 14: Average top validation accuracy and standard deviation on the *class partitioned* MNIST dataset for various federated learning algorithms with different numbers of clients and client heterogeneity settings.

| Method | $n = 5$ | | | $n = 10$ | | | $n = 20$ | | |
| --- | --- | --- | --- | --- | --- | --- | --- | --- | --- |
| | homo | normal | exp | homo | normal | exp | homo | normal | exp |
| FedAvg | 92.42±3.96 | 92.42±3.96 | 92.42±3.96 | 92.27±3.88 | 92.27±3.88 | 92.27±3.88 | 97.38±1.59 | 97.38±1.59 | 97.38±1.59 |
| FedAvgM | 96.22±2.45 | 96.22±2.45 | 96.22±2.45 | 97.11±1.45 | 97.11±1.45 | 97.11±1.45 | 98.27±0.18 | 98.27±0.18 | 98.27±0.18 |
| FedAsync | 90.60±6.96 | 92.53±5.57 | 92.44±3.07 | 86.73±6.62 | 92.27±3.82 | 96.25±2.46 | 84.26±7.96 | 90.98±2.48 | 98.01±0.89 |
| FedBuff | 95.77±1.93 | 94.69±3.52 | 95.68±1.78 | 91.93±4.50 | 94.83±3.36 | 96.82±1.91 | 93.04±3.61 | 98.12±0.54 | 98.56±0.22 |
| FedAT | 92.47±3.51 | 92.36±3.59 | 91.40±3.21 | 91.50±3.15 | 91.01±3.43 | 93.03±2.94 | 97.17±1.62 | 97.89±0.94 | 95.74±2.48 |
| FedCompass | 95.06±2.89 | 95.54±2.89 | 96.34±1.89 | 94.33±3.80 | 95.47±3.05 | 97.51±1.66 | 98.49±0.45 | 98.66±0.36 | 98.72±0.47 |
| FedCompass+M | 95.58±2.87 | 95.98±2.43 | 95.93±1.92 | 95.64±2.72 | 96.61±2.12 | 97.69±0.76 | 98.58±0.23 | 97.05±0.91 | 98.51±0.47 |
| FedCompass+N | 94.03±3.82 | 93.07±4.00 | 96.44±1.72 | 94.55±3.86 | 94.80±3.65 | 97.28±1.51 | 98.52±0.42 | 98.79±0.17 | 98.80±0.18 |

Table 15: Average top accuracy and standard deviation on the *dual Dirichlet partitioned* MNIST dataset for various federated learning algorithms with different numbers of clients and client heterogeneity settings.

| Method | $m = 5$ | | | $m = 10$ | | | $m = 20$ | | |
| --- | --- | --- | --- | --- | --- | --- | --- | --- | --- |
| | homo | normal | exp | homo | normal | exp | homo | normal | exp |
| FedAvg | 93.08±4.16 | 93.08±4.16 | 93.08±4.16 | 93.71±4.52 | 93.71±4.52 | 93.71±4.52 | 95.77±2.98 | 95.77±2.98 | 95.77±2.98 |
| FedAvgM | 93.88±3.16 | 93.88±3.16 | 93.88±3.16 | 94.00±3.49 | 94.00±3.49 | 94.00±3.49 | 96.06±2.17 | 96.06±2.17 | 96.06±2.17 |
| FedAsync | 92.11±2.99 | 92.96±3.70 | 93.29±3.08 | 86.31±4.51 | 90.72±4.30 | 94.06±5.91 | 78.49±6.28 | 85.82±5.31 | 95.90±1.58 |
| FedBuff | 93.87±4.30 | 94.11±3.60 | 94.94±2.49 | 92.27±4.60 | 94.29±4.04 | 94.50±3.72 | 87.04±6.26 | 95.27±1.84 | 96.90±0.98 |
| FedAT | 92.30±4.00 | 93.06±3.74 | 92.88±3.16 | 95.40±2.70 | 95.63±2.42 | 93.80±4.94 | 95.26±2.47 | 97.32±0.76 | 94.67±3.01 |
| FedCompass | 94.15±4.26 | 94.36±4.28 | 95.46±3.59 | 95.53±3.21 | 95.41±3.52 | 96.34±2.64 | 96.41±1.96 | 97.62±1.57 | 97.86±0.71 |
| FedCompass+M | 94.74±3.42 | 95.21±3.13 | 96.15±2.03 | 96.01±2.76 | 96.10±2.91 | 95.59±2.81 | 97.03±1.50 | 93.25±3.28 | 97.42±0.96 |
| FedCompass+N | 94.56±4.25 | 92.56±4.96 | 94.57±3.69 | 95.45±3.41 | 95.13±3.02 | 95.01±4.62 | 96.28±2.06 | 97.61±0.60 | 97.92±0.50 |

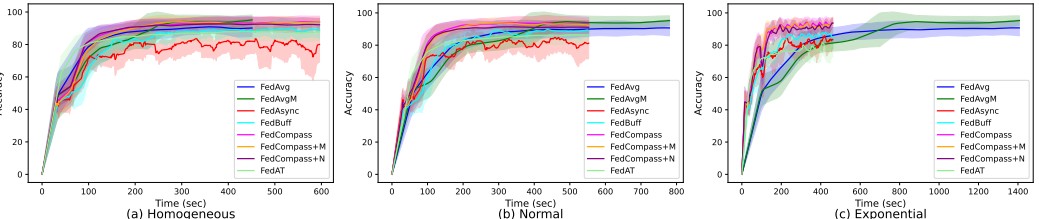

Figure 6: Change in validation accuracy during the training process for different FL algorithms on the *class partitioned* MNIST dataset with five clients and three client heterogeneity settings.

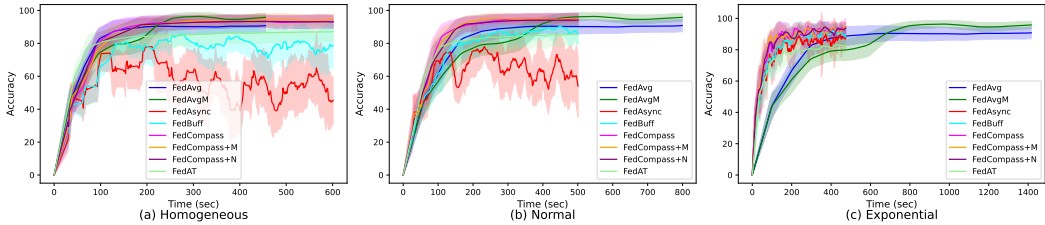

Figure 7: Change in validation accuracy during the training process for different FL algorithms on the *class partitioned* MNIST dataset with ten clients and three client heterogeneity settings.

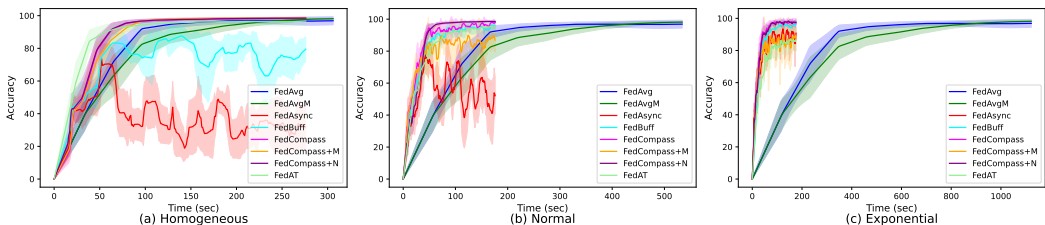

Figure 8: Change in validation accuracy during the training process for different FL algorithms on the *class partitioned* MNIST dataset with twenty clients and three client heterogeneity settings.

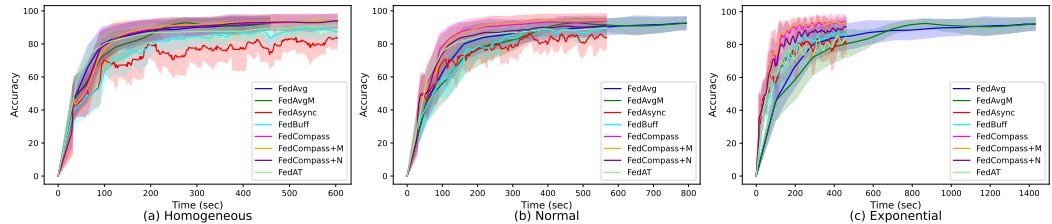

Figure 9: Change in validation accuracy during the training process for different FL algorithms on the *dual Dirichlet partitioned* MNIST dataset with five clients and three client heterogeneity settings.

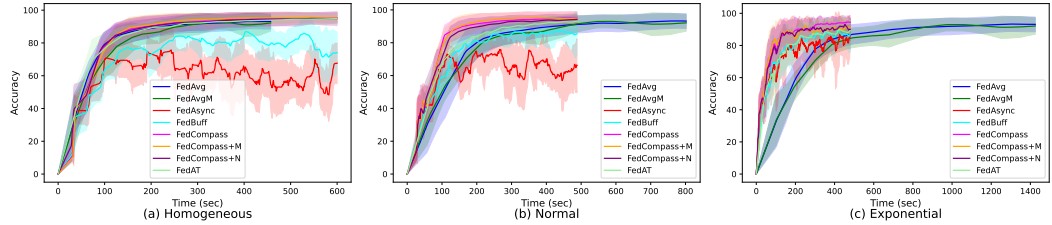

Figure 10: Change in validation accuracy during the training process for different FL algorithms on the *dual Dirichlet partitioned* MNIST dataset with ten clients and three client heterogeneity settings.

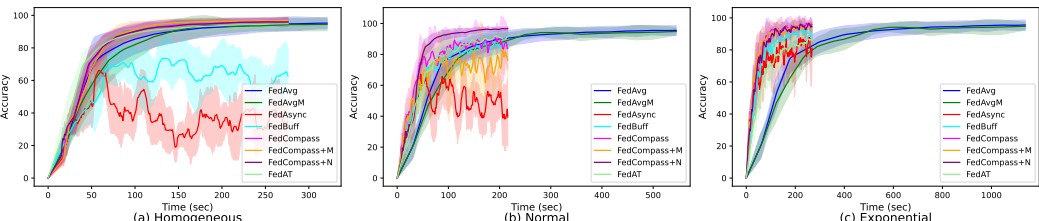

Figure 11: Change in validation accuracy during the training process for different FL algorithms on the *dual Dirichlet partitioned* MNIST dataset with twenty clients and three client heterogeneity settings.

## F.2 ADDITIONAL EXPERIMENT RESULTS ON THE CIFAR-10 DATASET

Table 16 shows the top validation accuracy and the corresponding standard deviation on the *class partitioned* CIFAR-10 dataset for various federated learning algorithms with different numbers of clients and client heterogeneity settings among ten independent experiment runs with different random seeds, and Table 17 shows that on the *dual Dirichlet partitioned* CIFAR-10 dataset. Figures 12–17 give the change in validation accuracy during the training process for various FL algorithms in different experiment settings on the CIFAR-10 datasets.

Table 16: Average top accuracy and standard deviation on the *class partitioned* CIFAR-10 dataset for various FL algorithms with different numbers of clients and client heterogeneity settings.

| | $m = 5$ | | | $m = 10$ | | | $m = 20$ | | |
|---|---|---|---|---|---|---|---|---|---|
| Method | homo | normal | exp | homo | normal | exp | homo | normal | exp |
| FedAvg | 67.10±6.33 | 67.10±6.33 | 67.10±6.3 | 66.80±2.71 | 66.80±2.71 | 66.80±2.71 | 70.03±2.37 | 70.03±2.37 | 70.03±2.37 |
| FedAvgM | 63.44±3.26 | 63.44±3.26 | 63.44±3.26 | 63.50±2.56 | 63.50±2.56 | 63.50±2.56 | 64.47±1.88 | 64.47±1.88 | 64.47±1.88 |
| FedAsync | 53.30±4.12 | 63.65±2.81 | 66.27±3.91 | 37.09±2.99 | 44.87±3.54 | 59.40±4.21 | 30.18±2.84 | 36.73±4.93 | 54.30±2.64 |
| FedBuff | 67.02±2.55 | 67.33±2.46 | 70.86±2.99 | 46.14±4.14 | 50.86±4.43 | 64.32±3.78 | 43.83±3.01 | 49.63±4.60 | 60.90±3.37 |
| FedAT | 72.41±3.31 | 72.92±2.52 | 71.15±3.18 | 65.25±4.21 | 65.70±4.34 | 66.57±4.48 | 66.09±1.72 | 66.38±1.50 | 66.31±1.87 |
| FedCompass | 72.66±2.44 | 73.64±1.80 | 73.96±1.83 | 66.27±2.41 | 67.07±2.40 | 68.73±1.95 | 66.83±1.29 | 67.14±1.32 | 67.41±1.59 |
| FedCompass+M | 72.82±2.27 | 74.03±1.77 | 74.22±1.98 | 66.79±1.93 | 67.72±1.77 | 69.14±1.78 | 66.80±1.18 | 67.48±1.38 | 67.91±1.45 |
| FedCompass+N | 72.79±2.50 | 73.69±2.33 | 74.85±1.91 | 65.53±2.52 | 66.78±2.31 | 68.04±1.99 | 66.32±1.41 | 66.76±1.40 | 67.51±1.50 |

Table 17: Average top accuracy and standard deviation on the *dual Dirichlet partitioned* CIFAR-10 dataset for various FL algorithms with different numbers of clients and client heterogeneity settings.

| | $m = 5$ | | | $m = 10$ | | | $m = 20$ | | |
|---|---|---|---|---|---|---|---|---|---|
| Method | homo | normal | exp | homo | normal | exp | homo | normal | exp |
| FedAvg | 48.88±8.03 | 48.88±8.03 | 48.88±8.03 | 51.91±7.65 | 51.91±7.65 | 51.91±7.65 | 49.28±2.62 | 49.28±2.62 | 49.28±2.62 |
| FedAvgM | 49.89±5.27 | 49.89±5.27 | 49.89±5.27 | 51.72±2.64 | 51.72±2.64 | 51.72±2.64 | 49.94±4.15 | 49.94±4.15 | 49.94±4.15 |
| FedAsync | 44.38±4.71 | 47.67±4.29 | 51.21±4.99 | 33.52±3.29 | 38.13±2.07 | 47.62±2.59 | 34.89±3.34 | 36.12±1.69 | 39.92±3.39 |
| FedBuff | 51.85±3.23 | 54.03±4.84 | 58.11±4.62 | 42.60±2.47 | 45.24±1.12 | 53.04±2.45 | 41.37±2.85 | 42.65±1.70 | 46.30±2.90 |
| FedAT | 58.91±4.63 | 58.34±3.77 | 55.40±4.05 | 52.82±6.37 | 53.54±7.12 | 53.83±5.36 | 48.63±3.02 | 50.56±1.31 | 50.37±2.38 |
| FedCompass | 59.29±3.49 | 59.98±3.65 | 61.23±2.67 | 54.51±3.50 | 54.95±3.68 | 57.29±1.98 | 51.01±2.20 | 52.03±2.75 | 51.39±2.12 |
| FedCompass+M | 59.29±2.65 | 60.20±3.01 | 59.73±3.97 | 54.24±4.21 | 54.53±3.14 | 56.30±2.87 | 51.48±2.54 | 52.73±2.61 | 51.70±2.06 |
| FedCompass+N | 59.38±3.59 | 60.38±2.20 | 61.08±3.74 | 54.28±4.62 | 54.31±6.03 | 55.90±2.97 | 52.13±2.61 | 52.44±1.91 | 52.23±2.20 |

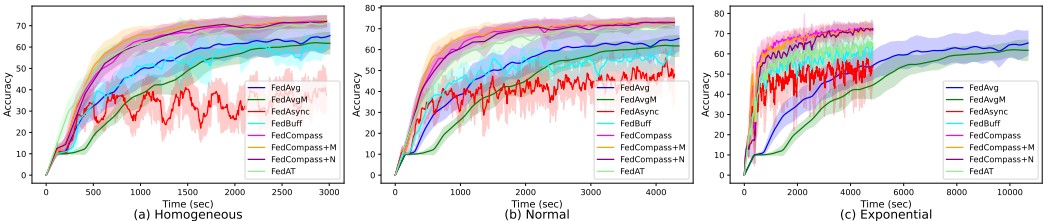

Figure 12: Change in validation accuracy during the training process for different FL algorithms on the *class partitioned* CIFAR-10 dataset with five clients and three client heterogeneity settings.

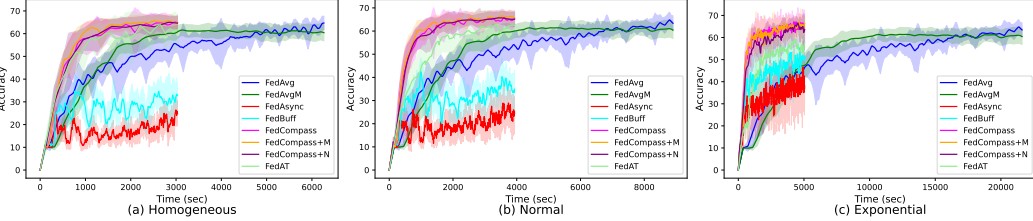

Figure 13: Change in validation accuracy during the training process for different FL algorithms on the *class partitioned* CIFAR-10 dataset with ten clients and three client heterogeneity settings.

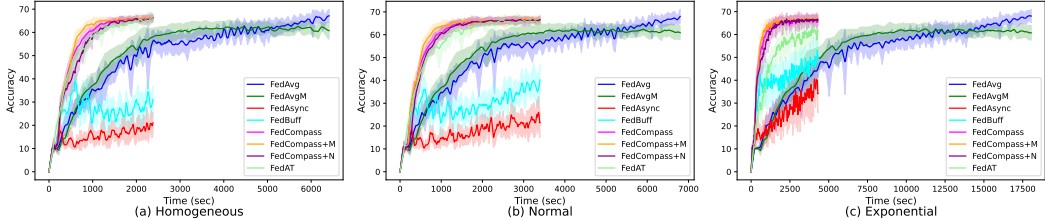

Figure 14: Change in validation accuracy during the training process for different FL algorithms on the *class partitioned* CIFAR-10 dataset with twenty clients and three client heterogeneity settings.

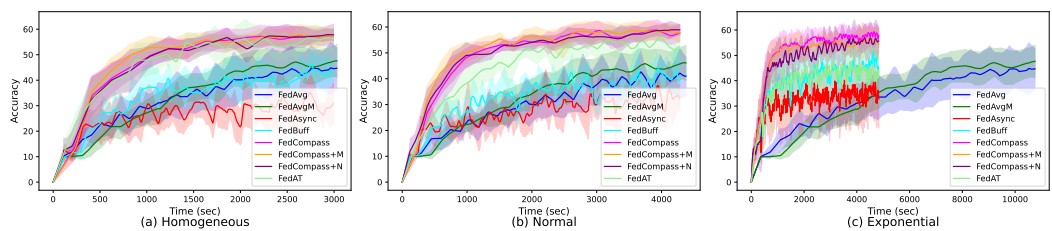

Figure 15: Change in validation accuracy during the training process for different FL algorithms on the *dual Dirichlet partitioned* CIFAR-10 dataset with five clients and three client heterogeneity settings.

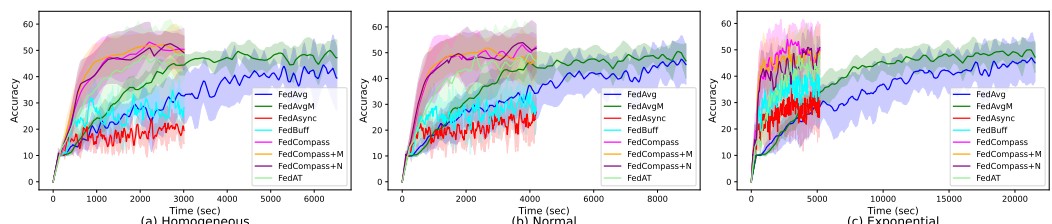

Figure 16: Change in validation accuracy during the training process for different FL algorithms on the *dual Dirichlet partitioned* CIFAR-10 dataset with ten clients and three client heterogeneity settings.

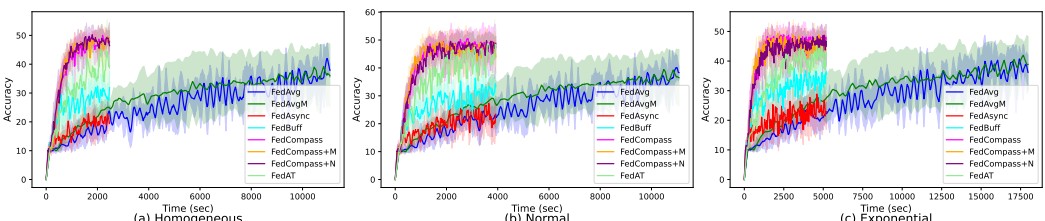

Figure 17: Change in validation accuracy during the training process for different FL algorithms on the *dual Dirichlet partitioned* CIFAR-10 dataset with twenty clients and three client heterogeneity settings.

### F.3  ADDITIONAL EXPERIMENT RESULTS ON THE FLAMBY DATASETS

Table 18 shows the top validation accuracy and the corresponding standard deviation on the Fed-IXI and Fed-ISIC2019 datasets from FLamby for various federated learning algorithms in different client heterogeneity settings among ten independent experiment runs with different random seeds. Figures 18 and 19 present the change in validation accuracy during the training process for various FL algorithms on the Fed-IXI and Fed-ISIC2019 datasets, respectively.

Table 18: Average top accuracy and standard deviation on the FLamby datasets for various federated learning methods in different client heterogeneity settings.

| | Fed-IXI | | | Fed-ISIC2019 | | |
|---|---|---|---|---|---|---|
| Method | homo | normal | exp | homo | normal | exp |
| FedAvg | 98.39±0.02 | 98.39±0.02 | 98.39±0.02 | 59.12±1.38 | 59.12±1.38 | 59.12±1.38 |
| FedAvgM | 98.39±0.00 | 98.39±0.00 | 98.39±0.00 | 50.25±2.57 | 50.25±2.57 | 50.25±2.57 |
| FedAsync | 98.05±0.25 | 98.24±0.18 | 98.32±0.20 | 59.43±1.18 | 59.66±3.04 | 56.34±4.34 |
| FedBuff | 98.36±0.03 | 98.35±0.03 | 98.35±0.08 | 57.65±2.07 | 57.46±2.60 | 55.13±4.74 |
| FedAT | 98.72±0.01 | 98.65±0.06 | 98.63±0.08 | 58.31±2.35 | 58.55±1.76 | 61.93±4.56 |
| FedCompass | 98.76±0.00 | 98.61±0.06 | 98.59±0.04 | 68.46±0.38 | 66.35±3.76 | 65.46±5.33 |
| FedCompass+M | 98.76±0.00 | 98.61±0.06 | 98.59±0.04 | 64.72±0.94 | 62.43±5.23 | 60.18±4.95 |
| FedCompass+N | 98.72±0.01 | 98.61±0.06 | 98.59±0.04 | 69.59±0.06 | 67.04±3.56 | 66.48±4.84 |

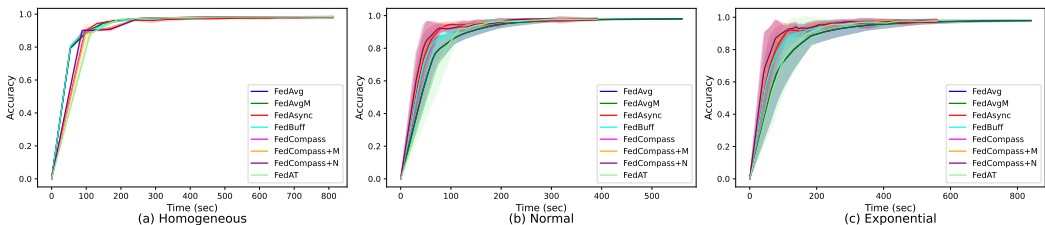

Figure 18: Change in validation accuracy during the training process for different federated learning algorithms on the FLamby Fed-IXI dataset with three different client heterogeneity settings.

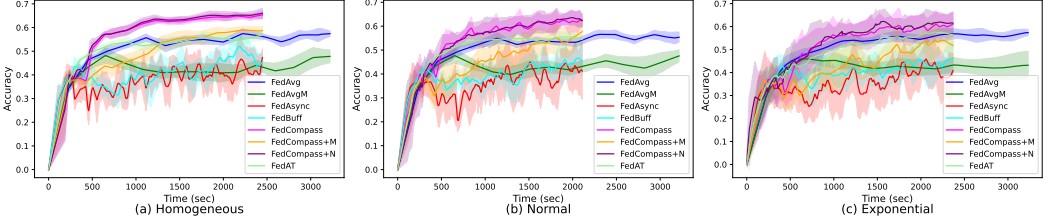

Figure 19: Change in validation accuracy during the training process for different federated learning algorithms on the FLamby Fed-ISIC2019 dataset with three different client heterogeneity settings.

## G  ABLATION STUDY

### G.1  ABLATION STUDY ON $Q_{\min}/Q_{\max}$

In this section, we conduct ablation studies on the values of $1/Q = Q_{\min}/Q_{\max}$ to investigate its impact on the performance of the FedCompass algorithm. Table 19 lists the relative wall-clock time for different values of $1/Q = Q_{\min}/Q_{\max}$ to reach 90% validation accuracy on the *class partitioned* and *dual Dirichlet partitioned* MNIST datasets, and 50% accuracy on the *dual Dirichlet partitioned* CIFAR10 datasets in different client heterogeneity settings, and Table 20 shows the corresponding the validation accuracy and standard deviations. The results are the average of ten different runs with random seeds. Table 19 also lists the results of FedBuff for reference. From Table 19, the following observations can be derived: (1) A smaller $Q_{\min}$ usually leads to faster convergence speed, which is consistent with the convergence analysis from Section 4 where smaller $1/Q$ results in fewer existing arrival groups at equilibrium and may help to reduce the impact of

device heterogeneity. (2) Though `FedCompass` achieves faster convergence rate with smaller *1/Q* (0.05–0.2) in general, it still can achieve reasonable convergence rate with large *1/Q* values (0.2–0.8) compared with `FedBuff`, which shows that `FedCompass` is not sensitive to the choice of the value of $Q$. (3) In the extreme case that $Q = 1$, i.e. $Q_{\min} = Q_{\max}$, there is a significant performance drop for `FedCompass`, as the `Compass` scheduler is not able to assign different number of local steps to different clients, so each client has its own group, and the `FedCompass` algorithm becomes equivalent to the `FedAsync` algorithm.

Table 19: Relative wall-clock time to first reach the target validation accuracy on the MNIST and CIFAR10 datasets for various values of *1/Q* $= Q_{\min}/Q_{\max}$ with different number of clients and client heterogeneity settings.

| Dataset | *1/Q* | $m = 5$ | | | $m = 10$ | | | $m = 20$ | | |
|---|---|---|---|---|---|---|---|---|---|---|
| | | homo | normal | exp | homo | normal | exp | homo | normal | exp |
| MNIST-*Class Partition* (90%) | 0.05 | 1.04× | 1.16× | 0.92× | 0.93× | 0.90× | 0.95× | 0.98× | 0.93× | 0.99× |
| | 0.10 | 1.00× | 1.38× | 0.93× | 1.03× | 0.92× | 0.93× | 1.01× | 1.03× | 1.02× |
| | 0.20 | 1.00× | 1.00× | 1.00× | 1.00× | 1.00× | 1.00× | 1.00× | 1.00× | 1.00× |
| | 0.40 | 0.99× | 1.67× | 2.48× | 0.98× | 0.98× | 1.41× | 1.01× | 1.19× | 1.10× |
| | 0.60 | 0.97× | 1.23× | 2.64× | 0.98× | 0.97× | 1.28× | 1.02× | 1.31× | 1.11× |
| | 0.80 | 0.96× | 1.33× | 1.51× | 0.98× | 1.21× | 1.23× | 1.01× | 1.61× | 1.28× |
| | 1.00 | 2.15× | 2.34× | 2.35× | - | 2.17× | 1.32× | - | 4.44× | 1.32× |
| `FedBuff` | | 1.30× | 1.57× | 1.81× | 1.58× | 1.39× | 1.43× | 1.37× | 1.18× | 1.39× |
| MNIST-*Dirichlet Partition* (90%) | 0.05 | 0.89× | 1.01× | 0.85× | 0.96× | 0.97× | 0.82× | 1.00× | 1.02× | 0.96× |
| | 0.10 | 0.91× | 1.05× | 0.92× | 0.96× | 0.95× | 0.85× | 0.97× | 1.02× | 0.96× |
| | 0.20 | 1.00× | 1.00× | 1.00× | 1.00× | 1.00× | 1.00× | 1.00× | 1.00× | 1.00× |
| | 0.40 | 1.02× | 1.03× | 1.70× | 1.04× | 1.05× | 1.28× | 1.00× | 1.14× | 1.25× |
| | 0.60 | 1.04× | 1.01× | 1.83× | 1.00× | 1.01× | 1.19× | 0.92× | 1.11× | 1.16× |
| | 0.80 | 1.04× | 1.13× | 1.58× | 0.91× | 1.25× | 1.22× | 0.86× | 1.22× | 1.21× |
| | 1.00 | 1.78× | 1.68× | 2.01× | - | 2.28× | 1.29× | - | - | 1.76× |
| `FedBuff` | | 1.23× | 1.52× | 1.86× | 1.81× | 1.51× | 1.49× | 2.77× | 1.22× | 1.40× |
| CIFAR10-*Dirichlet Partition* (50%) | 0.05 | 1.27× | 0.97× | 0.76× | 1.05× | 0.98× | 1.19× | 1.05× | 0.97× | 0.90× |
| | 0.10 | 1.15× | 1.01× | 0.73× | 0.98× | 1.11× | 1.27× | 1.01× | 1.04× | 1.01× |
| | 0.20 | 1.00× | 1.00× | 1.00× | 1.00× | 1.00× | 1.00× | 1.00× | 1.00× | 1.00× |
| | 0.40 | 1.28× | 1.06× | 1.36× | 1.02× | 1.06× | 2.19× | 0.96× | 1.14× | 1.08× |
| | 0.60 | 1.23× | 0.99× | 1.65× | 0.97× | 1.61× | 2.52× | 1.11× | 1.06× | 1.74× |
| | 0.80 | 1.23× | 1.24× | 1.64× | 1.32× | 1.91× | 3.44× | 1.33× | 2.95× | 1.64× |
| | 1.00 | - | 3.18× | 1.70× | - | - | 5.35× | - | - | - |
| `FedBuff` | | 1.62× | 1.99× | 1.67× | - | - | 2.42× | - | - | - |

## G.2 CLIENT SPEED CHANGES IN FEDCOMPASS AND ABLATION STUDY ON $\lambda$

In cross-silo federated learning scenarios, variations in computing power and speeds among clients are possible. For instance, a client utilizing a supercomputer might experience significant changes in computing power due to the auto-scaling after the allocation or deallocation of other tasks within the same cluster. Similarly, in cloud computing environments, a client might be subject to auto-scaling policies that dynamically adjust the number of virtual machines or container instances. In this section, we provide analyses and empirical results to study how `FedCompass` is resilient to client speed changes.

Figure 20 depicts an execution of `FedCompass` involving a client speedup scenario. During the second round of local training, client 3 enhances its computing speed from 4 step/min to 5 step/min. Consequently, client 3 transmits its local updates $\Delta_3$ at 10:36, earlier than the expected group arrival time $T_a = 12:00$. In this situation, `FedCompass` server first updates the speed record for client 3 and then waits for the arrival of clients 1 and 2 in the same group. Upon the arrival of clients 1 and 2 at 12:00, `FedCompass` server assigns all three clients to arrival group 2 for the subsequent round of local training, and the number of local steps for client 3 are assigned according to its new speed.

Figures 21 and 22 illustrate two executions of `FedCompass` with different levels of client slow-down. In Figure 21, the speed of client 3 changes from 4 step/min to 3.33 step/min during its second

Table 20: Average top validation accuracy and standard deviation for various values of $1/Q = Q_{\min}/Q_{\max}$ with different number of clients and client heterogeneity settings.

| | | $m = 5$ | | | $m = 10$ | | | $m = 20$ | | |
|---|---|---|---|---|---|---|---|---|---|---|
| Dataset | $1/Q$ | homo | normal | exp | homo | normal | exp | homo | normal | exp |
| MNIST Class Partition (90%) | 0.05 | 95.20±2.53 | 95.21±2.90 | 97.22±1.67 | 94.29±3.80 | 94.88±3.73 | 97.49±1.55 | 98.56±0.43 | 98.80±0.17 | 98.75±0.20 |
| | 0.10 | 95.03±3.13 | 95.35±2.15 | 97.07±1.67 | 94.72±3.56 | 94.65±3.74 | 97.71±1.25 | 98.57±0.38 | 98.80±0.11 | 98.82±0.18 |
| | 0.20 | 95.06±2.89 | 95.55±2.89 | 96.34±1.89 | 94.33±3.80 | 95.47±3.05 | 97.51±1.66 | 98.49±0.45 | 98.67±0.36 | 98.73±0.47 |
| | 0.40 | 94.44±3.26 | 95.34±3.63 | 92.73±3.09 | 94.42±3.94 | 95.01±3.52 | 96.91±2.13 | 98.58±0.36 | 98.74±0.26 | 98.75±0.32 |
| | 0.60 | 95.29±2.75 | 95.25±3.36 | 94.90±0.97 | 94.17±3.88 | 95.38±2.55 | 97.73±0.86 | 98.48±0.56 | 98.80±0.20 | 98.80±0.19 |
| | 0.80 | 95.61±2.34 | 95.35±2.30 | 95.04±3.13 | 94.77±3.59 | 96.08±2.60 | 97.35±1.49 | 98.60±0.45 | 98.72±0.25 | 98.74±0.25 |
| | 1.00 | 90.60±6.96 | 92.53±5.57 | 92.44±3.07 | 86.73±6.62 | 92.27±3.82 | 96.25±2.46 | 84.26±7.96 | 90.98±2.48 | 98.01±0.89 |
| MNIST Dirichlet Partition (90%) | 0.05 | 94.10±4.12 | 94.30±4.18 | 95.81±3.12 | 95.64±3.15 | 95.75±2.98 | 96.17±3.01 | 96.22±2.25 | 97.40±1.82 | 97.04±1.97 |
| | 0.10 | 94.14±4.08 | 94.14±4.44 | 96.11±3.04 | 95.49±3.32 | 95.65±3.23 | 96.78±2.56 | 96.39±1.92 | 97.59±1.28 | 97.68±1.28 |
| | 0.20 | 94.15±4.26 | 94.36±4.28 | 95.46±3.59 | 95.53±3.21 | 95.41±3.52 | 96.34±2.64 | 96.41±1.96 | 97.62±1.57 | 97.86±0.71 |
| | 0.40 | 94.12±4.11 | 94.27±4.11 | 95.40±2.46 | 95.46±3.47 | 95.67±3.14 | 95.86±3.35 | 96.07±2.45 | 98.03±0.93 | 97.85±1.00 |
| | 0.60 | 94.09±4.10 | 94.65±4.11 | 94.29±3.43 | 95.43±3.36 | 95.89±3.26 | 95.27±4.71 | 96.37±2.35 | 98.04±0.68 | 97.66±1.14 |
| | 0.80 | 94.23±3.89 | 94.71±3.56 | 95.09±2.62 | 95.32±3.55 | 95.46±3.20 | 95.26±4.36 | 96.72±2.20 | 97.51±0.84 | 97.66±0.62 |
| | 1.00 | 92.11±2.99 | 92.96±3.70 | 93.29±3.08 | 86.31±4.51 | 90.72±4.30 | 94.06±5.91 | 78.49±6.28 | 85.82±5.31 | 95.90±1.58 |
| CIFAR10 Dirichlet Partition (50%) | 0.05 | 59.40±2.28 | 60.86±1.94 | 61.79±3.27 | 54.04±4.68 | 53.85±6.74 | 57.36±1.90 | 50.58±2.26 | 50.62±2.58 | 51.40±1.68 |
| | 0.10 | 59.50±2.69 | 61.05±2.26 | 61.70±2.92 | 53.91±4.00 | 55.13±5.42 | 56.36±2.60 | 51.15±2.64 | 50.81±2.39 | 51.08±1.90 |
| | 0.20 | 59.29±3.49 | 59.98±3.65 | 61.23±2.67 | 54.51±3.50 | 54.95±3.68 | 57.29±1.98 | 51.01±2.20 | 52.03±2.75 | 51.39±2.12 |
| | 0.40 | 59.25±3.49 | 60.85±2.23 | 59.44±3.22 | 53.29±5.17 | 56.47±2.83 | 55.58±2.04 | 50.90±2.65 | 50.26±2.04 | 51.43±1.32 |
| | 0.60 | 59.25±2.79 | 60.76±2.24 | 59.01±3.90 | 52.92±4.00 | 56.45±3.05 | 54.76±2.71 | 50.67±2.20 | 50.33±2.04 | 51.44±1.79 |
| | 0.80 | 59.10±3.03 | 60.53±4.18 | 58.79±3.68 | 53.11±2.50 | 55.78±3.00 | 53.23±2.03 | 49.83±2.04 | 49.63±2.10 | 51.01±0.96 |
| | 1.00 | 44.38±4.71 | 47.67±4.29 | 51.21±4.99 | 33.52±3.29 | 38.13±2.07 | 47.62±2.59 | 34.89±3.34 | 36.12±1.69 | 39.92±3.39 |

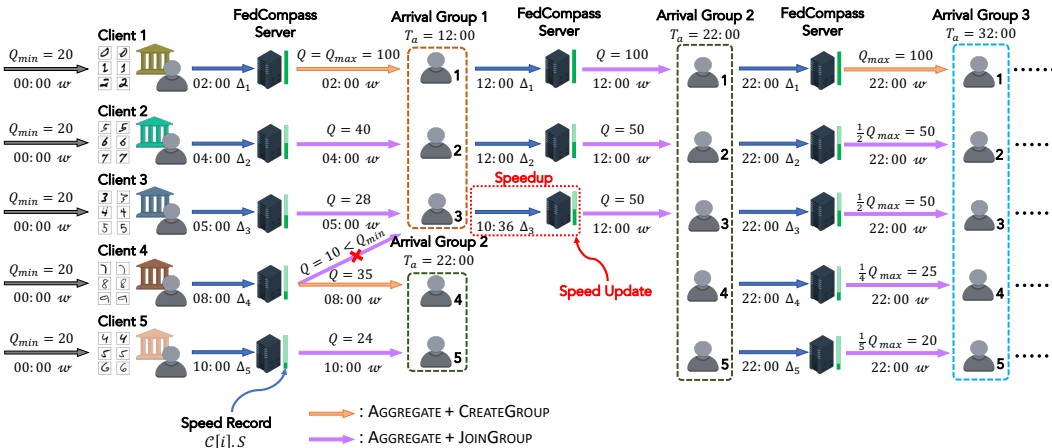

Figure 20: Overview of an execution of `FedCompass` on five clients with the minimum number of local steps $Q_{\min} = 20$, maximum number of local steps $Q_{\max} = 100$, and client speedup.

round of training. Consequently, the client transmits its local update $\Delta_3$ back to the server at 13:24, surpassing the expected arrival time $T_a = 12{:}00$. Nonetheless, it is important to recall that each arrival group has the latest arrival time $T_{\max}$ to accommodate potential client speed fluctuations. The time interval between $T_{\max}$ and the group's creation time is $\lambda$ times longer than the difference between $T_a$ and the group's creation time. For this example, we select $\lambda = 1.2$, resulting in $T_{\max} = 14{:}00$ for arrival group 1. Since client 3 arrives later than $T_a$ but earlier than $T_{\max}$, the `FedCompass` server waits until client 3 arrives at 13:24, subsequently assigning clients 1 to 3 to a new arrival group for the next training round. Figure 22 presents another case where the client has a relatively larger slowdown. In this example, the speed of client 3 diminishes from 4 step/min to 2.5 step/min, causing it to transmit local update $\Delta_3$ at 15:12, even exceeding $T_{\max} = 14{:}00$. When the `FedCompass` server does not receive an update from client 3 by the latest arrival time 14:00, it responds to clients 1 and 2. The server assigns these clients to arrival group 2 with corresponding numbers of local steps. Upon client 3 arrives at 15:12, the server first attempts to assign it to group 2, but this is unsuccessful due to the corresponding number of steps ($Q = (22{:}00{-}15{:}12) * 2.5$ (step/min) $= 17$) falling below the $Q_{\min} = 20$. As a result, the server establishes a new group 3 for client 3. Subsequently, once all clients in group 2 arrive at $22:00$, the server assigns all of them to group 3.

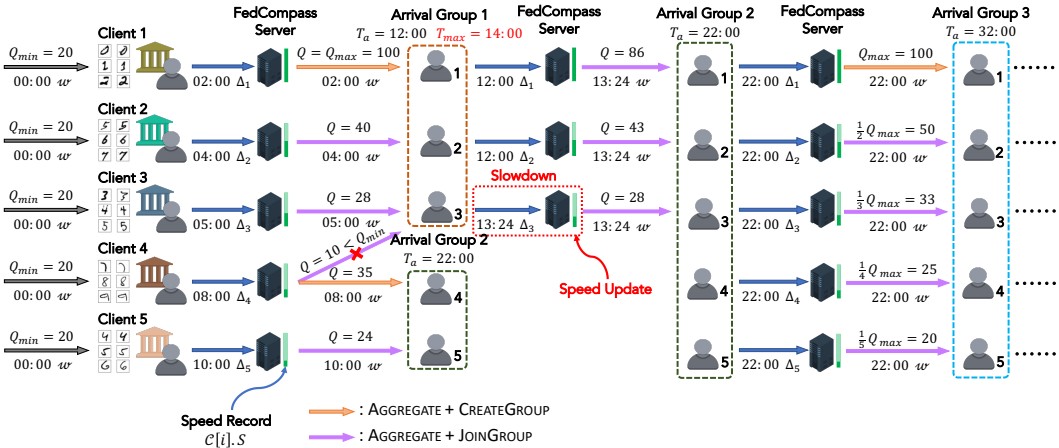

Figure 21: Overview of an execution of `FedCompass` on five clients with the minimum number of local steps $Q_{\min} = 20$ and maximum number of local steps $Q_{\max} = 100$ with client slowdown.

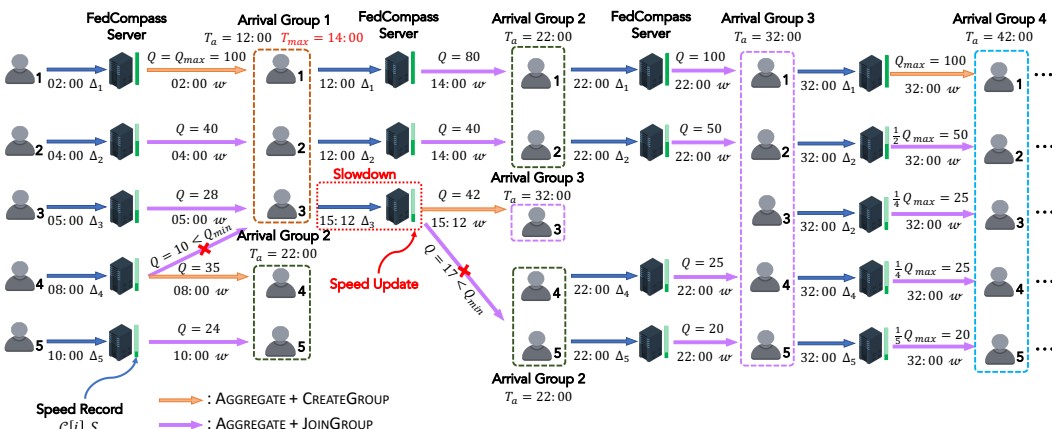

Figure 22: Overview of an execution of `FedCompass` on five clients with the minimum number of local steps $Q_{\min} = 20$ and maximum number of local steps $Q_{\max} = 100$ with client slowdown.

To empirically study how `FedCompass` is resilient to client speed changes compared to similar tiered FL methods such as `FedAT` (Chai et al., 2021) and how the value of $\lambda$ may impact the resilience, we set up a scenario such that each client has a 10% probability to change its computing speed by re-sampling a speed from the client speed distribution (normal or exponential distribution) in each local training round. This setup aims to mimic the occasional yet substantial changes in speed that may arise in real-world cross-silo federated learning environments as a result of auto-scaling. Table 21 shows the convergence speed for the `FedCompass` algorithm with different values of $\lambda$ and the `FedAT` algorithm, and Table 22 shows the corresponding average top accuracy and standard deviations. From Table 21, we can draw the following conclusions: (1) When $\lambda = 1.00$, the `Compass` scheduler is not resilient to even small client slowdown, so `FedCompass` converges relatively slower compared with some $\lambda$ values slightly above 1.00 in general. (2) $\lambda$ values slightly above 1.00 (e.g., $1.20 - 1.50$) result in faster convergence from the empirical results as the `Compass` scheduler accounts for minor client speed fluctuations without a long waiting time. (3) Large $\lambda$ values sometimes slow down the convergence as the server needs to experience a long waiting time when there are significant client speed slowdowns. (4) `FedCompass` converges faster than `FedAT` regardless of the value of $\lambda$, which further demonstrates its resilience to speed changes.

Finally, it is noteworthy that under extreme circumstances, where all client computing speeds are subject to dramatic and constant fluctuations, the scheduling mechanism of `FedCompass` may be unable to maintain the desired client synchronization. This is due to the reliance on prior speed data, which becomes obsolete in the face of such variability. In these scenarios, the training process would proceed in a completely asynchronous manner, effectively causing `FedCompass` to revert to a similar behavior of the `FedAsync` (Xie et al., 2019) algorithm. However, different from cross-

device FL where each client may experience frequent speed changes due to power constraints, the operation of other concurrent applications, or network connectivity issues. In practice, a cross-silo FL client usually has dedicated computing resources such as HPC with a job scheduler or some cloud computing facilities—where significant changes in computing speed are less frequent, generally arising from events like auto-scaling.

Table 21: Relative wall-clock time to first reach the target validation accuracy on the MNIST datasets for various values of $\lambda$ with different numbers of clients and client heterogeneity settings, and each client has 10% probability to change its speed in each local training round.

| Dataset | $\lambda$ | $m = 5$ | | $m = 10$ | | $m = 20$ | |
|---|---|---|---|---|---|---|---|
| | | normal | exp | normal | exp | normal | exp |
| MNIST-*Class Partition* (90%) | 1.00 | 1.00× | 1.00× | 1.00× | 1.00× | 1.00× | 1.00× |
| | 1.20 | 0.64× | 0.83× | 0.84× | 0.88× | 0.86× | 0.90× |
| | 1.50 | 0.67× | 1.00× | 0.72× | 1.01× | 1.01× | 1.00× |
| | 2.00 | 0.68× | 1.11× | 0.80× | 1.11× | 1.04× | 1.37× |
| FedAT | | 3.13× | 3.20× | 4.04× | 5.92× | 1.66× | 3.33× |
| MNIST-*Dirichlet Partition* (90%) | 1.00 | 1.00× | 1.00× | 1.00× | 1.00× | 1.00× | 1.00× |
| | 1.20 | 0.63× | 1.05× | 0.75× | 0.62× | 0.79× | 0.98× |
| | 1.50 | 0.55× | 1.20× | 0.75× | 0.62× | 0.85× | 1.16× |
| | 2.00 | 0.55× | 1.29× | 0.76× | 0.83× | 0.86× | 1.28× |
| FedAT | | 1.03× | 1.55× | 1.41× | 1.61× | 1.71× | 2.94× |

Table 22: Average top validation accuracy and standard deviation on the MNIST datasets for various values of $\lambda$ with different numbers of clients and client heterogeneity settings, and each client has 10% probability to change its speed in each local training round.

| Dataset | $\lambda$ | $m = 5$ | | $m = 10$ | | $m = 20$ | |
|---|---|---|---|---|---|---|---|
| | | normal | exp | normal | exp | normal | exp |
| MNIST-*Class Partition* (90%) | 1.00 | 97.01±1.19 | 97.57±1.01 | 96.55±1.07 | 97.45±2.17 | 98.78±0.03 | 98.76±0.15 |
| | 1.20 | 96.19±2.46 | 97.54±1.54 | 96.78±2.94 | 97.03±2.71 | 98.86±0.10 | 98.90±0.05 |
| | 1.50 | 95.99±2.30 | 97.59±1.98 | 96.25±3.26 | 97.44±1.90 | 98.86±0.09 | 98.71±0.26 |
| | 2.00 | 96.29±1.77 | 97.84±1.06 | 96.37±3.17 | 97.37±2.29 | 98.86±0.09 | 98.83±0.13 |
| FedAT | | 93.56±6.40 | 94.97±3.86 | 93.93±3.51 | 94.20±3.36 | 97.98±1.42 | 97.98±1.32 |
| MNIST-*Dirichlet Partition* (90%) | 1.00 | 91.15±4.48 | 94.34±2.84 | 94.88±4.40 | 96.90±1.53 | 98.08±0.34 | 98.09±0.23 |
| | 1.20 | 92.03±4.10 | 93.92±3.45 | 96.59±2.10 | 97.19±1.82 | 98.16±0.13 | 98.22±0.19 |
| | 1.50 | 91.82±4.35 | 93.32±4.03 | 96.04±3.36 | 97.47±1.42 | 98.28±0.13 | 98.16±0.19 |
| | 2.00 | 92.09±4.11 | 93.08±3.61 | 95.77±3.29 | 97.36±1.68 | 98.16±0.20 | 98.26±0.07 |
| FedAT | | 91.43±3.31 | 92.36±2.50 | 95.49±2.94 | 96.36±2.36 | 97.65±0.44 | 97.68±0.35 |

