# OpenReview forum: "FedCompass: Efficient Cross-Silo Federated Learning on Heterogeneous Client Devices Using a Computing Power-Aware Scheduler"
_ICLR.cc/2024/Conference — ICLR 2024 poster_

### Official Review · Reviewer_vABe · 2023-10-25

**Soundness:** 4 excellent
**Presentation:** 3 good
**Contribution:** 2 fair
**Rating:** 8
**Confidence:** 4

**Summary:**

This paper proposes a semi-asynchronous federated learning framework named FedCompass, which adjusts the local training steps of each client according to their computing power and groups clients with similar training times in asynchronous federated learning. The proposed framework effectively alleviates the staleness issue in asynchronous federated learning while avoiding prolonged waiting time in synchronous federated learning.

**Strengths:**

FedCompass is a semi-asynchronous federated learning framework that benefits from both asynchronous and synchronous federated learning while avoiding the drawbacks of these two frameworks. More specifically,

1.	FedCompass effectively addresses the staleness issue in asynchronous federated learning by grouping clients and thereby reducing the frequency of model aggregation on the server. By adaptively setting numbers of local epochs for each client, FedCompass ensures that the clients in each group can complete the training in a similar time, which avoids prolonged waiting time in synchronous federated learning.

2.	A theoretical analysis of the convergence of FedCompass is provided to support the effectiveness of FedCompass.

3.	Experimental results on four datasets with different statistical and systematic heterogeneity are conducted, and the results show that FedCompass outperforms previous asynchronous and synchronous methods.

**Weaknesses:**

FedCompass seems to be an incremental improvement based on Tiered Federated Learning by complementing adaptive numbers of local epochs. The differences between FedCompass and Tired Federated Learning are not clear and not significant. Though the authors claim that Tired Federated Learning cannot deal with time-varying computing power, it is not a serious issue in cross-silo federated learning where the local devices usually have sufficient computing power for running multiple tasks at the same time. In addition, experiments do not provide any comparison between FedCompass and Tired Federated Learning. To highlight the contribution of this work, the authors need to give more elaborations and empirical evidence for the significance of FedCompass compared with Tired Federated Learning.

In addition, the explanation of Algorithm 1 in Section 3.2 is more difficult to follow than the explanation in Figure 1. The authors may need to improve the clarity of this part since it is the core of the proposed method.

**Questions:**

See the weaknesses above.

---

> ### Author Response · Authors · 2023-11-14
> **Response to Reviewer vABe**
>
> Dear Reviewer vABe,
>
> First and foremost, we would like to express our sincere gratitude for taking the time to review our paper and providing insightful comments. Our responses to your comments are as follows.
>
> **Comparison with Tiered FL algorithms**
>
> The primary distinction between tiered FL algorithms and FedCompass lies in their targeted FL settings and client scale. Tiered FL algorithms are specifically designed for cross-device FL settings, where the client count can range from hundreds to thousands and even more. In contrast, FedCompass is designed for cross-silo FL scenarios and focuses on optimizing the use of dedicated computing resources at each silo, typically involving client numbers on the order of ten.
>
> The FedAT algorithm aims to develop a communication-efficient asynchronous FL method  (Chat et al., 2021). It features synchronous updates within tiers and asynchronous updates among them. The method employs random client sampling within tiers and rough tier assignment broadly based on client response latency. However, such design choices are not well-suited for cross-silo FL, where all clients are expected to participate in each round of training. This fundamental difference in settings makes it difficult to directly compare FedCompass with tiered FL methods. Moreover, the granularity of tiers in tiered FL algorithms is generally coarse. In scenarios with a small number of clients, it's likely that only one tier is formed, effectively making the algorithm the same as the FedAvg algorithm (McMahan et al., 2017).
>
> In terms of the resilience to client speed changes, we would like to first elaborate on what this means in the context of cross-silo FL. Unlike cross-device FL where a client experiences frequent changes due to the unreliability and intermittent computing of the devices, for cross-silo FL, each client usually has dedicated and reliable computing resources on HPC or cloud. It is possible that the clients have significant yet occasional speed changes due to some auto-scaling behaviors. FedCompass is designed to be responsive and resilient for those significant speed changes as showcased by the ablation study results in Appendix G.2. However, tier FL methods usually lack response to sudden speed changes and may experience performance degradation.
>
> **Unclarity in Section 3.2**
>
> Thanks a lot for pointing this out and we agree that the previous presentation lacks clarity. Now we have rewrote the paragraph for describing the server side algorithm of FedCompass by focusing on the higher-level idea instead of the low-level details (last two paragraphs at the end of page 5). We believe that now it is a more clear presentation of the algorithm.
>
> **Summary of changes in the paper in response to Reviewer vABe**
>
> (1) Rewrite Section 3.2 for better presentation clarity, and have the descriptions focus more on higher-level design ideas.
>
> (2) Add few sentences at the first and last paragraphs of Appendix G.2 to further explain the resilience of FedCompass to client speed changes in practical cross-silo FL settings.
>
> Chai, Z., Chen, Y., Anwar, A., Zhao, L., Cheng, Y., & Rangwala, H. (2021, November). FedAT: A high-performance and communication-efficient federated learning system with asynchronous tiers. In Proceedings of the International Conference for High Performance Computing, Networking, Storage and Analysis (pp. 1-16).
>
> McMahan, B., Moore, E., Ramage, D., Hampson, S., & y Arcas, B. A. (2017, April). Communication-efficient learning of deep networks from decentralized data. In Artificial intelligence and statistics (pp. 1273-1282). PMLR.

---

> > ### Comment · Reviewer_vABe · 2023-11-18
> >
> > Thanks for your reply. The clarity issue has been addressed in the revised version, but I am still not convinced by your claim that Tired FL is not comparable with FedCompass in cross-silo settings. You can replace the client sampling in Tired FL with full participation for the cross-silo setting, and you can also use more fine-grained tires in Tired FL such that you can have multiple tires in cross-silo settings. These are not technical difficulties of applying Tired FL in cross-silo settings. Meanwhile, an empirical comparison between FedCompass and Tired FL in your cross-silo settings can also provide support for your claim that Tired FL cannot deal with sudden speed changes. Accordingly, I will maintain my score at 6 unless more experimental comparisons between FedCompass and Tired FL are provided.

---

> > > ### Author Response · Authors · 2023-11-18
> > > **Response to Reviewer vABe Regarding Adding Tiered FL**
> > >
> > > Dear Reviewer vABe,
> > >
> > > Thanks for you response! Yes, we now agree that adding a comparison with a tiered FL algorithm will provide further empirical evidence to support FedCompass. We plan to start benchmarking the FedAT algorithm (Chai et al., 2021) and will let you know when we add the results.
> > >
> > > Chai, Z., Chen, Y., Anwar, A., Zhao, L., Cheng, Y., & Rangwala, H. (2021, November). FedAT: A high-performance and communication-efficient federated learning system with asynchronous tiers. In Proceedings of the International Conference for High Performance Computing, Networking, Storage and Analysis (pp. 1-16).

---

> > > ### Author Response · Authors · 2023-11-20
> > > **Response to Reviewer vABe with New Experiment Results for Tiered FL**
> > >
> > > Dear Reviewer vABe,
> > >
> > > Following your recommendation, we have incorporated new experimental results for a tiered FL algorithm, FedAT (Chat et al., 2021), in our revised paper, and we further compare FedCompass with FedAT in scenarios with sudden speed changes in our ablation studies. Additionally, the updated source codes are provided in the supplementary materials. Key highlights from our experiments are as follows:
> > >
> > > (1) First, we would like to express our great appreciation for your invaluable suggestions to incorporate a tiered FL algorithm (FedAT)  in the comparison, as we do find that FedAT outperforms the other two asynchronous algorithms (FedAsync and FedBuff) in cases with relatively small client heterogeneity or on more complex datasets such as CIFAR-10 and Fed-ISIC2019.
> > >
> > > (2) From the speedup results in Tables 1 and 2, we can see that FedCompass consistently outperforms FedAT (except in one case with 20 homogeneous clients on the class-partitioned MNIST dataset). Notably, the speedup gets larger as the client heterogeneity increases. For example, when client speeds are subject to the exponential distribution, FedAT is more than 2 times slower than FedCompass. We attribute this to FedAT’s increased inner-tier waiting time in heterogeneous client settings, whereas FedCompass effectively reduces the inner-group waiting time via dynamic local step assignments and enforcing the group's latest arrival time.
> > >
> > > (3) The accuracy tables (Tables 14 to 18) in the Appendix, along with the training process visualization figures (Figure 2 and Figures 6 to 19), clearly demonstrate an accuracy advantage of FedCompass over FedAT, especially on complex datasets such as CIFAR-10 and Fed-ISIC2019.
> > >
> > > (4) Finally, in Appendix G.2, we conducted additional experiments to substantiate our claim that “FedCompass is more resilient to speed changes”. The results in Table 21 indicate that FedCompss achieves faster convergence than FedAT regardless of the latest time factor $\lambda$, in cases where each client has a 10% probability of significant speed changes in each local round.
> > >
> > > In conclusion, we believe that your recommendation to conduct comparative experiments with FedAT has significantly strengthened our assertion regarding the effectiveness of FedCompass. We also hope these new findings will further convince you of FedCompass's efficiency and effectiveness. We are grateful for your guidance and look forward to your feedback on these additional insights.
> > >
> > > Chai, Z., Chen, Y., Anwar, A., Zhao, L., Cheng, Y., & Rangwala, H. (2021, November). FedAT: A high-performance and communication-efficient federated learning system with asynchronous tiers. In Proceedings of the International Conference for High Performance Computing, Networking, Storage and Analysis (pp. 1-16).

---

> > > > ### Comment · Reviewer_vABe · 2023-11-20
> > > >
> > > > Thank you for your response and the new experimental results. All of my concerns have been addressed in the current version and I would like to increase my score to 8.

---

> > > > > ### Author Response · Authors · 2023-11-20
> > > > >
> > > > > Dear Reviewer vABe, we really appreciate your time and effort for the review and comments!

---

### Official Review · Reviewer_DhHF · 2023-10-31

**Soundness:** 2 fair
**Presentation:** 2 fair
**Contribution:** 2 fair
**Rating:** 6
**Confidence:** 3

**Summary:**

- This work proposes a new semi-asynchronous FL for cross-silo FL. Specifically, FedCompass is designed to track the computing time of each client and adaptively adjust the local epoch, which enables the server to simultaneously receive local models from clients. The proposed method not only mitigates the straggler issue but also provides a great platform for global aggregation. The authors established a theoretical convergence bound and experimentally confirmed that the proposed method converges faster and higher than other baselines.

**Strengths:**

- The approach of adjusting local updates differently for each client to reduce the staleness gap between models is interesting.
- Compared to existing asynchronous methods, the proposed method enables global aggregation, which is an important advantage since various existing aggregation-based methods (e.g., robust aggregation methods in FL) can be combined.
- The proposed method is well supported by theoretical analysis.

**Weaknesses:**

- The motivation is a bit unconvincing. The straggler issue is considered important primarily in cross-device FL since the organizations are expected to have sufficient computational/communication resources in cross-silo FL [arXiv’22].
- Most experiments were conducted on datasets with a small number of classes. To confirm the scalability of the proposed method, could the authors conduct experiments on datasets with more classes like CIFAR-100?
- The idea of adjusting local epochs for each client is straightforward and the technical novelty of the proposed method seems somewhat limited.

**Questions:**

See Weaknesses and,

- In Line 21-23 of Algorithm 1, it seems that each client’s update is accumulated one-by-one into group specific buffer. How can existing aggregation strategies (e.g., [ICLR’21]) be performed in the global aggregation step of the proposed algorithm?
- Could the authors conduct ablation study on Q_min/Q_max using CIFAR-10 dataset and compare the results with other baselines?
- There are some typos in Appendix that should be corrected (e.g., in equation (6) of the proof for Lemma 2, the gradient symbol is missing)

[arXiv’22] Chao et al., Cross-silo federated learning: Challenges and opportunities

[ICLR’21] Reddi et al., Adaptive Federated Optimization

---

> ### Author Response · Authors · 2023-11-14
> **Response to Reviewer DhHF (Part 1 of 2)**
>
> Dear Reviewer DhHF,
>
> First and foremost, we would like to express our sincere gratitude for taking the time to review our paper and providing insightful questions and comments. Our responses to your comments and questions are as follows.
>
> **Unconvincing motivations**
>
> The straggler issues in cross-device FL are mainly caused by the unreliability of the clients due to intermittent compute availability and connectivity. On the other hand, in cross-silo FL, though each client is assumed to have reliable and dedicated resources, the straggler issues come from the disparity/difference of the computing power among clients. In practical settings, there usually exists a substantial gap in computing capability between cross-silo FL clients.
>
> Firstly, the increasing adoption of specialized ASICs like those from Cerebras, SambaNova, and Graphcore, particularly by large institutions engaged in FL projects such as national laboratories, is noteworthy. At the other end of the spectrum, institutions like hospitals, also interested in utilizing federated learning, may only have access to standard consumer GPUs or solely CPU-based computing resources. This diversity in the types of accelerators and computing machines used by cross-silo FL clients inevitably leads to substantial disparities in computational capabilities.
>
> Secondly, disparities in computing power are also evident even among clients using the same type of accelerator, such as GPUs. For instance, HPC clusters built with different generations of GPUs and associated support hardware, like networking components, can exhibit significant differences in training capabilities. This is further elaborated in the following NVIDIA post (https://developer.nvidia.com/blog/nvidia-hopper-architecture-in-depth/).
>
> Considering these disparities among cross-silo clients' computing resources, it becomes crucial to address resource utilization efficiently. The aim is to prevent resource wastage, which can occur by having faster clients idly wait for slower ones and underutilizing the available resources. A typical scenario in HPC clusters illustrates this: when a user requests a node with 8 GPUs through a job scheduler like Slurm, they are billed for a fixed number of compute hours, regardless of whether all 8 GPUs are fully utilized throughout the job duration.
>
> **Using CIFAR-100 as datasets**
>
> In our experiments, we selected the MNIST and CIFAR-10 datasets to evaluate the performance of FedCompass. These two datasets are widely recognized and frequently utilized in FL research, as evidenced by Ma et al. (2022), who noted their prevalent use in the field. According to a survey presented in their study (Figure 9), MNIST is used in 32% of FL experiments, while CIFAR-10 is used in 28%. CIFAR-100, in contrast, is seldom chosen for such experiments. Beyond these traditional datasets, we also assessed FedCompass using two datasets from FLamby (Terrail et al., 2022), a recently introduced benchmark specifically designed for cross-silo FL. These datasets are significantly larger, at 444MB and 9GB, respectively. We believe this diverse range of datasets, both in terms of size and complexity, adequately demonstrates the versatility and efficacy of FedCompass in various scenarios.
>
> **Limited technical novelty**
>
> At first glance, the concept of assigning varying local steps to clients based on clients' computing speeds might seem straightforward. However, in practice, this approach encounters several challenges and edge cases. For example, these include synchronizing clients when there is no prior knowledge of their computing speeds, addressing minor and significant fluctuations in client speeds, and handling scenarios where there is a substantial disparity in client computing power (e.g., exponential client computing speed distributions).
>
> FedCompass not only addresses these practical challenges effectively but also provides theoretical proof of convergence and demonstrates robust performance through an extensive series of experiments. Therefore, we believe that this paper possesses sufficient novelty, underscoring its contribution to advancing FL methodologies.
>
> Ma, X., Zhu, J., Lin, Z., Chen, S., & Qin, Y. (2022). A state-of-the-art survey on solving non-IID data in Federated Learning. Future Generation Computer Systems, 135, 244-258.
>
> Terrail, J. O. D., Ayed, S. S., Cyffers, E., Grimberg, F., He, C., Loeb, R., ... & Andreux, M. (2022). Flamby: Datasets and benchmarks for cross-silo federated learning in realistic healthcare settings. arXiv preprint arXiv:2210.04620.

---

> ### Author Response · Authors · 2023-11-14
> **Response to Reviewer DhHF (Part 2 of 2)**
>
> **Integrating other server optimization strategies**
>
> Algorithm 1 only presents a simple demonstration of the FedCompass algorithm without combining it with other server-side optimization strategies. If the individual client update is needed to combine FedCompass with other strategies such as FedAdam, FedAdagrad, and FedYogi (Reddi et al., 2020), the server can simply buffer each client’s update instead of adding them up. Therefore, it only requires some minor changes in the implementation.
>
> **Adding ablation study on CIFAR-10**
>
> Sure! We have now added ablation study results for $Q_{\min}/Q_{\max}$ on the class partitioned CIFAR-10 datasets in Table 3 as well. The results are consistent with those on the MNIST dataset, namely, FedCompass can outperform FedBuff (Nguyen et al., 2022) with a large range of $Q_{\min}/Q_{\max}$ values.
>
> **Typos**
>
> Thanks for the reminder! We have fixed it in the revised paper.
>
> **Summary of changes in the paper in response to Reviewer DhHF**
>
> (1) Cite the survey paper to further explain our choice of experiment datasets in the first paragraph of Section 5.1
>
> (2) Add results for ablation study on CIFAR-10 in Table 3.
>
> (3) Fix typos in equation 6.
>
> Reddi, S., Charles, Z., Zaheer, M., Garrett, Z., Rush, K., Konečný, J., ... & McMahan, H. B. (2020). Adaptive federated optimization. arXiv preprint arXiv:2003.00295.
>
> Nguyen, J., Malik, K., Zhan, H., Yousefpour, A., Rabbat, M., Malek, M., & Huba, D. (2022, May). Federated learning with buffered asynchronous aggregation. In International Conference on Artificial Intelligence and Statistics (pp. 3581-3607). PMLR.

---

> ### Comment · Reviewer_DhHF · 2023-11-19
>
> I really appreciate the authors for their efforts to address the concerns and clarify the unclear aspects. But I have the remaining concerns below.
>
> 1)  Unconvincing motivation: Now, I understood there might be a significant gap in computing time between cross-silo FL clients, even if they have sufficient computing resources. However, I think the gap is more meaningful in large models/datasets, and it is unlikely to be a critical issue in small models/datasets such as MNIST and CIFAR-10, which are the main experimental results of this paper.
>
> 2) CIFAR-100 dataset: As the number of classes is large, more diverse heterogeneous scenarios can be simulated in FL. Therefore, I was just curious about the scalability of the proposed method with respect to the number of classes. Since this work addresses cross-silo FL where each client has sufficient computing power, I think this analysis would be valuable. I also found that the CIFAR-100 dataset has been adopted in many well-known studies such as FedDC, FedDyn, FedMD.
>
> 3) Limited technical novelty: I admit this work tackles a practically important problem in cross-silo FL. However, I found that the authors mentioned the convergence analysis is not a core contribution (in the response to Reviewer UVPW). Considering this, the main contributions of this work seem not to lie in machine learning algorithms but rather in the simple assistance for implementation. Therefore, I am not sure if this is above the acceptance threshold bar of ICLR.
>
> Therefore, for now, I will keep my original score.

---

> ### Author Response · Authors · 2023-11-19
> **Response to Reviewer DhHF**
>
> Dear Reviewer DhHF,
>
> Thanks for your reply and we would like to further clarify your concerns more specifically.
>
> **Unconvincing Motivation**
>
> While MNIST and CIFAR-10 are relatively small datasets, our experimental design is still challenging. We use two partition strategies, resulting in each client holding only a few classes. Further, this differs from cross-device FL scenarios where there are thousands of clients and each client may hold data of 3 to 4 classes. In cross-silo FL, there are only around 10 clients and each client only holds 3 to 4 classes, and this makes the problem much more challenging. As illustrated in the x-axis of Figures 12 to 17, experiments on the CIFAR-10 dataset usually take 1 (asynchronous algorithms) to 6 hours (asynchronous algorithms) to finish, indicating a significant saving of more than 5 GPU hours per client compared to synchronous methods like FedAvg. This translates to a total of **100 GPU hours** saved if there are 20 clients. Additionally, we incorporate two datasets (444MB and 9GB, both are larger than CIFAR100 - 161MB) from the FLamby cross-silo benchmark (Terrail et al., 2022) to further demonstrate FedCompass’s effectiveness. Therefore, we believe that our experiment design is still well-aligned with our motivation.
>
> **CIFAR-100 Datasets**
>
> In the experiment setting of this paper, each speedup/accuracy number is obtained from the average of 10 independent runs, taking 10 to 60 hours **per client** on CIFAR10 or CIFAR100 in total. Consequently, obtaining all accuracy/speedup numbers on CIFAR100 could exceed 10,000 GPU hours. Due to the limited time available for discussion, acquiring comprehensive CIFAR100 results is currently unfeasible, and we request your understanding in this matter.  Nonetheless, we still believe that our findings on the heterogeneous CIFAR10 dataset and the two FLamby datasets could highlight FedCompass’s efficiency and accuracy advantages over synchronous and other asynchronous methods, especially given the fact that CIFAR10 and CIFAR100 actually have the same dataset size (~161 MB) and the used two FLamby datasets have even larger sizes.
>
> **Limited Technical Novelty**
>
> We think there may exist some miscommunication or misunderstanding in terms of the theoretical contribution of this paper. Here is what we really want to express in terms of our convergence analysis part, and we also updated the responses the reviewer UVPW as well to further clarify things.
>
> (1) The main focus/contribution of our convergence analysis section is deriving the convergence behavior of our newly proposed semi-asynchronous algorithm FedCompass using the commonly-used assumptions and obtaining further insights, instead of trying to get rid of any commonly-used assumptions such as bounded staleness.
>
> (2) Though we are not the first ones to study the convergence behavior of the general asynchronous FL algorithms or FL algorithms with different client local steps, this paper actually first studies the convergence behavior of a semi-asynchronous FL algorithm with a scheduler assigning different numbers of local steps to clients, and we really believe this is still a good contribution to the FL community.
>
> We apologize for any previous confusion and look forward to further discussions.
>
> Terrail, J. O. D., Ayed, S. S., Cyffers, E., Grimberg, F., He, C., Loeb, R., ... & Andreux, M. (2022). Flamby: Datasets and benchmarks for cross-silo federated learning in realistic healthcare settings. arXiv preprint arXiv:2210.04620.

---

> > ### Comment · Reviewer_DhHF · 2023-11-22
> >
> > Thanks for your additional responses to my questions. I have further comments as follows:
> >
> > It is not clear that the current experimental setting (through gaussian assumption) regarding client heterogeneity could reflect practical scenarios. For example, if each client has sufficient resources (high quality CPU or GPU) above a certain level, it is unlikely that there will be a significant difference in computing time when performing small MNIST or CIFAR-10 tasks.
> >
> > Meanwhile, one additional question is, why does the proposed FedCompass converge faster than FedAvg in the homogeneous client setting where each client has the same computing time?

---

> > > ### Author Response · Authors · 2023-11-22
> > > **Response to Reviewer DhHF**
> > >
> > > Dear Reviewer DhHF,
> > >
> > > First, thank you for your continued engagement and valuable feedback on our paper! Here are our responses to your comments.
> > >
> > > For client heterogeneity simulation, we would like to mention that the key difference between the cross-silo FL client and the cross-device FL client is the **reliability** of the computing machine, as a cross-silo client is assumed to have a dedicated computing machine and reliable network, while a cross-device client may suffer from low battery power or network disconnection (such as our mobile phones).
> > >
> > > However, it is important to note that this reliability does not inherently imply that the cross-silo FL machine is always extremely powerful. In practice, cross-silo FL participants often allocate computing resources based on the requirements of the training task. For instance, if clients want to train a small CNN model, they might only allocate one GPU or just several CPUs for training (this allocation also varies among clients and causes heterogeneity). If they want to train a deeper neural network, they may allocate various numbers of GPUs.
> > >
> > > A suitable analogy here is the use of High-Performance Computing (HPC) resources. Even though a user might have access to an entire HPC cluster with numerous high-quality CPUs and GPUs, the user typically allocates only the necessary amount of resources for the computing job. Similarly, the cross-silo FL clients also allocate resources according to the training task, and of course, different clients may allocate different amounts of resources, which leads to client heterogeneity.
> > >
> > > Additionally, we would like to mention that employing random distributions to simulate the client computing speeds is a widely accepted approach in federated learning research. This technique models client computing heterogeneity as evidenced in several well-known papers (Xie et al., 2019, Charles et al., 2021, Reisizadeh et al., 2022, Nguyen et al., 2022).
> > >
> > > In terms of the difference between FedAvg and FedCompass in the “homogeneous” settings, this is a great question. Notably, the difference gets larger as the number of clients increases and the training tasks get more complicated (CIFAR10 is more obvious than MNIST). The key reason for that is the clients are actually not completely homogeneous (i.e., have exactly the same computing speeds), as we also apply another normal distribution with a small variance in each round to simulate the speed variability experienced by the clients during different training rounds. In scenarios with inherent client heterogeneity, this minor speed variance almost does not impact the relative speeds among clients. However, when all the clients have the same speed, such a small variation in each round matters and causes the scheduler’s speed estimation to have some fluctuations and cannot come to the conclusion that all clients have the same speed. As a consequence, this scheduler does not assign $Q_{\max}$ to all clients to make it exactly the same as FedAvg. Actually, many clients only obtain $0.3Q_{\max}\sim0.8Q_{\max}$ training steps (we upload one training log file `results_server.txt` in the supplementary materials to showcase this), and this adds some asynchronous features to the whole training process even in the so-called “homogeneous” scenarios. In such cases, FedCompass updates the global model much more frequently than FedAvg, namely, the clients train on the up-to-date global model more frequently, and this can be beneficial to the convergence, especially on the relatively more complicated training sets. Additionally, as the number of clients increases, the speed variance gets larger and the difference between FedCompass and FedAvg gets larger as well. We now add sentences of explanations to this as well in the first paragraph of appendix F.1. (we put it into the appendix as this paper mainly focuses on heterogeneous clients and it is almost impossible to have all clients having exactly the same inherent speed in practice).
> > >
> > > Xie, C., Koyejo, S., & Gupta, I. (2019). Asynchronous federated optimization. arXiv preprint arXiv:1903.03934.
> > >
> > > Charles, Z., Garrett, Z., Huo, Z., Shmulyian, S., & Smith, V. (2021). On large-cohort training for federated learning. Advances in neural information processing systems, 34, 20461-20475.
> > >
> > > Reisizadeh, A., Tziotis, I., Hassani, H., Mokhtari, A., & Pedarsani, R. (2022). Straggler-resilient federated learning: Leveraging the interplay between statistical accuracy and system heterogeneity. IEEE Journal on Selected Areas in Information Theory, 3(2), 197-205.
> > >
> > > Nguyen, J., Malik, K., Zhan, H., Yousefpour, A., Rabbat, M., Malek, M., & Huba, D. (2022, May). Federated learning with buffered asynchronous aggregation. In International Conference on Artificial Intelligence and Statistics (pp. 3581-3607). PMLR.

---

> > > > ### Comment · Reviewer_DhHF · 2023-11-22
> > > >
> > > > I appreciate your detailed responses to my further questions about "unconvincing motivation". Now, I feel this concern is well addressed, and therefore I have increased my score. However, since I still have concerns regarding the technical novelty in terms of machine learning literature, my confidence remains somewhat weak.

---

> > > > > ### Author Response · Authors · 2023-11-23
> > > > >
> > > > > Dear Reviewer DhHF,
> > > > >
> > > > > Thanks for your response, and we are glad that your concerns have been addressed.

---

### Official Review · Reviewer_7w8f · 2023-11-06

**Soundness:** 3 good
**Presentation:** 3 good
**Contribution:** 2 fair
**Rating:** 6
**Confidence:** 4

**Summary:**

This paper takes aim at tackling the combined problems of system and data heterogeneity in the context of cross-silo Federated Learning. The authors propose FedCompass a semi-asynchronous federated algorithm that assigns adaptively different amounts of training task to clients with different computational capabilities. Additionally, FedCompass ensures that received models are received in groups almost simultaneously reducing the staleness of local models while the overall process remains asynchronous eliminating long waiting periods for fast nodes. Theoretical results on the convergence of the proposed method are presented and experiments on both academic and real-world datasets support the theoretical findings.

**Strengths:**

-The paper tackles an interesting problem in the literature of FL, namely the problem of system and data heterogeneity. Improving on prior works this paper attempts to circumvent the weaknesses observed in synchronous and asynchronous federated methods (which boil down to long delays due to stragglers and stale models respectively).

-Theoretical results that prove the convergence of the proposed FedCompass are presented providing useful insights.

-Substantial experimental evidence has been provided showcasing the superiority of the proposed method compared to established synchronous and asynchronous baselines.

**Weaknesses:**

-The description of the algorithm in section 3.2 is unclear and further discussion is required. Specifically, the assignment of groups and the group aggregation are major components of the proposed method and they are not presented or discussed sufficiently in the main body of the paper (I strongly believe that including these components or at least an extensive discussion about them in the main body of the paper would significantly improve the presentation of this work). It is unclear how the implementation of these components alleviate the bias introduced by faster nodes performing more updates than slower nodes (a common issue met in previously proposed asynchronous methods). It is also unclear how the overall method  reaches an equilibrium (as discussed in section 3.1) and what properties it has. I would appreciate it if the authors could elaborate on the above.

-In the description in section 3.2 it seems that when a client arrives later than $G[g].T_{max}$ it is immediately assigned a new training task whereas the clients that arrive earlier than $G[g].T_{max}$ are assigned training tasks when the next aggregation takes place. What is the justification for this differentiation?

-Some of the assumptions required for the theoretical results are very restrictive. Specifically, the bounded gradient assumption rules out simple function such as the quadratic. Further, the bounded heterogeneity  and staleness assumption is rather strong and the derived results in Theorem 1 and Corollary 1 have a heavy dependence on  quantity $\mu$. As a result the impact of those results is diminished.

-The related work section could be extended with works on device heterogeneity (Reisizadeh et al., 2022; Horvath et al., 2022) and on asynchronous FL (So et al., 2021)

Reisizadeh, A., Tziotis, I., Hassani, H., Mokhtari, A., and Pedarsani, R. (2022). Straggler-resilient federated
learning: Leveraging the interplay between statistical accuracy and system heterogeneity.

Horváth, S., Sanjabi, M., Xiao, L., Richtárik, P., and Rabbat, M. (2022). Fedshuffle: Recipes for better use
of local work in federated learning.

So, J., Ali, R. E., Güler, B., and Avestimehr, A. S. (2021). Secure aggregation for buffered asynchronous
federated learning.

**Questions:**

See the weaknesses section.

---

> ### Author Response · Authors · 2023-11-14
> **Response to Reviewer 7w8f (Part 1 of 2)**
>
> Dear Reviewer 7w8f,
>
> First and foremost, we would like to express our sincere gratitude for taking the time to review our paper and providing insightful questions and comments. Our responses to your comments are as follows.
>
> **Unclear description in Section 3.2**
>
> For Section 3.2, we now rewrote the paragraph for describing the server-side algorithm of FedCompass by focusing on the higher-level idea instead of the low-level details (last two paragraphs at the end of page 5). We believe that now it is a more clear presentation of the algorithm.
>
> **How FedCompass alleviates the bias towards faster clients**
>
> Compared with other asynchronous FL algorithms such as FedAsync (Xie et al., 2019) and FedBuff (Nguyen et al., 2022), FedCompass alleviates the bias towards faster clients by using group aggregation to reduce global aggregation frequency and avoids the frequent application of staleness factors. We would like to further elaborate this via a simple example:
>
> Suppose that client A is 4 times faster than client B. In FedAsync, if A and B perform the same number of local steps in each local training round, then client A will perform 4 times more local training rounds than client B. When client B’s model arrives, the global model has already been updated four times using A’s model, so B’s local model will become very stale/outdated, and another penalty staleness factor (e.g., say 0.25) will be applied to B’s model during the aggregation. In such a case, B not only performs fewer updates, but its model also experiences an additional penalty factor to further cause the global model to drift away from it. For FedBuff, although it has a buffer to reduce the global update frequency, it is very possible that the buffer contains several local models from the same fast clients and slower clients may also experience some staleness factor. However, for FedCompass, we are grouping the arrival of clients, so in this example, A will perform 4 times more local steps than B and they will arrive almost simultaneously at the server for a group aggregation. In this case, no staleness factor (i.e., that 0.25) is applied to B, so the drift towards A is mitigated compared to other methods.
>
> Of course, from the above description, it is clear that FedCompass only mitigates the drift caused by the stateless factor but does not eliminate it, as client A still performs 4 times more steps than B. The drift can be further mitigated by combining FedCompass with other strategies such as normalized averaging (Nova) proposed in FedNova (Wang et al., 2020). Nova basically normalizes the client’s local gradient according to the number of local steps to mitigate objective bias towards faster clients. Now we also include the experiment results for this combination (FedCompss+N) in the last row of Tables 1 and 2 of the revised paper (which is also the response to another review’s comments). However, it is noteworthy that there is still a tradeoff here: though Nova can mitigate the objective bias, it also diminishes the updates from the faster nodes and may sometimes even lead to slower convergence of the global model.
>
> **Questions about equilibrium**
>
> Equilibrium means that the number of existing groups and the group assignments do not change. Take the scenario in Figure 1 as an example. Initially, there are two groups {1,2,3} and {4,5}. At 12:00, when clients {1,2,3} finish local training together, the Compass scheduler assigns them to the group of clients 4 and 5 and now all clients are within the same group. Then, we can see that all the clients will continue the training following the same grouping (assuming no drastic changes in the computing speed of clients during training), and the number of local steps performed by each client is proportional to the computing speed of each client.
>
> Xie, C., Koyejo, S., & Gupta, I. (2019). Asynchronous federated optimization. arXiv preprint arXiv:1903.03934.
>
> Nguyen, J., Malik, K., Zhan, H., Yousefpour, A., Rabbat, M., Malek, M., & Huba, D. (2022, May). Federated learning with buffered asynchronous aggregation. In International Conference on Artificial Intelligence and Statistics (pp. 3581-3607). PMLR.
>
> Wang, J., Liu, Q., Liang, H., Joshi, G., & Poor, H. V. (2020). Tackling the objective inconsistency problem in heterogeneous federated optimization. Advances in neural information processing systems, 33, 7611-7623.

---

> ### Author Response · Authors · 2023-11-14
> **Response to Reviewer 7w8f (Part 2 of 2)**
>
> **Different behaviors for clients arriving before and after $T_{\max}$**
>
> All clients within the same group are expected to arrive at almost the same time for a group aggregation. Therefore, if a client arrives earlier than the latest arrival time $T_{\max}$, then the server will wait for other pending clients (or $T_{\max}$, whichever is earlier) for a group aggregation. Then the clients can obtain new training tasks with the **most updated global model**. However, if a client arrives later than $T_{\max}$, it means that all other clients within the same group have already finished the group aggregation after waiting until $T_{\max}$. Therefore, there is no point in waiting and the server will immediately assign new training tasks to the client. We think the confusion is mainly caused by the unclarity in the original draft and we hope the revised version conveys the idea more clearly.
>
> **Restrictive assumptions for convergence analysis**
>
> The first three assumptions are commonly used in convergence analysis of non-convex functions in federated learning and distributed learning literature (Stich, 2019; Li et al., 2019, Yu et al., 2019, Nguyen et al., 2022). In terms of the bounded heterogeneity and staleness assumption, it is valid and reasonable in cross-silo FL, as each client is considered to be reliable. We would also like to clarify that the main focus/contribution of our convergence analysis section is deriving the convergence behavior of our newly proposed semi-asynchronous algorithm FedCompass using the commonly-used assumptions and obtaining further insights (such as the impacts of different hyperparameters/setting on convergence), instead of trying to get rid of any commonly-used assumption such as bounded staleness. Additionally, this paper actually first studies convergence behavior of a semi-asynchronous FL algorithm with a scheduler, and we really believe this is still a good contribution to the FL community.
>
> **Citations to other related works**
>
> Thanks a lot for providing those additional papers. We find them relevant and useful and have cited them in suitable places in the revised paper.
>
> **Summary of changes in the paper in response to Reviewer 7w8f**
>
> (1) Rewrite Section 3.2 for better presentation clarity, and have the descriptions focus more on higher-level design ideas.
>
> (2) Add one sentence at the end of Section 3.1 to explain the property of equilibrium - fixed number of existing groups and group assignments.
>
> (3) Cite additional papers at suitable places.
>
>
> Stich, S. U. (2018). Local SGD converges fast and communicates little. arXiv preprint arXiv:1805.09767.
>
> Li, X., Huang, K., Yang, W., Wang, S., & Zhang, Z. (2019). On the convergence of fedavg on non-iid data. arXiv preprint arXiv:1907.02189.
>
> Yu, H., Yang, S., & Zhu, S. (2019, July). Parallel restarted SGD with faster convergence and less communication: Demystifying why model averaging works for deep learning. In Proceedings of the AAAI Conference on Artificial Intelligence (Vol. 33, No. 01, pp. 5693-5700).
>
> Nguyen, J., Malik, K., Zhan, H., Yousefpour, A., Rabbat, M., Malek, M., & Huba, D. (2022, May). Federated learning with buffered asynchronous aggregation. In International Conference on Artificial Intelligence and Statistics (pp. 3581-3607). PMLR.

---

> > ### Comment · Reviewer_7w8f · 2023-11-23
> > **Post Rebuttal**
> >
> > I would like to thank the reviewers for addressing my concerns. After reading carefully the updated version of their manuscript and the comments from the other reviewers I find that the revision of section 3.2 has significantly improved the presentation of their work and the additional experiments that have been included exhibit the strengths of the proposed method and increase the impact of this paper. As a result I have decided to increase my score to 6. Having said that I still believe that the theoretical contribution of this work is to some extent limited.

---

> > > ### Author Response · Authors · 2023-11-23
> > >
> > > Dear Reviewer 7w8f,
> > >
> > > We greatly appreciate your response and we are glad that most of your concerns are addressed.

---

### Official Review · Reviewer_UVPW · 2023-11-07

**Soundness:** 2 fair
**Presentation:** 3 good
**Contribution:** 2 fair
**Rating:** 5
**Confidence:** 3

**Summary:**

The paper presents a way of tracking the computation speed of different clients in federated learning (FL), based on which a time threshold is determined to collect the clients' updates with possibly different numbers of local updates. Compared to fully asynchronous FL and some semi-asynchronous FL baselines, the experimental results show that the proposed FedCompass algorithm gives better performance.

**Strengths:**

- This work tackles the important problem of system heterogeneity in FL, focusing on the cross-silo FL setup.

**Weaknesses:**

- The adaptation of the number of local updates based on clients' computation speed is not new. An important and well-known baseline is FedNova (Wang et al., 2020), which has been cited in the paper but not compared against. Compared with the proposed FedCompass method, FedNova is much easier to implement and supports both cross-silo FL and cross-device FL with partial client participation. The advantage of FedCompass over FedNova is not clear.

- The proposed FedCompass approach is heuristic and a lot of its description is based on simplified examples. The paper argues that FedBuffer includes the buffer size $K$ as a hyper parameter that needs to be tuned. However, FedCompass also includes hyper parameters $Q_\mathrm{min}$, $Q_\mathrm{max}$, and parameters in sub-procedures, such as $\lambda$ in Algorithm 2, that need to be tuned heuristically. It is worth noting that the experiments in this paper use FedBuffer with small values of $K$ up to 5 (as listed in Appendix E) while the original FedBuffer paper (Nguyen et al., 2022) uses $K=10$ in the majority of experiments. It is not clear how the authors selected $K$ in the experiments presented in this paper, but I would expect that FedBuffer with a properly tuned $K$ would give a similar performance as FedCompass with properly tuned hyper parameters.

    Side note: The algorithm is called FedBuff in the original paper (Nguyen et al., 2022), not sure why the authors call it FedBuffer in this paper.

- In practice, the clients' computation speeds can vary over time due to the varying amount of concurrent tasks running in the system. The paper has a mentioning of this but only studies the behavior of the algorithm in some oversimplified cases of computation speed variation. It is unclear whether the scheduling procedure in Section 3.1 is robust to all types of speed variation, or does there exist a worst case scenario where the scheduler completely loses track in its estimation of clients' computation speeds. This needs a separate theoretical analysis IMHO, which is different from the convergence bound.

- The convergence result appears fairly straightforward, since results with bounded staleness and different numbers of local updates both exist in the literature. It is not quite clear what is the challenge and novelty in the convergence analysis.

**Questions:**

Basically all the things mentioned under weaknesses. Some specific questions include:
- How does FedCompass compare with FedNova?
- How is $K$ chosen for FedBuffer in the experiments? Was any hyper parameter optimization or grid search implemented?
- Is it possible to theoretically show the robustness of the scheduler when the clients' computation speeds can vary arbitrarily?
- What is the challenge and novelty in the convergence analysis?

---

> ### Author Response · Authors · 2023-11-14
> **Response to Reviewer UVPW (Part 1 of 3)**
>
> Dear Reviewer UVPW,
>
> First and foremost, we would like to express our sincere gratitude for taking the time to review our paper and providing insightful questions and comments. Our responses to your questions are as follows.
>
> **Comparison between FedCompass and FedNova**
>
> The main focus of FedNova (Wang et al., 2020) is how to eliminate objective inconsistencies when the number of local steps for clients is different (via adjusting weighting factors). In practice, FedNova is typically implemented in the following two **synchronous** manners:
> (1) Each client performs the same number of local epochs, thus different number of local steps due to different data sizes. In this case, the algorithm suffers a lot from client computing speed variations like the FedAvg algorithm. In the worst case, the slowest client may have the largest dataset, and the whole training process can be extremely slow.
> (2) Setting a universal completion time in each round and letting each client perform local updates until the completion time is reached. This approach eliminates the long waiting time and also has a simple implementation process, but it requires the universal completion time to be manually selected for different training tasks, model sizes, and average client computing capabilities.
>
> On the other hand, FedCompass is designed to be an **asynchronous** federated learning algorithm, and focuses on how to employ a scheduler to have asynchronous clients arrive almost simultaneously. This is achieved by assigning different numbers of steps dynamically using real-time performance metrics. This design choice, while marginally complicating the implementation, significantly streamlines the experimental setup, as users only need to provide a range of desired local update steps in each round (which is more intuitive to users) and do not need to empirically estimate a suitable universal completion time. There is a tradeoff between ease of implementation and usage convenience.
>
> Furthermore, we would like to clarify that FedNova and FedCompass are not analogous but actually “orthogonal” to each other. As FedCompass involves different client local steps in each round, it can be combined with FedNova’s normalized averaging strategy as a manner to further mitigate the drifting to faster clients. To further emphasize the FedNova paper and explain the orthogonal relationship between FedCompass and FedNova, we now provide a new set of experiment results for their combination in the last rows of Tables 1 and 2 (FedCompass+N) of the revised paper (as well as all result tables and figures in the Appendix).
>
> Wang, J., Liu, Q., Liang, H., Joshi, G., & Poor, H. V. (2020). Tackling the objective inconsistency problem in heterogeneous federated optimization. Advances in neural information processing systems, 33, 7611-7623.

---

> ### Author Response · Authors · 2023-11-14
> **Response to Reviewer UVPW (Part 2 of 3)**
>
> **Hyperparameter problems for FedBuff and FedCompass**
>
> First, thanks for the reminder to adopt FedBuff instead of FedBuffer for consistency with the original paper (Nguyen et al., 2022). We have changed all FedBuffer to FedBuff in the revised paper now.
>
> Second, we would like to answer why we do not simply use $K=10$ as the original FedBuff paper does, especially given that their paper mentions that “Our extensive empirical evaluation finds that $K=10$ is a good setting across benchmarks and does not require tuning”. This is because all the experiments in FedBuff are in cross-device FL settings, and have **at least thousands of clients in all experiment settings** (CelebA dataset: 9343 clients, Sent140: 660120 clients, and CIFAR-10: 5000 clients). However, that statement does not hold for cross-silo cases, as we have less than or equal to 10 clients in most experiments. When the number of clients becomes small, the users have to decide the buffer size themselves instead of directly using $K=10$.
>
> Then, we would like to describe how we decide the value of $K$ for different numbers of clients. We use the *class partitioned* MNIST dataset with {5, 10, 20} clients and normal distribution as the client heterogeneity setting. For 5 clients, we search $K$ among values {2,3}, for 10 clients, we search $K$ among {2, 3, 5}, and for 20 clients, we search $K$ among {3, 5, 10}. We select $K$ with the highest average accuracy of three independent runs. According to the table below, we finally select $K=3/3/5$ for 5/10/20 clients. For Fed-IXI, we choose $K=2$ as there are only three clients. For Fed-ISIC2019 with 6 clients, we choose $K=3$ as we use this buffer size for both 5 and 10 clients. Additionally, we would like to mention that from the training process visualization figures (Figure 2 and more in Appendix F), we can clearly see that there is a non-marginal gap between the FedCompass curve and the FedBuff curve in many cases, especially for the relatively complicated datasets, CIFAR10 and Fed-ISIC2019.
>
> | Number of Clients | K=2     | K=3     | K=5     | K=10    |
> |-------------------|---------|---------|---------|---------|
> | 5 clients         | 94.13   | 94.77   |    -     |    -     |
> | 10 clients        | 93.88   | 94.60   | 94.54   |    -    |
> | 20 clients        |    -     | 97.65   | 98.21   | 98.16   |
>
> Finally, regarding the hyperparameter tuning, we would like to elaborate on the reason why we state that FedCompass requires easier tuning. For FedCompass, it has three algorithm-specific hyperparameters: $Q_{\min}$, $Q_{\max}$, and $\lambda$, or equivalently, $Q_{\max}$, $Q_{\min}/Q_{\max}$, and $\lambda$. For FedBuff, it requires $Q$ and $K$. $Q_{\max}$ in FedCompass and $Q$ in FedBuff are both used to define the amount of computational work we want each client to do in each round, so they cancel each other out. The ratio $Q_{\min}/Q_{\max}$ is irrelevant to the number of clients, and we empirically show in the ablation study that a large range of values can lead to faster convergence than FedBuff (we now also added new ablation study results on the CIFAR-10 datasets to address another reviewer’s request, and the observation remains on the more difficult CIFAR-10 dataset). Similarly, for $\lambda$, its impact on the convergence is also minor and a number slightly above 1 is fine. On the other hand, for $K$ in FedBuff, though their paper mentions $K=10$ is a good choice when the client numbers are large, there does not exist any rule of thumb for selecting $K$ when the number of clients is small in cross-silo settings, so we had to do some small preliminary experiments to choose $K$ for different numbers of clients. For FedCompass, we simply choose $Q_{\min}/Q_{\max}=0.2$ and $\lambda=1.2$ for all experiments (which are not even optimal in some cases), and FedCompass still outperforms FedBuff all the time with a non-marginal gap.
>
> Nguyen, J., Malik, K., Zhan, H., Yousefpour, A., Rabbat, M., Malek, M., & Huba, D. (2022, May). Federated learning with buffered asynchronous aggregation. In International Conference on Artificial Intelligence and Statistics (pp. 3581-3607). PMLR.

---

> ### Author Response · Authors · 2023-11-14
> **Response to Reviewer UVPW (Part 3 of 3)**
>
> **FedCompass’s behavior with speed changes**
>
> Different from cross-device FL where each client may experience frequent speed changes due to power constraints, the operation of other concurrent applications, or network connectivity issues. In practice, a cross-silo FL client usually has dedicated computing resources such as HPC with a job scheduler or some cloud computing facilities—where significant changes in computing speed are less frequent, generally arising from events like auto-scaling. Our experiment design in Appendix G.2 is an overestimate of the practical cases, where each client has a 10% probability in each round to resample a “speed” from the client speed distribution (a simulation for auto-scaling behaviors), and we think that this setup could be a reasonable simulation for speed variations in practical cross-silo FL settings. Of course, in the worst-case scenario where all client speeds change dramatically all the time, then the scheduling algorithm of FedCompass will fail to synchronize the client arrival as desired, since the previous speed information becomes obsolete. In such a scenario, the training will go completely asynchronously and FedCompass will downgrade to the FedAsync algorithm (Xie et al., 2019).
>
> **Convergence analysis [updated]**
>
> (1) The main focus/contribution of our convergence analysis section is deriving the convergence behavior of our newly proposed semi-asynchronous algorithm FedCompass using the commonly-used assumptions and obtaining further insights (such as the impacts of different hyperparameters/settings on the convergence), instead of trying to get rid of any commonly-used assumptions such as bounded staleness.
>
> (2) Though we are not the first ones to study the convergence behavior of the general asynchronous FL algorithms or FL algorithms with different client local steps, this paper actually first studies convergence behavior of a semi-asynchronous FL algorithm with a scheduler, and we believe this is still a good contribution to the FL community.
>
> **Summary of changes in the paper in response to Reviewer UVPW**
>
> (1) Add experiment results for combining FedCompass and normalized averaging (Nova) to further clarify the “orthogonal” relationship between FedCompass and FedNova in Tables 1 and 2.
>
> (2) Add a few sentences at the end of the first paragraph in Appendix E.2 to explain the choice of buffer size K and the reason why we do not simply choose $K=10$.
>
> (3) Add sentences at the end of Appendix G.2 to explain that our experiment design is aimed to mimic the occasional yet substantial changes in speed that may arise in real-world cross-silo federated learning environments as a result of auto-scaling. We also point out the limitations of FedCompass if all clients experience dramatic and very frequent speed changes.
>
>
>
> Xie, C., Koyejo, S., & Gupta, I. (2019). Asynchronous federated optimization. arXiv preprint arXiv:1903.03934.

---

> > ### Comment · Reviewer_UVPW · 2023-11-18
> >
> > Thanks for the additional explanation and experiments. I will update my score to 5.
> >
> > Some further concerns remain:
> > - "FedCompass is designed to be an asynchronous federated learning algorithm, and focuses on how to employ a scheduler to have asynchronous clients arrive almost simultaneously." This sentence is somewhat self-contradictory. When all the clients arrive simultaneously, it becomes a synchronous algorithm, although the specific parameter that is being synchronized here may be different from other synchronous algorithms.
> > - Since the scheduling mechanism is essentially heuristic, I would expect it to be evaluated in more realistic settings. Currently, the system is simulated with parameters derived from some simple models. It is hard to tell whether the performance will be similar in real systems or not. There are other open questions, such as whether the algorithm will be robust to drastic changes in the system, which are also unclear from the current evaluation.
> > - Algorithm 1 is quite complicated. It seems to be a carefully engineered approach probably designed by someone with strong hands-on experience. However, it is generally difficult for most people in the research community to understand the rationale and key insights behind such a design. Since the scheduling algorithm is not backed by theory, we do not know whether it is optimal or not, we do not even know what is the concrete objective that the scheduling algorithm is trying to optimize.
> >
> > Since most of my research papers have a theoretical emphasis, I feel it a bit difficult for me to evaluate a paper like this, so I will reduce my confidence to 3.

---

> > > ### Author Response · Authors · 2023-11-18
> > > **Response to Further Comments from Reviewer UVPW (Part 1 of 2)**
> > >
> > > Dear Reviewer UVPW,
> > >
> > > Thank you for your response and for increasing our score; we genuinely appreciate it! We would like to address your additional concerns as outlined below.
> > >
> > > **Why do we mention the asynchronous algorithm is trying to have clients arrive almost simultaneously?**
> > >
> > > In our work, we describe the FedCompass algorithm as a semi-asynchronous algorithm, as indicated in both our abstract and the main text. This term signifies that FedCompass is an algorithm that lies between synchronous and asynchronous algorithms.  Here are more detailed descriptions of this semi-asynchronous nature.
> > >
> > > (1) In distributed or parallel computing, synchrony implies that the server waits, blocking further action, until it receives messages/updates from all clients. Conversely, asynchrony entails that the server responds to each client immediately upon receiving their individual message/update, without any delay.
> > >
> > > (2) For FedCompass, the synchronous aspect comes from the fact that the server orchestrates the arrival of each client within a group to be nearly simultaneous, and waits for all the clients within the group for a grouped aggregation.
> > >
> > > (3) On the other hand, FedCompass also has several asynchronous features:
> > >
> > > - At the beginning of the FedCompass execution, when the server has no prior information about client speeds, all clients operate asynchronously. Namely, the server sends the global model back immediately after receiving the first client update.
> > >
> > > - With the help of the scheduler, FedCompass aims to minimize the server blocking time. Generally, unless there are significant fluctuations, this blocking time is brief. From a client's perspective, it appears as though the server almost immediately responds after receiving updates (as an asynchronous server).
> > >
> > > - Unlike fully synchronous algorithms where the server waits for every client, FedCompass has a mechanism for dealing with delays. If a client fails to arrive before the latest arrival time (T_max in our algorithm) due to severe slowdowns, the server proceeds without further waiting, responding to other clients that have arrived. The server will also respond immediately to the delayed client upon its arrival, without any additional waiting.
> > >
> > > We hope these points offer a clearer understanding of the semi-asynchronous nature of the FedCompass algorithm.
> > >
> > > **Why did we opt not to evaluate FedCompass in a realistic setting?**
> > >
> > > There are three primary reasons behind our decision to use common distributions for simulating client speeds, rather than employing real heterogeneous machines.
> > >
> > > (1) Simulations enable the reproducibility of experiment results. By utilizing specific distribution parameters and random seeds, anyone can accurately recreate the client heterogeneity in our experiments. This level of control is not feasible with real-world heterogeneous machines, even when all machine metrics are provided.
> > >
> > > (2) Simulations allow for a completely fair and equitable performance comparison across various FL algorithms. By fixing the distribution parameters and random seeds, we can ensure consistent client speed distribution during evaluations. In contrast, using actual machines introduces unavoidable variations in computing capabilities across different experiment runs and makes it impossible for a fair comparison.
> > >
> > > (3) Employing simulations, rather than real-world client scenarios, is a widely accepted approach in FL research. For instance, FedAsync (Xie et al., 2019) simulates synchrony by sampling staleness from a uniform distribution. Similarly, FedBuff (Nguyen et al., 2022) uses an exponential distribution for client speeds, and FedNova (Wang et al., 2020) utilizes a uniform distribution to determine the number of local steps each client performs in a training round, as seen in their official GitHub implementation (https://github.com/JYWa/FedNova/blob/47b4e096dfb19dc43c728896fff335a5befb645d/train_LocalSGD.py#L292).
> > >
> > > In light of these reasons and in alignment with established practices in prior works, we chose not to evaluate FedCompass in real-world heterogeneous settings.
> > >
> > > Xie, C., Koyejo, S., & Gupta, I. (2019). Asynchronous federated optimization. arXiv preprint arXiv:1903.03934.
> > >
> > > Nguyen, J., Malik, K., Zhan, H., Yousefpour, A., Rabbat, M., Malek, M., & Huba, D. (2022, May). Federated learning with buffered asynchronous aggregation. In International Conference on Artificial Intelligence and Statistics (pp. 3581-3607). PMLR.
> > >
> > > Wang, J., Liu, Q., Liang, H., Joshi, G., & Poor, H. V. (2020). Tackling the objective inconsistency problem in heterogeneous federated optimization. Advances in neural information processing systems, 33, 7611-7623.

---

> > > ### Author Response · Authors · 2023-11-18
> > > **Response to Further Comments from Reviewer UVPW (Part 2 of 2)**
> > >
> > > **Evaluating the algorithm under drastic speed variations**
> > >
> > > In our ablation study on FedCompass’s resilience to client speed changes in Appendix G.2, every client has a 10% chance of randomly sampling a new speed from the established speed distribution, irrespective of its previous speed. Consequently, when client speeds are modeled using an **exponential distribution**, there's a substantial likelihood of encountering drastic speed changes—some clients might experience speed alterations by a factor of 5x or even 10x. Take our experiments with the MNIST dataset as an example. By calculating the area under the curve of the exponential distribution's probability density function, we find that each client has a 15.35% chance of operating at speeds less than 0.025 seconds per update. Conversely, there's an 18.89% probability for speeds to exceed 0.25 seconds per update, which translates to more than a 10x difference in speed. These numbers intuitively illustrate the potential for drastic speed variations in our evaluation settings, emphasizing the robustness and adaptability of FedCompass in handling such extreme changes.
> > >
> > > **Theoretical foundation of the FedCompass scheduler**
> > >
> > > First, our scheduler aims to minimize the blocking time in a group of clients by varying the number of local steps, facilitating their near-simultaneous arrival. The contribution of this design lies in three key aspects:
> > >
> > > (1) The scheduler introduces a novel approach for running FL experiments in a semi-asynchronous manner, which combines the benefits of synchronous algorithms (less client drift) with the quick convergence typical of asynchronous algorithms.
> > >
> > > (2) The separation of the scheduling and the server aggregation makes FedCompass have the compatibility with a wide range of FL optimization techniques found in the literature, including server-side optimizations (Hsu et al., 2019; Reddi et al., 2020) and normalized averaging (Wang et al., 2020).
> > >
> > > (3) FedCompass has shown superior performance across various datasets and client heterogeneity settings empirically and also comes with proof for convergence under certain assumptions.
> > >
> > > Regarding the scheduler's optimality, proving such a claim is exceedingly challenging, if not impossible. The scheduler designs inherently involve trade-offs, and defining a universally "optimal" scheduler is complex. In FedCompass, this trade-off manifests in balancing synchrony and asynchrony. Also, we would like to mention that the trade-off happens in the scheduler design across various computer science fields. For instance, the SLURM job scheduler (Yoo et al., 2003), widely used for resource management in HPC clusters and supercomputers, is based on a complex finite state machine without a formal "optimality" proof. Similarly, in the realm of the operating system CPU schedulers, numerous algorithms exist (https://www.geeksforgeeks.org/comparison-of-different-cpu-scheduling-algorithms-in-os/), each with its trade-offs and none universally acknowledged or formally proved as the optimal.
> > >
> > > In conclusion, while this paper does not present proof of the scheduler's optimality, considering the aforementioned advantages, we believe that FedCompass is still a great contribution to the FL community.
> > >
> > > Hsu, T. M. H., Qi, H., & Brown, M. (2019). Measuring the effects of non-identical data distribution for federated visual classification. arXiv preprint arXiv:1909.06335.
> > >
> > > Reddi, S., Charles, Z., Zaheer, M., Garrett, Z., Rush, K., Konečný, J., ... & McMahan, H. B. (2020). Adaptive federated optimization. arXiv preprint arXiv:2003.00295.
> > >
> > > Wang, J., Liu, Q., Liang, H., Joshi, G., & Poor, H. V. (2020). Tackling the objective inconsistency problem in heterogeneous federated optimization. Advances in neural information processing systems, 33, 7611-7623.
> > >
> > > Yoo, A. B., Jette, M. A., & Grondona, M. (2003, June). Slurm: Simple linux utility for resource management. In Workshop on job scheduling strategies for parallel processing (pp. 44-60). Berlin, Heidelberg: Springer Berlin Heidelberg.

---

> > > > ### Comment · Reviewer_UVPW · 2023-11-21
> > > >
> > > > Thanks for these additional answers. A few more comments are as follows:
> > > > - Evaluation in realistic setting: I agree that simulations are helpful for controllable experiments, but it can be made more realistic by simulating the processing speed etc. using real-world data traces collected from data centers. Some public datasets of this kind exist, such as https://research.google/resources/datasets/google-cluster-workload-traces-2019/. As of now, it is not clear how well the random distributions match with real system behavior. The related works mentioned in the response have strong theoretical contributions, which this paper lacks. This is why I would emphasize on experiments especially for this paper.
> > > > - Theoretical foundation: It should be possible to at least formulate an objective that the proposed algorithm tries to optimize, although it may not be possible to optimize the objective exactly, which is why the heuristic is needed. This would greatly improve the readability of the paper.

---

> > > > > ### Author Response · Authors · 2023-11-21
> > > > >
> > > > > Dear Reviewer UVPW,
> > > > >
> > > > > We greatly appreciate your time and effort in the continuous discussions about our paper! Here are our responses to your further comments.
> > > > >
> > > > > First, we concur with your suggestion regarding the need to clarify the objectives of the Compass scheduler. To this end, we have revised the motivation of FedCompass (first paragraph of Section 3). We added a sentence emphasizing that the scheduler aims to strike a balance between the time efficiency of asynchronous algorithms (faster convergence) and the objective consistency of synchronous ones (i.e., less drift and higher accuracy). We believe this could help the readers better understand the design philosophy of the scheduler.
> > > > >
> > > > > However, formulating the above mentioned objectives is still quite challenging as the time efficiency of asynchronous FL algorithms is usually demonstrated in its wall-clock time speed up towards convergence, and this usually requires empirical study to justify the efficiency. The objectives do necessitate extensive empirical evidence to substantiate the claim that the scheduler achieves a good balance/tradeoff between synchronous and asynchronous algorithms. Therefore, we perform a comprehensive set of experiments by examining all factors impacting the convergence behavior of FedCompass, as delineated in Equations 1 and 2:
> > > > >
> > > > > (1) Number of Clients: We evaluate different client numbers using artificially partitioned datasets and real-world federated datasets.
> > > > >
> > > > > (2) Client Heterogeneity: We test three heterogeneous settings to assess performance under varied conditions.
> > > > >
> > > > > (3) Data Heterogeneity: We employ two partition strategies for generating non-IID data from centralized datasets, supplemented with real-world heterogeneous distributed datasets.
> > > > >
> > > > > Furthermore, we want to delve deeper into our client heterogeneity simulation choices. Using certain random distributions such as exponential distribution to simulate the client computing speeds is also prevalent in several engineering-focused papers (Charles et al., 2021, Reisizadeh et al., 2022), so we still believe that our experiment findings could serve as strong evidence for the effectiveness and efficiency of the proposed algorithm.
> > > > >
> > > > > Charles, Z., Garrett, Z., Huo, Z., Shmulyian, S., & Smith, V. (2021). On large-cohort training for federated learning. Advances in neural information processing systems, 34, 20461-20475.
> > > > >
> > > > > Reisizadeh, A., Tziotis, I., Hassani, H., Mokhtari, A., & Pedarsani, R. (2022). Straggler-resilient federated learning: Leveraging the interplay between statistical accuracy and system heterogeneity. IEEE Journal on Selected Areas in Information Theory, 3(2), 197-205.

---

### Meta-Review · Area_Chair_nJDD · 2023-12-11

**Metareview:**

The authors propose FedCompass, a semi-asynchronous federated learning algorithm with a computing power-aware scheduler, which adaptively assigns varying amounts of training tasks to different clients. The improved convergence properties of the algorithm are established through guarantees (that make rather strong assumptions, albeit common within relevant literature), and supported through extensive experiments

This is generally a very densely populated area; understandably, the reviewers raised several concerns about novelty, mentioning explicitly papers the authors should constrast against. The authors have incorporated these suggestions both by revising the paper but also incorporating a large amount of experiments during the rebuttal phase, all of which have improved the paper significantly. It would be great if additional discussions/arguments that arose during the discussion phase (e.g., on the motivation) also appear somewhere in the paper, at the supplement if necessary.

**Justification For Why Not Higher Score:**

The topic is somewhat saturated, and this is "yet another paper on", regardles...

**Justification For Why Not Lower Score:**

...the paper does advance the state of the art, is a solid effort, and moves the needle forward both in terms of theory and experimentation.

---

### Decision · Program_Chairs · 2024-01-16

Accept (poster)